# Spatial and temporal variability in the response of phytoplankton and bacterioplankton to B-vitamin amendments in an upwelling system

Vanessa Joglar[1*], Antero Prieto[1], Esther Barber-Lluch[1], Marta Hernández-Ruíz[1], Emilio Fernández[1] and Eva Teira[1]

[1] Centro de Investigación Mariña da Universidade de Vigo (CIM-UVigo), Departamento Ecoloxía e Bioloxía Animal, Universidade de Vigo, Campus Lagoas-Marcosende, Vigo, 36310, Spain

*Correspondence to*: Vanessa Joglar +34 986 818790 (vjoglar@uvigo.es)

**Abstract.** We experimentally evaluated the temporal (inter-day and inter-season) and spatial variability in microbial plankton responses to vitamin B12 and/or B1 supply (solely or in combination with inorganic nutrients) in coastal and oceanic waters of the northeast Atlantic Ocean. Phytoplankton and, to a lesser extent, prokaryotes were strongly limited by inorganic nutrients. Inter-day variability in microbial plankton responses to B-vitamins was limited compared to inter-season variability, suggesting that B-vitamins availability might be partially controlled by factors operating at seasonal scale. Chlorophyll-*a* (Chl-*a*) concentration and prokaryote biomass (PB) significantly increased after B-vitamin amendments in 13 % and 21 %, respectively, of the 216 cases (36 experiments × 6 treatments). Most of these positive responses were produced by treatments containing either B12 solely or B12 combined with B1 in oceanic waters, which was consistent with the significantly lower average vitamin B12 ambient concentrations compared to that in the coastal station. Negative responses, implying a decrease in Chl-*a* or PB, represented 21 % for phytoplankton and 26 % for prokaryotes. Growth stimulation by B1 addition was more frequent on prokaryotes than in phytoplankton, suggesting that B1 auxotrophy in the sampling area could be more widespread in prokaryotes than in phytoplankton. Negative responses to B-vitamins were generalized in coastal surface waters in summer, and were associated to a high contribution of Flavobacteriales to the prokaryote community. This observation suggests that the external supply of B12 and/or B1 may promote negative interactions between microbial components when B-vitamin auxotrophs are abundant. The microbial response patterns to B12 and/or B1 amendments were significantly correlated with changes in the prokaryotic community composition, highlighting the pivotal role of prokaryotes in B-vitamins cycling in marine ecosystems.

## 1 Introduction

Phytoplankton accounts for almost half of the global net primary production (Field et al., 1998) and may eventually cause toxic episodes, such as those caused by harmful algae blooms of *Alexandrium* spp. or *Gymnodinium* spp., entailing human health problems and large economic losses (Hallegraeff, 1993; van Dolah et al., 2001). Recent emerging evidence suggests the role of biologically active organic compounds, such as B-vitamins, on the control of marine productivity in both coastal and oceanic waters (Panzeca et al., 2006; Bertrand et al., 2007; Gobler et al., 2007; Koch et al., 2011; Browning et al., 2017, 2018). B-vitamins act as cofactors for enzymatic reactions and are involved in many important metabolic pathways (Madigan et al., 2005; Koch et al., 2011; Monteverde et al., 2017). Vitamin B12 (B12 herein), which is exclusively synthesized by some bacteria and archaea (Roth et al., 1996; Martens et al., 2002; Warren et al., 2002), acts as a cofactor of three enzymes in eukaryotes (methionine synthase, methylmalonyl-coA mutase and ribonucleotide reductase type II) (Helliwell et al., 2011; Bertrand and Allen, 2012). In comparison, over 20 different B12-dependent enzymes are found in bacteria (Roth et al., 1996), making B12 critically important also for these organisms. Vitamin B1 (B1 herein) plays a pivotal role in intermediary carbon metabolism and is a cofactor for a number of enzymes involved in primary carbohydrate and branched-chain amino acid metabolism (Croft et al., 2006).

Most eukaryote phytoplankton species are auxotrophs for one or more B-vitamins, consequently requiring an exogenous supply of these molecules (Bertrand and Allen, 2012; Carlucci and Bowes, 1970; Haines and Guillard, 1974; Helliwell et al., 2011). Moreover, genomic data also indicate widespread B-vitamins auxotrophy among many bacterial taxonomic groups (Sañudo-Wilhelmy et al., 2014; Paerl et al., 2018), which implies that phytoplankton and bacterioplankton may eventually compete for the

acquisition of these compounds (Koch et al., 2012). Auxotrophic microorganisms may acquire the required vitamins from the environment or through biotic interactions with prototrophic (biosynthetically competent) microorganisms (Droop, 2007; Grant et al., 2014; Kazamia et al., 2012). A well-known example is the mutualistic interaction between B12 or B12 and B1 dependent phytoplankton and bacterioplankton (Croft et al., 2005; Amin et al., 2012; Cooper and Smith, 2015; Cruz-López and Maske, 2016).

Even though B-vitamins appear to be important and potentially limiting factors for microbial plankton, our understanding of B-vitamins cycling in the ocean is largely limited by the complex and still evolving analytical methodology for its quantification in natural waters (Okbamichael and Sañudo-Wilhelmy, 2004, 2005; Suffridge et al., 2017). Sañudo-Wilhelmy et al. (2012) found extensive areas of coastal waters with close to undetectable B12 concentrations, suggesting that microbes might be well adapted to thrive under limiting conditions for this growth factor.

The factors limiting phytoplankton and bacterial growth in marine ecosystems are known to vary over different spatial and temporal scales (Cullen et al., 1992; Arrigo, 2005; Martínez-García et al., 2010b; Moore et al., 2013), in accordance with the dynamic nature of microbial communities (Pinhassi et al., 2003; Fuhrman et al., 2008; Hernando-Morales et al., 2018). Compared to mineral nutrient and trace elements, much less is known about B vitamin limitation and its spatial and temporal variability in marine ecosystems.

Some studies have shown enhanced phytoplankton biomass associated to B12 amendments in both temperate coastal and polar waters (Bertrand et al., 2007; Gobler et al., 2007; Koch et al., 2011, 2012). The simultaneous effect of vitamin B12 supply on both phytoplankton and bacteria has been barely explored (Koch et al., 2011, Barber-Lluch et al., 2019). To our knowledge, the effect of B1 amendments on marine natural microbial plankton community succession has been only assessed by Gobler et al. (2007),

who suggested that high concentration of B-vitamins, associated with high bacterial
abundance, caused an increase in auxotrophs, mostly dinoflagellates.
The Ría de Vigo (NW Spain) is a coastal embayment affected by intermittent upwelling
of subsurface cold and inorganic nutrient-rich water from March to September and the
downwelling of open ocean surface water from October to March (Fraga, 1981; Barton
et al., 2015). In addition to this seasonality, fluctuations of wind patterns in the area
generate upwelling and downwelling events occurring within each season (Alvarez-
Salgado et al., 1993; Figueiras et al., 2002). A recent study by Barber-Lluch et al. (2019)
at a shelf station off the Ría de Vigo (NW Spain) showed monthly variation in the
response of phytoplankton and bacteria to nutrient and/or B12 additions in surface waters,
likely related to variation in the ambient concentration of B12 and the taxonomic
community composition. Unfortunately, these authors did not specifically assess the role
of these factors on the microbial response to the amendments.
Within this context, the aim of our study was to explore spatial (horizontal and vertical)
and temporal (inter-day and inter-season) variability patterns in B12 and B1 vitamin
limitation in relation to the prevailing initial abiotic (e.g., nutrient and B12
concentrations) and biotic (eukaryote and prokaryote community composition)
conditions in this productive ecosystem. We conducted a total of thirty-six microcosm
bioassays in February, April, and August 2016 to evaluate the response of heterotrophic
bacteria and phytoplankton biomasses to the addition of B12 and/or B1.
Considering that a large fraction of eukaryotic phytoplankton and bacterial taxa require
exogenous B-vitamins and considering the different requirements and capabilities to
synthesize B-vitamins by different microbial taxa, we hypothesize that microbial
community composition play a relevant role in explaining B-vitamins limitation patterns
in microbial plankton.

## 2 Methods

### 2.1 Sampling strategy

Thirty-six enrichment experiments were performed in the upwelling system near Ría de Vigo on board "B/O Ramón Margalef" in three different oceanographic cruises (ENVISION I, II & III) conducted in 2016. Two different locations of the East Atlantic Ocean, one coastal station (C) (42º N, 8.88º W) and one oceanic station (Oc) (42º N, 9.06º W) (Fig. 1a), were sampled during three different seasons aimed to cover a wide range of initial hydrographic and ecological conditions. The 10-day cruises were conducted in February (ENVISION I), coinciding with the spring bloom, and April (ENVISION II) and August (ENVISION III) during the early and late summer upwelling, respectively. During each cruise, 12 enrichment experiments were carried out on board, 3 experiments in each station (C-a, C-b & C-c and Oc-a, Oc-b & Oc-c, respectively) with water from two different depths. Each experiment began on the first (day 0), third (day 2) and sixth (day 5) of each cruise for the coast and on the second (day 1), fourth (day 3) and seventh (day 6) of each cruise for the ocean (Fig. 1b, c). Water was collected using 20 l Niskin metal-free bottles. Surface (5 m) and sub-surface chlorophyll maximum (SCM) (between 10 m and 50 m according to the CTD data) samples were taken (Fig. 2a-f). We failed to sample the SCM on two occasions (C-a in February and C-a in April), due to large vertical displacements between the downward and the upward casts. Vertical profiles of temperature, salinity and chlorophyll fluorescence were obtained using a regular stainless CTD-rosette down to 60 m in the coastal station and to 200 m in oceanic station. Samples for chlorophyll-*a* (Chl-*a*), prokaryotic biomass (PB), dissolved nutrient concentration, including vitamin B12, and microbial plankton community were collected at the

beginning (time zero, hereafter referred to as t0) of each enrichment experiment. Daily
upwelling index (UI) values were computed by the Instituto Español de Oceanografía
(www.indicedeafloramiento. ieo.es/) in a 2º x 2º geostrophic cell centered at 42 ºN, 10
ºW, using data from atmospheric pressure at sea level, derived from the WXMAP model
(Gonzalez-Nuevo et al., 2014). Precipitation data was obtained from the Regional
Weather   Forecast   Agency-Meteogalicia   (http://www.meteogalicia.gal)   in   the
meteorological station Illas Cies (ID 10125).
**2.2. Experimental design**
Seawater samples were gently pre-filtered through a 200 μm mesh to exclude large
zooplankton in order to ensure good replicability and collected into a 20 l acid-cleaned
polyethylene carboy. It is important to note that incidental trace-metal contamination
could have occurred during water collection. Following sample collection, 300 ml PAR
and UVR transparent, sterile, and non-toxic (whirl-pak) bags were filled and nutrients
were added establishing eight different enrichment treatments as follows: (1) control
treatment (C); (2) inorganic nutrient treatment (I); (3) vitamin B12 (Sigma, V2876)
treatment; (4) vitamin B1 (Sigma, T4625) treatment; (5) Inorganic nutrients and vitamin
B12 (I+B12) treatment; (6) Inorganic nutrients and vitamin B1 (I+B1) treatment; (7)
vitamins B12 and B1 (B12+B1) treatment and (8) Inorganic nutrients with vitamins B12
and B1 (I+B12+B1) treatment (see Table 1 for details). Inorganic nutrients were added to
avoid that inorganic nutrient limitation masked the responses to B vitamins. The nutrient
concentrations of the additions were the same as previously used in similar enrichment
experiments in the sampling area (Martinez-García et al., 2010a). The amount of B12 and
B1 vitamin experimentally added approximated maximum concentrations previously
observed in coastal areas (Okbamichael and Sañudo-Wilhelmy 2004, 2005, Sañudo-
Wilhelmy et al., 2006). Each treatment had 3 replicates resulting in 24 whirl-pack bags
per experiment. To assess short-term effects of nutrient inputs, experimental bags were
incubated on-deck during 72 h. In-situ temperature was reproduced by submerging the
bags in tanks filled with constantly circulating surface seawater. To simulate light
intensity at the SCM the incident light was attenuated by covering the tanks with mesh
screens.
**2.3 Chlorophyll-*a***
Chlorophyll-*a* (Chl-*a*) concentration was measured at t0 and after 72 h incubation as a
phytoplankton biomass proxy. 300 ml of water samples were filtered through 0.2 μm
polycarbonate filters and frozen at -20ºC until further analysis. Chl-*a* was extracted with
90 % acetone and kept in darkness at 4ºC overnight. Fluorescence was determined with a
TD-700 Turner Designs fluorometer calibrated with pure Chl-*a* standard solution
(absorption coefficient at 663 nm = 87.7; Lorenzen and Newton Downs, 1986).
**2.4 Flow cytometry**
Samples for prokaryote abundance quantification (2 ml) were preserved with 1 %
paraformaldehyde + 0.05 % glutaraldehyde (final concentrations). Samples were
incubated 20 min for the fixative to act on cells, immersed in liquid nitrogen for 15 min,
and frozen at -80ºC. Abundance of prokaryotes was determined using a FACSCalibur
flow cytometer equipped with a laser emitting at 488nm. Samples were stained with
SYBR Green DNA fluorochrome, and bacterial abundance was detected by their
signature of side scatter (SSC) and green fluorescence as described by Gasol and Del
Giorgio (2000). The empirical calibration between light side scatter (SSC) and cell
diameter described by Calvo-Díaz and Moran (2006) were used to estimate cell
biovolume (BV). BV was converted into biomass by using the allometric factor of
Norland (1993: fg C cell$^{-1}$ = 120 × BV$^{0.72}$) for the coastal experiments and using the open
ocean conversion factor for the oceanic experiments (fg C cell$^{-1}$ = 350 × BV).
**2.5 Nutrients**
Aliquots for inorganic nutrient determinations (ammonium, nitrite, nitrate, phosphate,
and silicate) were collected before all other variables and directly from the Niskin bottle
in order to avoid contamination. Polyethylene bottles (50 ml) precleaned with 5 % HCl
were filled with the sample using contamination-free plastic gloves and immediately
frozen at -20°C until analysis using standard colorimetric methods with a Bran-Luebbe
segmented flow analyzer (Hansen and Grasshoff 1983). The detection limit was 0.1 μmol
l$^{-1}$ for nitrate, 0.02 μmol l$^{-1}$ for nitrite and phosphate and 0.05 μmol l$^{-1}$ for ammonium
and silicate. Dissolved inorganic nitrogen (DIN) concentration was calculated as the sum
of the ammonium, nitrite and nitrate concentrations.
**2.6 Vitamin B12**
Seawater samples for dissolved vitamin analysis were taken at surface and SCM depth on
day 1, day 3 and day 5 in the coastal, and on day 1, day 3 and day 6 oceanic station of
each cruise (Table S1 in the Supplement). Samples were filtered through 0.2 μm sterivex
filters and frozen at -20ºC until further analysis. Samples (1 l) were preconcentrated using
a solid-phase extraction with a C18 resin (Bondesil C18, Agilent) at pH 6.5 and rate of
1ml/min. Elution was performed with 12 ml of methanol (MeOH) LCMS grade that was
removed via evaporation with nitrogen in a Turbovap. Gas pressure was initially set at 5
PSI and was slowly increased to 15 PSI until 300-500 μl of sample remained. The
concentrated samples were frozen at -20ºC until further analysis using liquid
chromatography coupled to mass spectrometry system.
The concentrate was filtered again through a cellular acetate membrane 0.2 μm
(Phenomenex) prior to the analysis. Ultra Performance Liquid Chromatography tandem
Mass Spectometry 3Q (UPLC-MS/MS) methodology was adapted from Sañudo-
Wilhelmy et al. (2012), Heal et al. (2014) and Suffridge et al. (2017). Detection and
quantification of dissolved vitamin B12 (cyanocobalamin and hydroxocobalamin) was
conducted using an Agilent 1290 Infinity LC system (Agilent Technologies, Waghaeusel-
Wiesental, Germany), coupled to an Agilent G6460A triple quadrupole mass
spectrometer equipped with an Agilent Jet Stream ESI source. The LC system used a C18
reversed-phase column (Agilent Zorbax SB-C18 Rapid Resolution HT (2.1 inner
diameter × 50 mm length, 1.8 μm particle size) with a 100 μl sample loop. Agilent
Technologies software was used for data acquisition and analysis. Chromatographic
separation was performed using MeOH and water LCMS grade, both buffered to pH 5
with 0.5 % acetic acid, as mobile phases in a 15 minutes' gradient. Gradient starting at 7
% MeOH for 2 min, changing to 100 % MeOH by minute 11, continuing at 100 % MeOH
until 13.5 min and returning to initial conditions to complete 15 min. Limits of detection
(LODs) and limits of quantification (LOQs) were determined using sequential dilutions
of the lowest point of the calibration curves. LODs were defined as the lowest detectable
concentration of the analyte with a signal-to-noise (S/N) ratio for the qualitative transition
of at least 3. In the same way, LOQs were defined as the lowest quantificable
concentration with a S/N ratio of 10 for the quantitative transition. S/N ratios were
calculated using the Mass Hunter Workstation software B.04.01. The LODs obtained
were 0.04 pmol l$^{-1}$ for hydroxocobalamin (OHB12) and 0.01 pmol l$^{-1}$ for cyanocobalamin
(CNB12), while the LOQs values were 0.05 and 0.025 pmol l$^{-1}$ for OHB12 and CNB12,
respectively. The average B12 recovery percentage after pre-concentration and extraction
of B-vitamin spiked samples was 93%. B-vitamin free seawater was spiked with CNB12
and OHB12 standards for recovery percentage analysis. We failed to detect B1 vitamin
in the pre-concentrated samples, likely due to a low ambient concentration and low pre-
concentration volume.

**2.7 Microbial plankton community**

DNA samples were taken during the experimental period at surface and SCM depth in
the coastal and oceanic station. In particular, sampling of the microbial plankton
community was carried out on day 0, day 1, day 3 and day 5 of each cruise. Community
composition was assessed by sequencing the V4 and V5 regions from 16S rRNA gene
(16S rDNA) for prokaryotes and the V4 region from 18S rRNA gene (18S rDNA) for
eukaryotes. Two liters of water samples were sequentially filtered through 3 μm pore size
polycarbonate filters and 0.2 μm pore size sterivex filter and immediately frozen in liquid
nitrogen and conserved at -80 ºC. DNA retained in the 3 μm and 0.2 μm filters was
extracted by using the PowerSoil DNA isolation kit (MoBio Laboratories Inc., CA, USA)
and the PowerWater DNA isolation kit (MoBio Laboratories Inc., CA, USA),
respectively, according to the manufacturer's instructions. Prokaryotic DNA from 0.2 μm
filters was amplified using the universal primers "515F and 926R" (Parada et al., 2016)
and eukaryotic DNA from both, 3 μm and 0.2 μm filters, using the primers
"TAReuk454FWD1" and "TAReukREV3" (Logares et al., 2014). Amplified regions
were sequenced in an Illumina MiSeq platform and the sequences obtained were analyzed
with software package DADA2 (Callahan et al., 2016). SILVA reference database (Quast
et al., 2012) was used to taxonomic assignment of 16S amplicon sequence variants
(ASVs) and PR2 (Guillou et al., 2013) and the marine protist database from the BioMarks
project (Massana et al., 2015) were used to taxonomic assignment of 18S ASVs. The data
for this study have been deposited in the European Nucleotide Archive (ENA) at EMBL-
EBI (https://www.ebi.ac.uk/ena) under accession numbers PRJEB36188 (16S rDNA

sequences) and PRJEB36099 (18S rDNA sequences). ASV table is an analogue of the traditional OTU table which records the number of times each exact amplicon sequence variant was observed in each sample (Callahan et al., 2016).

The raw ASV tables of prokaryotes and eukaryotes were subsampled to the number of reads present in the sample with the lowest number of reads, which was 2080 and 1286, for 16S rDNA and 18S rDNA, respectively. The abundance of ASVs was averaged for coastal and oceanic samples, differentiating surface and SCM. A total of 1550 unique ASVs of prokaryotes were identified. As many ASVs of eukaryotes were present in both size fractions (e.g. those having a cell size range including 3 μm), we combined datasets derived from the 0.2 and the 3 μm filters for eukaryotic community analyses. As explained in Hernández-Ruiz et al. (2018), we normalized the reads from each filter size by the filter DNA yield, as recommended in Dupont et al. (2015), obtaining 2293 unique ASVs. The sequence abundances of the subsampled ASV tables were transformed using the centered log ratio (clr) (Fernandes et al., 2014; Gloor et al., 2017). Before clr transformation, zeros were replaced by the minimum value that is larger than 0 divided by 2 (Aitchison, 1982; Martín-Fernández et al., 2003).

**2.8 Statistical analysis**

To compare the effect of different nutrient additions on the response variables, chlorophyll-*a* concentration and prokaryote biomass, we calculated response ratios (RR) by dividing each observation (mean of triplicates) of each treatment by the respective control treatment mean. A value equal to 1 implies no response, a value < 1 implies a negative response and a value > 1 implies growth stimulation after nutrient addition. Secondary limitation by B vitamins was calculated by dividing the mean value in the inorganic nutrients and B vitamin combined treatment by the mean value in the inorganic

nutrient addition treatment. In the same way, a value < 1 implies a negative effect of B
vitamins and a value > 1 implies stimulation positive effect of B vitamin treatment
through secondary limitation.
Normal distribution was tested by a Kolmogorov-Smirnov test and non-normal variables
such as temperature, salinity, DIN, $SiO_4^{2-}$, and Chl-*a* and PB response ratios, were log
transformed to attain normality. All statistical analysis were considered significant at the
0.05 significance level and p-value was standardized as proposed by Good (1982) in order
to overcome the low number of replicates. Differences between station and depth (spatial
variability) and among sampling months (temporal variability) in the responses to B
vitamins were evaluated with factorial analysis of variance (ANOVA). Bonferroni post
hoc tests analyses were conducted to test which treatments were significantly different
from the control treatment in each experiment. Non-metric multidimensional scaling
(nMDS) was used to analyze the similarities between the samples based on microbial
assemblage structure using the PRIMER6 software (Clarke and Warwick, 2001; Clarke
and Gorley, 2006). The similarities were evidenced in a multidimensional space by
plotting more similar samples closer together. Analysis of similarity (ANOSIM) was used
to verify that microbial community composition from the same season and station were
more similar to each other than to communities from a different season and station. Z-test
was used to test if averaged B vitamins response ratios were significantly different from
1. The RELATE analysis implemented in PRIMER6 was used to relate the B-vitamin
response patterns (Bray-Curtis resemblance matrix built from phytoplankton and bacteria
response ratios) with: (1) environmental factors (Euclidean resemblance matrix built from
normalized values of ammonium, nitrite, nitrate, phosphate, silicate, B12, temperature,
salinity, Chl-*a* and prokaryote biomass), (2) prokaryote community composition
(Euclidean resemblance matrix built form clr-transformed sequence abundance of major
taxonomic groups), or (3) eukaryote community composition (Euclidean resemblance
matrix built form clr-transformed sequence abundance of major taxonomic groups).
RELATE calculates the Spearman rank correlations (Rho) between two resemblance
matrices, and the significance is tested by a permutation test (999 permutations). In order
to highlight which specific taxonomic groups are associated to changes of microbial
plankton (prokaryote plankton and phytoplankton) responses to vitamin B1 and B12, we
conducted a distance based redundancy analysis (dbRDA) combined with a distance
linear-based model (DistLM) using a step-wise procedure and adjusted $R^2$ as selection
criteria using the PRIMER6 software.

**3 Results**
**3.1 Initial conditions**
Different hydrographic conditions were found during each cruise (Fig. 1 and Fig. 2). In
February, heavy rainfall (Fig. 1c) combined with relaxed winds caused a halocline at 10
m depth (Fig. 2m). High levels of Chl-*a* (as derived from the calibrated CTD fluorescence
sensor) were observed at the coastal station, being maximum (4.97 µg l$^{-1}$) by the end of
the cruise (Fig. 2a). At the oceanic station, Chl-*a* levels remained low (less than 3 µg l$^{-1}$)
throughout the cruise, being slightly higher in the subsurface layer (Fig. 2d).
Strong precipitation during the April cruise (Fig. 1c) caused a persistent surface halocline
at the coastal station (Fig. 2n). Maximum Chl-*a* concentrations ranged from 0.99 to 2.73
µg l$^{-1}$, declining from day 5 onwards (Fig. 2b), coinciding with an increase in water
temperature associated to a downwelling situation. At the oceanic station, a persistent
subsurface Chl-*a* maximum (up to 1.61 µg l$^{-1}$) was observed throughout the cruise (Fig.
2e).

In August, strong thermal stratification was observed at both stations (Fig. 2i and Fig. 2l). At the beginning of the cruise, high Chl-$a$ concentration (close to 20 μg l$^{-1}$) was observed in subsurface water (Fig. 2c). Chl-$a$ was relatively low at the oceanic station, and increased by the end of the sampling period (Fig. 2f) as a consequence of an upwelling event (Fig. 1b), that brought cold and nutrient rich water to the surface, at day 5.

Abiotic and biotic conditions at the beginning of each experiment are shown in Fig. 3 and in the supplementary Table S2. Overall, the concentration of dissolved inorganic nitrogen (DIN) was higher at the coastal than at the oceanic station, where very low levels were measured in August (Fig. 3i). At the coastal station, higher DIN concentrations were observed in surface compared to subsurface waters. The DIN:DIP (dissolved inorganic phosphorous) ratio was always lower in open ocean than in the coastal station and mostly below the Redfield ratio (16:1). Phosphorous limitation (DIN:DIP > 16) was frequent in coastal surface waters in February and April (Fig. 3j and Fig. 3k).

On average, chl-$a$ concentration varied greatly between stations and months but was always higher at the coastal than at the oceanic station (Fig. 3a-c). Prokaryote biomass (PB) increased from winter (February) to summer (August) at the two stations (Fig. 3d-f). In February, Chl-$a$ concentrations increased by the end of the cruise at both coastal and oceanic stations (Fig. 3a), while PB remained very low throughout this sampling period (Fig. 3d). In April, both PB and Chl-$a$ were similar in the ocean and the coast, and showed reduced temporal variability (Fig. 3b and Fig. 3e), irrespective of the observed nutrient variability (Fig. 3h). In August, Chl-$a$ concentration was much higher at the coastal than at the oceanic station, and showed reduced temporal variability (except at the SCM in the coast) (Fig. 3c). At the beginning of the sampling period, PB was higher in the ocean than in the coast, and tended to decline by the end of the cruise (Fig. 3f).

A MDS analysis revealed that microbial community composition showed a relatively reduced variability within period, with samples clustering according to the sampling period (ANOSIM, p = 0.001) and station (ANOSIM, p = 0.001) (Fig. S1 in the Supplement). Consequently, we averaged the microbial community composition for each period and sampling site. The sampling period-averaged composition of the eukaryote community showed a clear variability among cruises, while differences between sampling locations and depths were less pronounced (Fig. 4a). At the coastal location, Mamiellophyceae (*Ostreococcus* and *Micromonas*) were relatively abundant in February and April, but their relative abundance sharply decreased in August. By contrast, the relative abundance of Dinophyceae was highest in August at both sampling locations. The contribution of diatoms (Bacillariophyta) was very low in summer at the oceanic station and marine alveolates (MALV) groups (MALV-I and MALV-II) were most representative in February at both locations. Flavobacteriales and Rhodobacterales were the dominant prokaryotes (Fig. 4b) in coastal waters, particularly in August, when both represented more than 80 % of sequences, while the Cyanobacteria *Synechococcus* were mostly present in February and April. In oceanic waters, Flavobacteriales and *Synechococcus* were the dominant prokaryotes. SAR11 clade and Archaea (Euryarchaeota and Thaumarchaeota) were most abundant in February at both sampling locations.

B12 concentration was low, ranging from 0.06 to 0.66 pmol $l^{-1}$ (Table S1 in the Supplement). Average B12 concentration was significantly higher in the coast (0.30±0.13 pmol $l^{-1}$) than in the ocean (0.15±0.12 pmol $l^{-1}$) (t-test, t = 3.17, df = 10, p = 0.01), and showed less variability at the coastal than at the oceanic station (Fig. 4c).

**3.2 Short-term phytoplankton and prokaryote responses to inorganic nutrients and vitamin additions**

The temporal development of the phytoplankton (as estimated from changes in Chl-*a*
concentration) and prokaryote biomass in the control treatments showed different
patterns. Chl-*a* remained either stable or increased after 72 h of incubation in 87.5% of
the experiments conducted in February and April. However, Chl-*a* mostly decreased in
the coastal experiments conducted in August (Fig. 5a and Fig. 5c). A very similar pattern
was observed for prokaryote biomass, although the decrease in biomass occurred both in
the coastal and in the oceanic stations during summer (Fig. 6).
The response ratios (RRs) of Chl-*a* and prokaryote biomass were calculated as a measure
of the magnitude of phytoplankton and prokaryote responses to nutrient and vitamin
treatments (Fig S2, S3 and S4 in the supplement). The RRs differed between sampling
stations (ANOVA, F (1,502) = 18.059, p < 0.001) and among sampling periods (ANOVA,
F (2,501) = 6.54, p = 0.002). The most prominent responses of phytoplankton, compared
to the control treatment, occurred after inorganic nutrient amendments, especially in
surface oceanic waters (Fig. 5c and Fig. S2b, f and j in the Supplement). The magnitude
of the phytoplankton response to inorganic nutrients was significantly higher in oceanic
than in coastal waters (ANOVA, F (1,34) = 5.22, p = 0.028). Prokaryotes responded less
than phytoplankton to inorganic nutrients and, in addition, heterotrophic prokaryote
responses to inorganic nutrients were similar between coastal and oceanic waters
(ANOVA, F (1,34) = 1.68, p = 0.203). The addition of inorganic nutrients caused
significant increases in Chl-*a* in 31 out of the 36 experiments (Fig. 5 and Fig S2 in the
supplement), while prokaryotes increased their biomass in 19 out of 36 experiments (Fig.
6 and Fig. S2 in the Supplement).
The addition of B12 stimulated phytoplankton in 5 out of 36 experiments (Fig. 5 and Fig.
S3 in the Supplement) and prokaryotes in 6 experiments (Fig. 6 and Fig. S4 in the
Supplement). Chl-a increased in 3, and prokaryote biomass in 7 out of 36 experiments
after adding B1 (Fig. 5 and Fig. 6). B vitamins also caused negative responses of
phytoplankton (Fig. 5 and Fig. S3 in the Supplement) and prokaryote biomass (Fig. 6 and
Fig. S4 in the Supplement). The addition of vitamins induced decreases of Chl-*a* in 6
experiments (4 after adding B12 and 2 after adding B1) and prokaryote biomass in 14
experiments (6 after adding B12 and 8 after adding B1). Secondary limitation by B1
and/or B12 was occasionally observed when inorganic nutrients were limiting, leading to
a higher biomass increase in the treatments including both inorganic nutrients and
vitamins as compared to the inorganic nutrient addition alone (Fig. 5, Fig. 6 and Fig. S3
and Fig. S4 in the Supplement). In the case of Chl-*a*, secondary limitation by B-vitamins
was found in the C-b-surface, Oc-a-SCM and Oc-b-SCM experiments in February, in the
C-b-surface and C-b-SCM experiments in April, and in the C-b-SCM, Oc-b-SCM and
Oc-c-surface experiments in August (Fig. 5).
In order to quantify the relevance of inter-day variability, we calculated the mean
coefficient of variation (CV) of the responses to B vitamins (i.e., excluding the responses
to inorganic nutrients, and normalizing the responses of the nutrient and vitamin
combined treatments to the corresponding response to inorganic nutrients alone) within
sampling periods for each sampling point (2 stations and 2 depths). The CV ranged from
9%, in subsurface oceanic waters in April, to 34% in surface coastal waters in April,
averaging 16±6 (SD) % (data not shown). Considering that short-term (within sampling
period) variability was overall very low, and for simplicity, we averaged the responses to
B vitamins in the 3 experiments conducted at each of the 12 sampling points to further
describe spatial and temporal patterns in the response to B vitamin amendments (Fig. 7).
When averaging the responses within each sampling point (Fig. 7), some general patterns
emerge. Both phytoplankton and prokaryotes showed more negative than positive
responses to B1 and/or B12 amendments. Most positive responses occurred at the oceanic
station (83.3%), while negative responses dominated in the coast (61.5%). Phytoplankton
significant positive responses mostly occurred in February, showing an average increase
of up to 1.2-fold in coastal subsurface waters after B12+B1 amendment (Fig. 7a). The
largest significant increase in Chl-a (ca. 1.4-fold) occurred in April after the combined
addition of B12 and B1 in coastal surface waters. Significant positive prokaryote
responses mainly occurred in August, when the largest increase (ca. 1.3-fold) occurred in
coastal subsurface waters after B1 amendment (Fig. 7b). Most positive responses were
associated with treatments containing B12 either alone or combined with B1 (Fig. 7b).
Phytoplankton primary B1 limitation was only found at the oceanic SCM in February
(Fig. 7a), while prokaryote primary B1 limitation only occurred at the coastal SCM in
August. In addition, prokaryote secondary B1 limitation occurred in oceanic surface
waters in February and August.
**3.3 B-vitamin response patterns in relation to environmental factors and prokaryote**
**and eukaryote community composition**
In order to explore the controlling factors of the observed B-vitamin response patterns,
the correlation between the B-vitamin response resemblance matrix and the
corresponding resemblance matrices obtained from the initial environmental factors, the
initial prokaryotic community composition, or the initial eukaryotic community
composition were calculated. While eukaryotic community composition did not show a
significant correlation with the B-vitamin responses (Spearman Rho = 0.05, p = 0.39), the
prokaryotic community composition was significantly correlated with the B-vitamin
responses (Spearman Rho = 0.31, p = 0.041). We then used distance-based linear
modelling (DistLM) to identify the prokaryotic taxa which best explained the microbial
plankton responses to B-vitamins (Fig. 8). The resulting model explained 78% of the
variation and included seven prokaryotic groups: *Planktomarina* (24%), Actinobacteria
(14%), SAR11 (8.2%), Cellvibrionales (8.5%), Euryarchaeota (8.7%), Flavobacteriales
(9%) and *Synechococcus* (6.1%). The sequential test identified *Planktomarina* and
Actinobacteria as the taxa explaining the largest fraction of variation (ca. 24 % and 14%,
respectively, data not shown). The total variation explained by the db-RDA1 (34.9%) and
db-RDA2 (24.5%) was 59.4 %, both represented as x and y axis, respectively (Fig. 8).
The db-RDA1 axis separated, to some extent, coastal samples, where negative responses
to B vitamins dominated, from oceanic samples, where most positive responses were
found (Fig. 7). The db-RDA plot showed that Cellvibrionales and *Planktomarina*
positively correlated with axis 1, while SAR11 and *Synechococcus* showed negative
correlation with axis 1. Flavobacteriales and Actinobacteria mostly correlated with the
db-RDA2 axis.

## 4 Discussion

Although the dependence of phytoplankton on B vitamin has been previously observed
in cultures (e.g. Croft et al., 2006; Droop, 2007; Tang et al., 2010) and in natural microbial
assemblages in coastal areas (e.g. Sañudo-Wilhelmy et al., 2006; Gobler et al., 2007;
Koch et al., 2011, 2012; Barber-Lluch et al., 2019), this is, to the best of our knowledge,
the most complete study about responses of phytoplankton and prokaryotes to vitamin
B12 and/or B1 addition. The 36 experiments developed in this study contributed to
increase our understanding of the role of vitamins B12 and B1 at different spatial and
temporal scales.
Considering the high short-time variability of the hydrographic conditions in the area
(Alvarez-Salgado et al., 1996), we expected a large inter-day variation in the responses
to B vitamin amendments. By contrast, inter-day variability of microbial responses to B
vitamins and microbial plankton community composition was relatively small (Fig. 5,
Fig. 6, Fig. S1 and Fig. S2 in the supplement). The reduced short-term variability in the
responses to B vitamins additions suggested that B vitamin availability might be
controlled by factors operating at larger temporal scales, such as the succession of
microbial communities associated to seasonal environmental variation (Hernández-Ruiz
et al., 2018; Hernando-Morales et al., 2018). Considering this, and for further discussion,
we averaged the responses from the three experiments conducted during each sampling
period, resulting in 12 experimental situations (2 stations × 2 depths × 3 periods). Overall,
phytoplankton and/or prokaryote growth enhancement in at least one B vitamin treatment
was frequent but relatively small in this productive ecosystem, showing 1.1 to 1.3-fold
increases in 75% of the experimental situations for phytoplankton and in 50% for
prokaryotes. On the other hand, negative responses to at least one B vitamin treatment
occurred in 83% of the experimental situations for phytoplankton and in 67% for
prokaryotes (Fig. 7). The low and constant B12 ambient concentration (Fig. 4c) and the
reduced magnitude of microbial responses suggest a close balance between production
and consumption of this growth factor. Different patterns of response to B-vitamin
amendments were observed in phytoplankton and prokaryotes (Fig. 7), which appear to
be mostly explained by the prokaryotic community composition (Fig. 8).
**4.1 Positive responses to vitamin B1 and B12 amendments**
The experimental design allowed the detection of two categories of B vitamin dependency
of the microbial plankton community. A primary limitation by B vitamins occurs when
microorganisms respond to additions of B vitamins alone. A secondary limitation by B
vitamins arises when the response to the combined addition of B vitamins and inorganic
nutrients is significantly higher than that to inorganic nutrients alone. Such response
occurs because of the ambient B-vitamin depletion associated to the plankton growth after
inorganic nutrient enrichment. Most positive (72% for phytoplankton and 60 % for
prokaryotes) responses occurred after single B-vitamins additions, suggesting that
inorganic nutrient availability enhance B-vitamin production by the prototrophic
microbes. Under nutrient-limiting conditions, the external supply of vitamins could
reduce the energy costs associated to its synthesis (Jaehme and Slotboom, 2015),
stimulating the growth not only of auxotrophs but also of prototrophs.
The significant positive effects of B12 and/or B1 addition, suggest that these compounds
may be eventually limiting microbial growth in marine productive ecosystems, as
previously observed by other authors (e.g., Panzeca et al., 2006; Sañudo-Wilhelmy et al.,
2006; Bertrand et al., 2007; Gobler et al., 2007; Koch et al., 2011; 2012; Barber.-Lluch et
al., 2019). Most positive responses to B vitamin amendments were observed in oceanic
waters, where B12 concentration was significantly lower than in coastal waters (Fig. 4c).
Unfortunately we lack B1 measurements in this study, but, according to previous field
studies in other oceanographic regions, a similar pattern to that observed for B12 can be
expected (Cohen et al., 2017; Sañudo-Wilhelmy et al., 2012; Suffridge et al., 2018). The
overall low and stable concentration of B12 at both sampling locations suggests a high
turnover time of this compound in these productive, well-lit waters. Rapid cycling of B12
in surface waters may occur due to high biological uptake rates (Taylor and Sullivan,
2008; Koch et al., 2012) and/or photochemical degradation (Carlucci et al., 1969;
Juzeniene and Nizauskaite, 2013; Juzeniene et al., 2015). The measured B12
concentrations were in the lower range reported for coastal sites, and similar to that found
in the upwelling system off the California coast in the San Pedro Basin during winter,
spring and summer (Panzeca et al., 2009).
The increase of Chl-*a* was mostly associated to B12 amendments, which is consistent
with the known incapability of eukaryotes to synthesize this vitamin (Croft et al., 2005;
Tang et al., 2010; Sañudo-Wilhelmy et al., 2014). Considering the very low concentration
of B12 in the sampling area, the relatively limited phytoplankton response to B vitamins
suggests that the existing species might have adapted to overcome B12 shortage. For
example, changes in external B12 availability may cause shifts from vitamin B12-
dependence to vitamin B12-independence in taxa possessing the vitamin B12-
independent methionine synthase (MetE) gene (Bertrand et al., 2013; Helliwell et al.,
2014). Other strategies used by phytoplankton to cope with low B12 concentration
include, increased cobalamin acquisition machinery, decreased cobalamin demand, and
management of reduced methionine synthase activity through changes in folate and S-
adenosyl methionine metabolism (Bertrand et al., 2012). The available data on B12 half-
saturation constants for phytoplankton (0.1–10 pmol $l^{-1}$) (Droop, 1968, 2007; Taylor and
Sullivan, 2008; Tang et al., 2010; Koch et al., 2011) are similar or higher than the B12
concentrations measured here (0.3 pmol $l^{-1}$ in the coastal and 0.15 pmol $l^{-1}$ in the oceanic
waters, on average), reinforcing the hypothesis of a phytoplankton community adapted to
B12 limiting concentrations in this upwelling system.
The positive responses of phytoplankton in surface oceanic waters in February seemed to
be associated with high abundance of *Synechococcus* and SAR11 (Fig. 4b and Fig. 8).
*Synechococcus* produce a B12 analog known as pseudocobalamin, where the lower ligand
base adenine replaces 5,6-dimethylbenzimidazole (DMB) (Helliwell et al., 2016). In
natural conditions, pseudocobalamin is considerably less bioavailable to eukaryotic algae
than other cobalamin forms (Helliwell et al., 2016; Heal et al., 2017). SAR11 do not
require B12 and do not have pathways for its synthesis (Sañudo-Wilhelmy et al., 2014;
Gómez-Consarnau et al., 2018), suggesting that B12 synthesis could be limited in oceanic
waters in winter, due to the low abundance of potential B12 producers.
Microbial responses to B vitamins in subsurface oceanic waters in February were
associated to high abundance of *Synechococcus* and, to some extent, of Actinobacteria
(Fig. 8). In these experiments, positive effects of B1 addition on phytoplankton and
prokaryotes were observed (Fig. 7). While *Synechococcus* is capable of B1 synthesis
(Carini et al., 2014; Sañudo-Wilhelmy et al., 2014; Gómez-Consarnau et al., 2018),
Actinobacteria seems to have a strong dependence on this vitamin (Gómez-Consarnau et
al., 2018). Among the sequenced eukaryote genomes, only Stramenopiles contain genes
codifying for the synthesis of thiamine monophosphate (Sañudo-Wilhelmy et al., 2014;
Cohen et al., 2017). While Stramenopiles, dominated by Bacillariophyta, were ubiquitous
in the sampling area, their relative contribution was lower in oceanic waters (Fig. 4a).
The simultaneous stimulation of phytoplankton and prokaryotes by B1 addition in
subsurface oceanic waters in winter suggest a strong demand for this compound under
these particular conditions, however what triggers the observed responses remain unclear.
Even though B1 caused a significant effect on phytoplankton only in subsurface waters
in winter, half of the positive responses of prokaryotes were associated to B1 supply (Fig.
7b). This pattern is consistent with the recently described widespread dependence of
bacterioplankton on external B1 supply (Paerl et al., 2018). B1 stimulated prokaryote
growth in subsurface coastal waters and surface oceanic waters in summer (Fig. 7b), when
the B vitamin response patterns were associated to high abundance of *Planktomarina* and
Actinobacteria (Fig. 8),which are expected to strongly depend on external B1 sources
(Giebel et al., 2013; Gómez-Consarnau et al., 2018). The generalized significant and
positive responses of prokaryotes to vitamin treatments in surface oceanic waters in
summer, when the prokaryote biomass was high and dissolved inorganic nitrogen
concentration was very low (Fig. 3i), suggest that prokaryotes may have an advantage in
the uptake and assimilation of B vitamins under nitrogen limiting conditions. This is
consistent with the observation of small (0.7–3 μm)-plankton cells containing more B1
than larger cells (Fridolfsson et al., 2019). Following this, it has been speculated that
bacteria and small phytoplankton can transfer B1 to large cells through predation by
acting as an important source of this compound in the marine environment (Fridolfsson
et al., 2019).
**4.2 Negative responses to vitamin B1 and B12 amendments**
Similar experiments conducted in this area also reported negative responses of microbial
plankton to vitamin B12 additions (Barber-Lluch et al., 2019). The predominantly
negative prokaryote responses after vitamin amendments in the coast during summer (Fig.
7b), when nutrient concentrations were low (Fig. 3), suggest either a strong competition
between phytoplankton and prokaryotes or a stimulation of predation. Dinoflagellates
were particularly abundant in summer at both sampling sites and depths. Many
dinoflagellate species are auxotrophs for B1 and/or B12 (Croft et al, 2006; Tang et al.,
2010), and also many of them are phagotrophs (Stoecker and Capuzzo, 1990; Smayda,
1997; Sarjeant and Taylor, 2006; Stoecker et al., 2017), thus the external supply of B
vitamins may have promoted their growth, ultimately leading to net decreases in
microbial biomass at the end of the experiments. Several studies demonstrated that
vitamin B12 is implicated in the occurrence of dinoflagellate blooms around the world
(Aldrich, 1962; Carlucci and Bowes, 1970; Takahashi and Fukazawa, 1982; Yu and
Rong-cheng, 2000). It has been suggested that the B12-dependent enzyme
methylmalonyl-CoA mutase in dinoflagellate, euglenoid, and heterokont algae allows
them to grow heterotrophically when B12 is available (Croft et al., 2006). Therefore, the
B12 enrichment could trigger such nutritional strategy, particularly in summer, when
mineral nutrients are less available, resulting in an increased predation pressure on
prokaryotes.

The B vitamin response patterns in surface coastal waters in summer (Fig. 7), seemed to be associated with high abundance of Flavobacteriales (Fig. 8). All isolates of Bacteroidetes sequenced so far are predicted to be B12 auxotrophs (Sañudo-Wilhelmy et al., 2014; Gómez-Consarnau et al., 2018) and recent metatranscriptomic analyses reveal that B1 synthesis gene transcripts are relatively low in Flavobacteria as a group (Gómez-Consarnau et al., 2018). As both phytoplankton and prokaryotes are dominated by potentially B12 and B1 auxotrophs (dinoflagellates and Flavobacteriales) in the coast during summer (Fig. 4b), the negative responses could be the result of strong competition for B vitamins. However, the negative responses to B vitamins of both phytoplankton and prokaryotes in surface coastal water in summer suggests an increase in phytoplankton and prokaryote predation by mixotrophs rather than competition between them. By contrast, prokaryotes and phytoplankton showed opposite patterns of response to B vitamins in subsurface coastal waters in summer, which suggests competition between both microbial compartments (Fig. 7). While phytoplankton negatively responded only to single B vitamin additions, prokaryotes responded negatively only when both inorganic nutrients and B vitamins were added (Fig. 7). It is conceivable that phytoplankton had an advantage over prokaryotes when mineral nutrients were added. This hypothesis contrasts with previous studies reporting that B12 and B1 vitamin uptake is dominated by picoplankton (Koch et al., 2011, 2012), strongly suggesting that bacteria could outcompete larger phytoplankton for vitamin uptake. By contrast, Koch et al. (2014), found that carbon-specific B12 uptake by large phytoplankton was significantly lower during non-bloom (low nutrient concentration) compared to bloom conditions (high nutrient concentration), which suggests better competitive ability under nutrient-rich conditions.

**5 Conclusions**

In conclusion, our findings suggest that the heterogeneous responses of microbial plankton to B1 and B12 vitamins supply in this coastal upwelling system could be partially controlled by the composition of the prokaryote community, which is consistent with their previously reported major role as B12 producers and B1 consumers. Even though we lack data on B1 concentration, the overall moderate responses together with the low ambient B12 concentration, suggest that the microbial plankton community in this area could be well adapted to cope with B vitamin shortage and that a close balance exists between production and consumption of these important growth factors.

*Author contribution.*

Eva Teira designed the experiments and Vanessa Joglar carried them out with contributions from all co-authors. Vanessa Joglar analyzed the data, Vanessa and Eva Teira interpreted the results and Vanessa Joglar prepared the manuscript under Eva Teira supervision.

*Competing interests.* The authors declare that they have no conflict of interest.

*Acknowledgements*

We thank all the people involved in the project ENVISION for helping with sampling and analytical work. We also thank the crew of the Ramón Margalef for their help during the work at sea. From IIM-CSIC, V. Vieitez and M.J. Pazó performed the nutrient analyses. This research was supported by the Spanish Ministry of Economy and Competitiveness through ENVISION (CTM2014-59031-P) and INTERES (CTM2017-83362-R) projects. Vanessa Joglar was supported by a FPI fellowship from the Spanish Ministry of Economy and Competitiveness.

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

Table 1


| | Treatment | Nutrient included | Concentration |
|---|---|---|---|
| 1. | Control (C) | No nutrient added | |
| 2. | Inoganic nutrients (I) | $NO_3^-$ | 5 µmol l$^{-1}$ |
| | | $NH_4^+$ | 5 µmol l$^{-1}$ |
| | | $HPO_4^{2-}$ | 1 µmol l$^{-1}$ |
| | | $SiO_4^{2-}$ | 5 µmol l$^{-1}$ |
| 3. | Vitamin B12 (B12) | B12 | 100 pmol l$^{-1}$ |
| 4. | Vitamin B1 (B1) | B1 | 600 pmol l$^{-1}$ |
| 5. | B12 + B1 | B12 | 100 pmol l$^{-1}$ |
| | | B1 | 600 pmol l$^{-1}$ |
| 6. | I + B12 | $NO_3^-$ | 5 µmol l$^{-1}$ |
| | | $NH_4^+$ | 5 µmol l$^{-1}$ |
| | | $HPO_4^{2-}$ | 1 µmol l$^{-1}$ |
| | | $SiO_4^{2-}$ | 5 µmol l$^{-1}$ |
| | | B12 | 100 pmol l$^{-1}$ |
| 7. | I + B1 | $NO_3^-$ | 5 µmol l$^{-1}$ |
| | | $NH_4^+$ | 5 µmol l$^{-1}$ |
| | | $HPO_4^{2-}$ | 1 µmol l$^{-1}$ |
| | | $SiO_4^{2-}$ | 5 µmol l$^{-1}$ |
| | | B1 | 600 pmol l$^{-1}$ |
| 8. | I + B12 + B1 | $NO_3^-$ | 5 µmol l$^{-1}$ |
| | | $NH_4^+$ | 5 µmol l$^{-1}$ |
| | | $HPO_4^{2-}$ | 1 µmol l$^{-1}$ |
| | | $SiO_4^{2-}$ | 5 µmol l$^{-1}$ |
| | | B12 | 100 pmol l$^{-1}$ |
| | | B1 | 600 pmol l$^{-1}$ |



**6 Tables and Figures**

**Table 1:** Eight different treatments were applied consisting of: (1) control treatment (C): no nutrients added; (2) inorganic (I) nutrient treatment: 5 µM nitrate ($NO_3^-$), 5 µM ammonium ($NH_4^+$), 5 µM silicate ($SiO_4^{2-}$) and 1 µM phosphate ($HPO_4^{2-}$); (3) vitamin B12 treatment: 100 pmol l$^{-1}$; (4) vitamin B1 treatment: 600 pmol l$^{-1}$); (5) inorganic nutrients and vitamin B12 (I+B12) treatment; (6) Inorganic nutrients and vitamin B1 (I+B1) treatment; (7) vitamins B12 and B1 (B12+B1) treatment and (8) Inorganic nutrients with vitamins B12 and B1 (I+B12+B1) treatment.

**Figure 1:** (a) The NW Iberian margin (rectangle) and locations of the stations that were sampled in the Ría de Vigo (C) and on the shelf (Oc) (diamonds), (b) distribution of daily coastal upwelling index (UI) and (c) registered precipitations during each sampling period showing the initial time of each experiment (C-a, C-b, C-c and Oc-a, Oc-b, Oc-c).

**Figure 2:** Vertical distribution over time in the coastal station of Chl-*a* (µg l$^{-1}$) in (a) February, (b) April and (c) August; temperature (ºC) in (g) February, (h) April and (i) August; and salinity (PSU) in (m) February, (n) April and (o) August. Vertical distribution over time in the oceanic station of Chl-*a* (µg l$^{-1}$) in (d) February, (e) April and (f) August; temperature (ºC) in (j) February, (k) April and (l) August; and salinity (PSU) in (p) February, (q) April and (r) August Dots show the t0 of the experiments. Chl-*a*: Chlorophyll-*a* concentration.

**Figure 3:** Initial biological conditions and abiotic factors at the coastal and oceanic sampling stations. Each bar corresponds to one of the 3 experiments performed in each depth and station during February, April and August. (a, b, c), Chl-*a*, total Chl-*a* (µg l$^{-1}$).

Note that the y-axis is broken; (d, e, f) PB, prokaryote biomass ($\mu$g C l$^{-1}$); (g, h, i) DIN,
dissolved inorganic nitrogen ($\mu$mol l$^{-1}$) and (j, k, l) DIN:DIP, ratio inorganic
nitrogen:phosphate. The blue line shows the Redfield ratio (16:1) and SCM refers to the
sub-surface chlorophyll maximum. Chl-*a*: Chlorophyll-*a* concentration.

**Figure 4:** Averaged relative contribution of reads to the major taxonomic groups of (a)
eukaryotes and (b) prokaryotes at surface (surf) and SCM in the coastal and oceanic
station in February, April and August. (c) Averaged B12 concentration (pmol l$^{-1}$) at
surface (surf) and SCM in the coastal and oceanic station in February, April and August.
Error bars represent standard error. SCM refers to the sub-surface chlorophyll maximum.

**Figure 5**: Chlorophyll-*a* concentration ($\mu$g l$^{-1}$) in the t0 of each experiment (striped bars)
and in the endpoint of each treatment (colored bars) in the experiments conducted at (a)
surface and (b) SCM in the coastal and at (c) surface and (d) SCM in the oceanic station
in February, April and August. Error bars represent standard error. Note that the y-axis is
broken. SCM: sub-surface chlorophyll maximum.

**Figure 6**: Prokaryote biomass ($\mu$g C l$^{-1}$) in the t0 of each experiment (striped bars) and in
the endpoint of each treatment (colored bars) in the experiments conducted at (a) surface
and (b) SCM in the coastal and at (c) surface and (d) SCM in the oceanic station in
February, April and August. Error bars represent standard error. Note that the y-axis is
broken. SCM: sub-surface chlorophyll maximum.

**Figure 7:** Monthly averaged response ratio (RR) of (a) Chl-*a* or (b) prokaryote biomass
at surface and SCM in the coastal and oceanic station. Horizontal line represents a
response equal to 1, that means no change relative to control in the pink dots (treatments
with vitamins alone) and no change relative to inorganic (I) treatment in the green dots
(vitamins combined with I treatments). Asterisks indicate averaged RRs that were
significantly different from 1 (Z-test; * $p < 0.05$) and "a" symbols indicate averaged RRs
that were marginally significant (Z-test; [a] $p = 0.05$-$0.06$). Error bars represent standard error.
SCM: sub-surface chlorophyll maximum.

**Figure 8:** Distance based redundancy analysis (dbRDA) of B vitamin responses by
phytoplankton and prokaryotes based on Bray-Curtis similarity. Only prokaryotic taxa
that explained variability in the B vitamin responses structure selected in the DistLM
model (step-wise procedure with adjusted $R^2$ criterion) were fitted to the ordination.
Filled and open symbols represent samples from coastal and oceanic station, respectively,
triangles and circles represent samples from surface and SCM, respectively, and colours
correspond to the months: (green) February, (blue) April and (pink) August. SCM: sub-
surface chlorophyll maximum.

Fig. 01

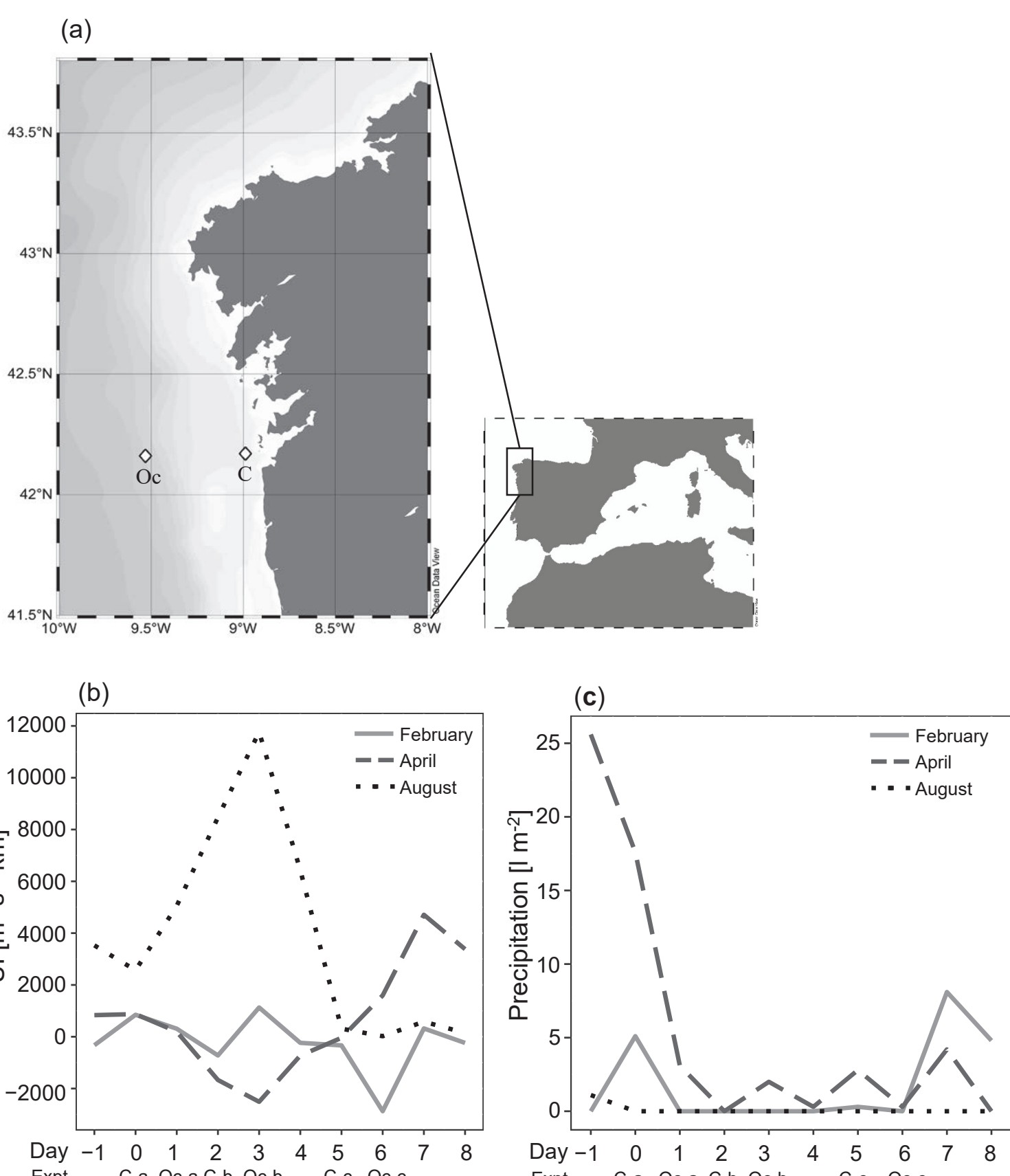

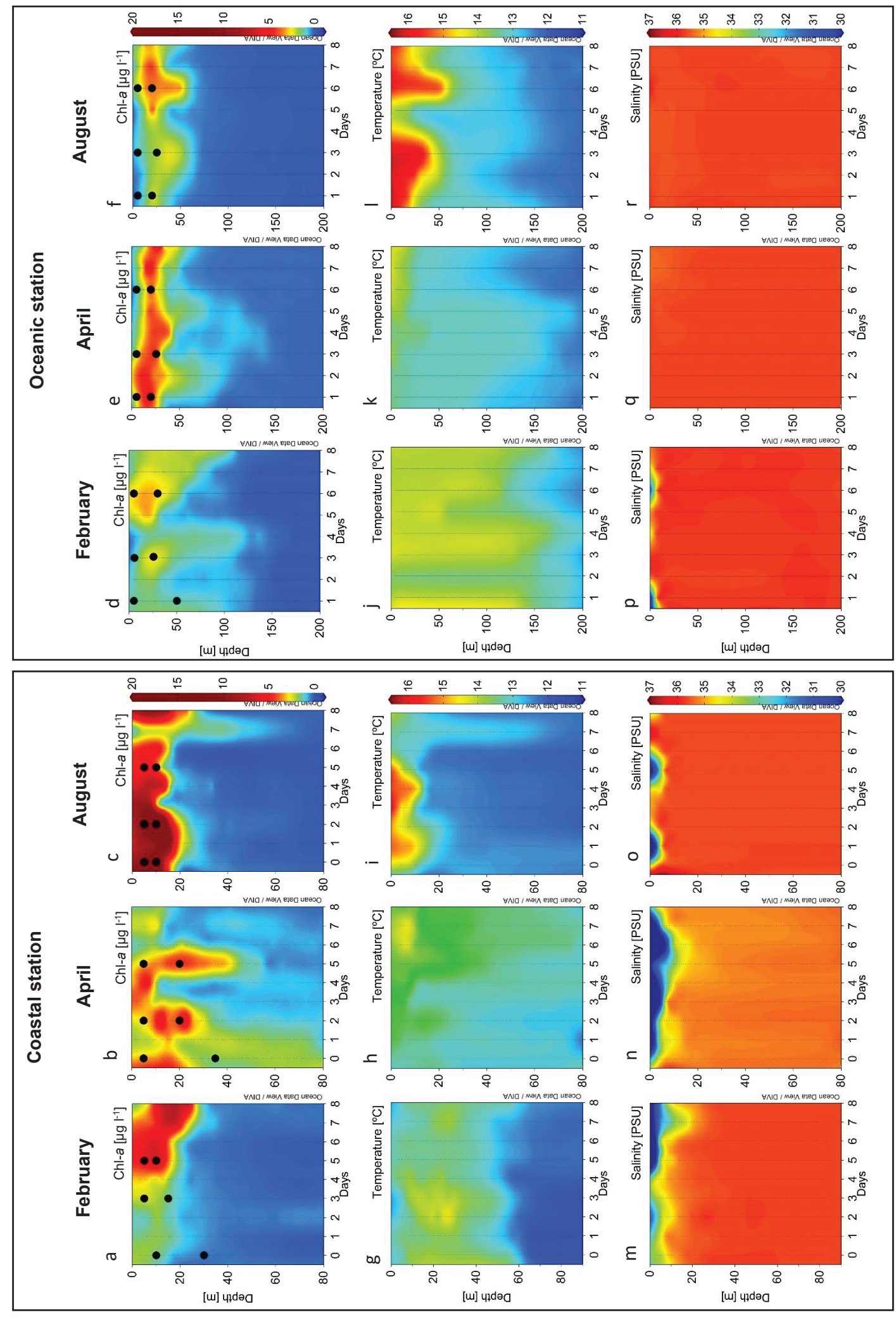

Fig. 03

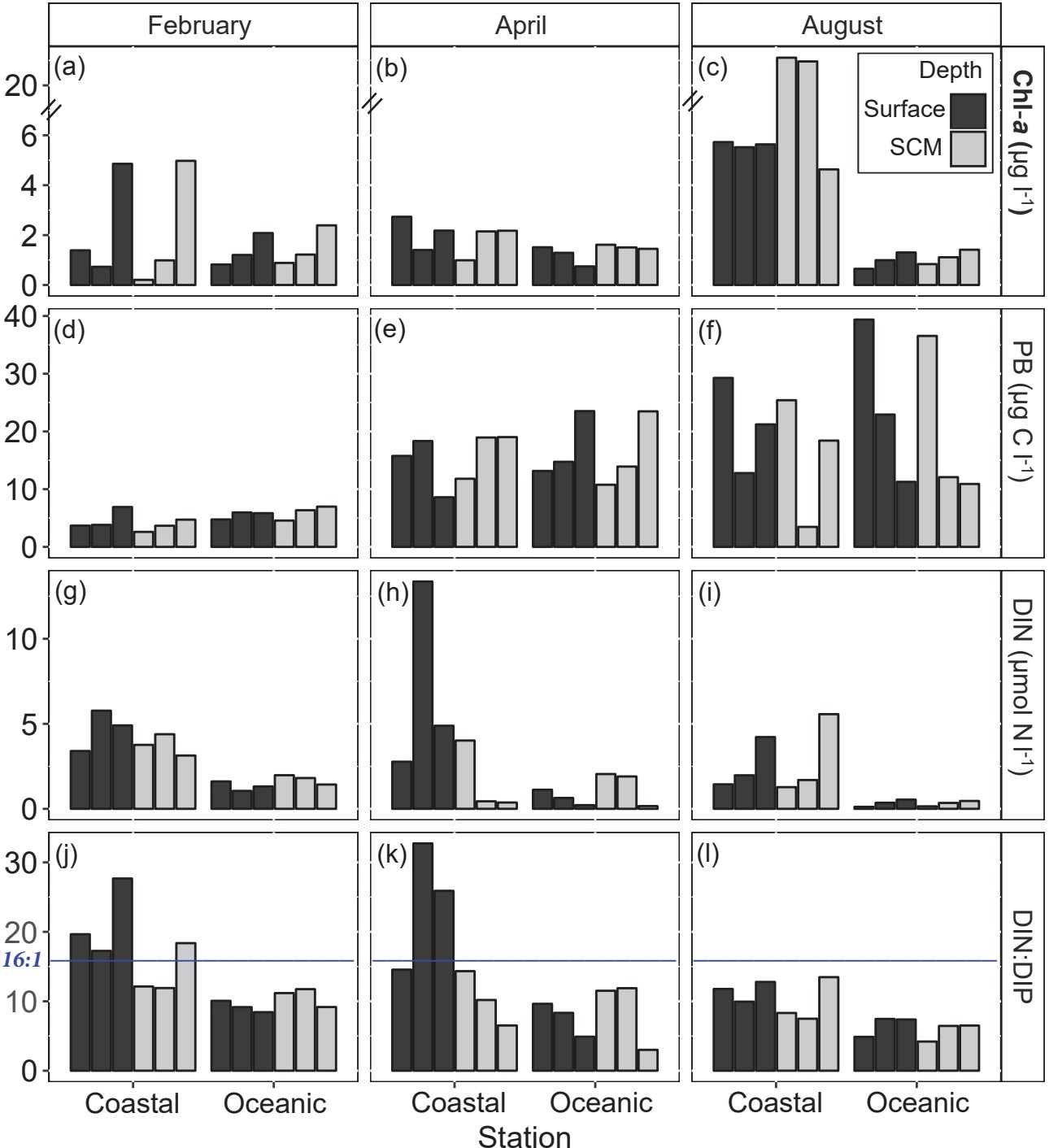

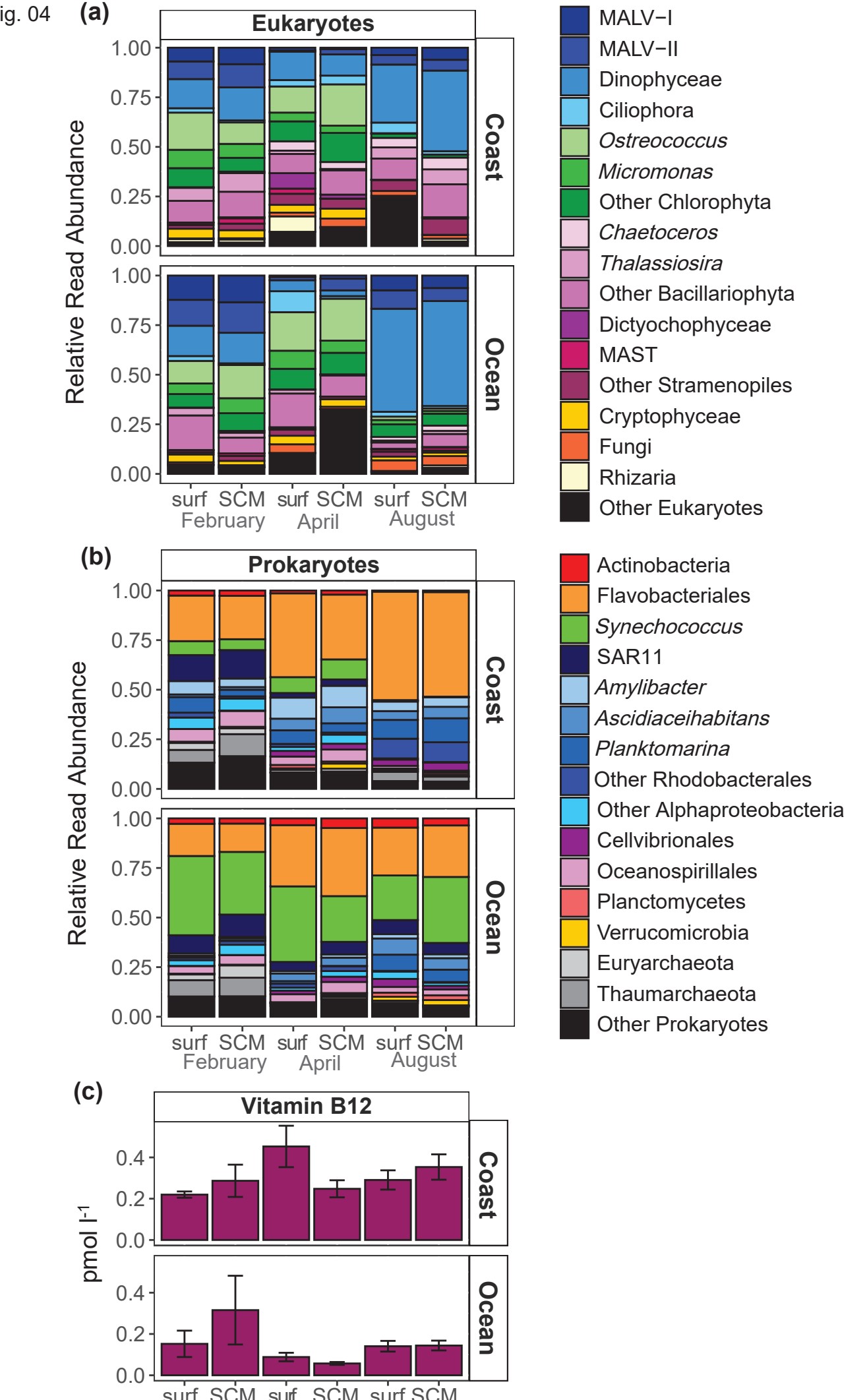

Fig. 04

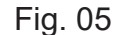
Fig. 05

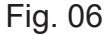

**(a)** Surface Coastal station

**(b)** SCM Coastal station

T0
C
I
B12
B1
B12+B1
I+B12
I+B1
I+B12+B1

a --------February--------- b    c    a -----------April----------- b    c    a ---------August--------- b    c

**(c)** Surface Oceanic station

T0
C
I
B12
B1
B12+B1
I+B12
I+B1
I+B12+B1

**(d)** SCM Oceanic station

a ----------February--------- b    c    a -----------April------------ b    c    a ---------August--------- b    c

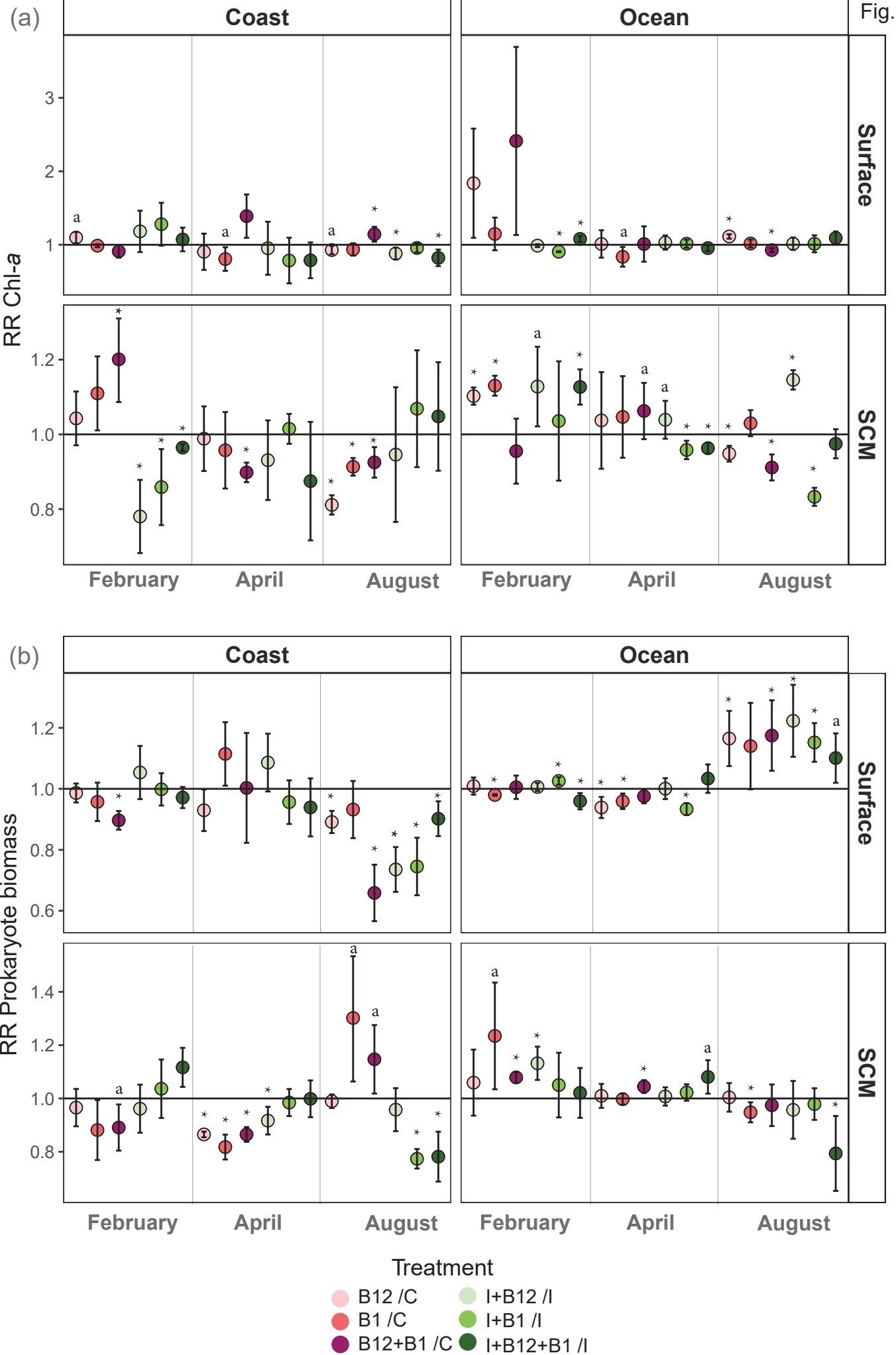

Fig. 07

Fig. 08

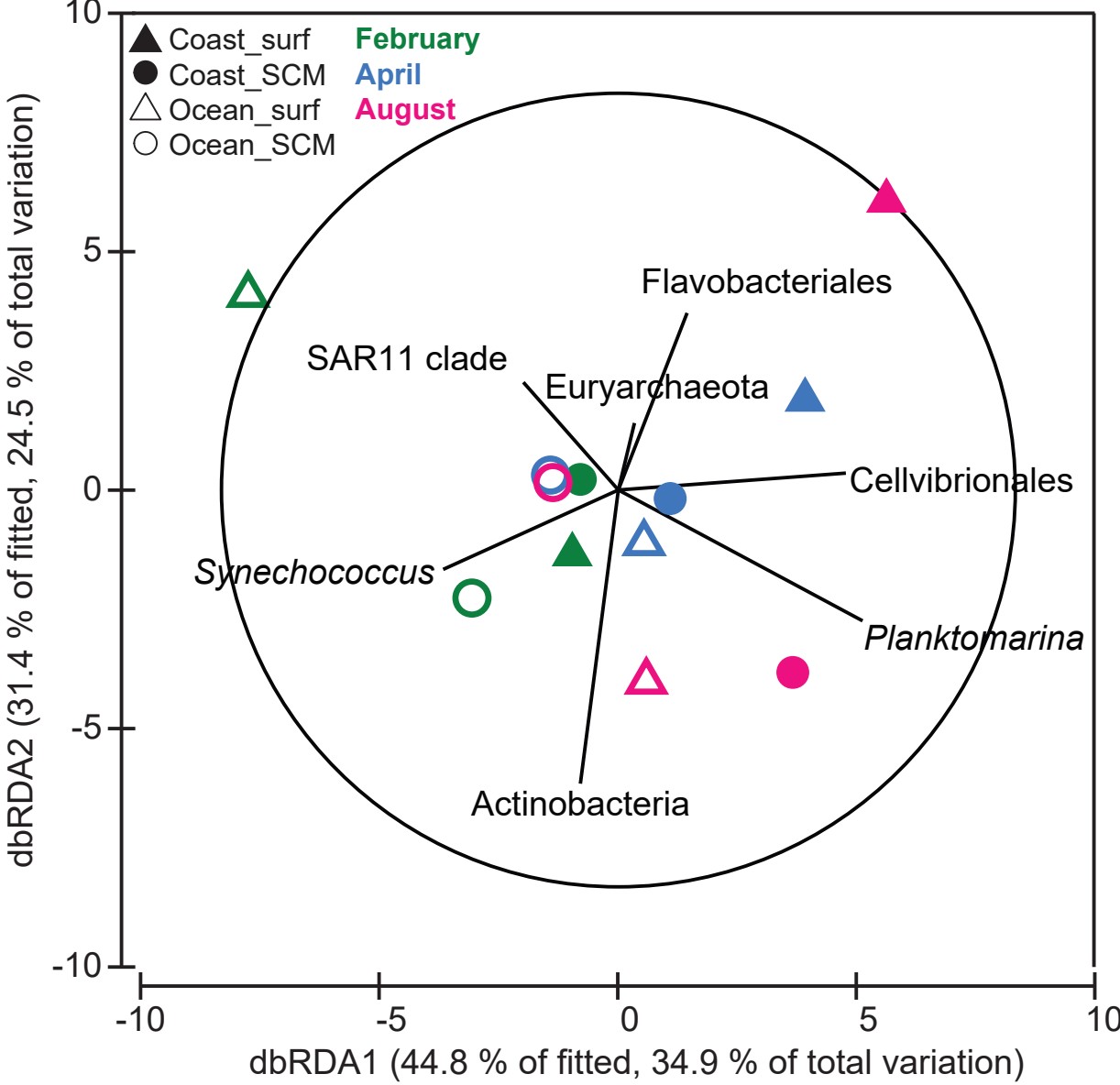