# Peer review of "Spatial and temporal variability in the response of"

_Biogeosciences, 2019_

## Referee Comment (RC1) · Anonymous Referee #1 · 4 Sep 2019

The question asked by Joglar and co-authors is an important one ('how does vitamin B1/12 availability influence coastal and oceanic microbial communities?), which they have addressed using a large number of detailed bioassay experiments. The research question fits within the scope of Biogeosciences and I suggest that ultimately the results should be published in this journal. However in its current form the manuscript suffers a bit from a lack of clarity and succinct conclusions, making it hard to understand what the take-home messages of this work are. Given the very large amount of work this study has involved, this is a shame. Below I make some recommendations for improvement. The manuscript would also benefit from checking by a native English speaker.

[Figure]

My main initial request is to include figures for the actual bacterial and phytoplankton biomass changes in the experiments, rather than simply ratios, including the values for the initial conditions. I believe this should be in the main manuscript, not just the Supporting Information. These data can be displayed as a mean with error bars representing the spread across the three treatment replicates. I believe this will give a better indication of how the community responded in the experiments. The ratio figures can be included too for discussion/interpretation purposes. Please also label the treatments below each bar in each case – I found treatment identification a little difficult in the current figures.

Secondly I think the manuscript should also note how trace metal contamination could have biased the results. This is currently not discussed at all, but could have had an important influence. For instance, if contaminating iron had been inadvertently included in the treatments. Contamination would likely originate from the metal CTD-rosette, the rosette bottles, during bottle sampling, from the incubation bags, from the nutrient additions etc. Where certain procedures were carried out to reduce this, these should be described. This is significant, as this microbes in this region could be experiencing primary iron limitation – see Blain et al. (2004).

Blain, S., Guieu, C., Claustre, H., Leblanc, K., Moutin, T., Quéguiner, B., Ras, J. and Sarthou, G., 2004. Availability of iron and major nutrients for phytoplankton in the northeast Atlantic Ocean. Limnology and Oceanography, 49(6), pp.2095-2104.

Specific comments

In the abstract I would recommend making reference to the study region (i.e. 'North east Atlantic', or 'off the northwest coast of Spain')

Figure 1b and c: please indicate when experiments were sampled for (i.e. which day? day 0?)

Line 15–16: rephrase 'was not of great concern'

I would recommend noting the microbial responses to major nutrient supply, in addition to B12/B1, in the abstract.

I would recommend stating the number of the 36 experiments where bacteria/phytoplankton responded positively/negatively to vitamin supply in the abstract.

Line 21 'Growth stimulation by B1 addition was more frequent on bacteria' – relative to phytoplankton?

Lines 35–36 and elsewhere: I would recommend seeing the more recent studies of Browning et al., 2017 and Browning et al., 2018, which also perform trace-metal-clean B12 addition bioassay experiments in upwelling/coastal/offshore regions.

Browning, T.J., Achterberg, E.P., Rapp, I., Engel, A., Bertrand, E.M., Tagliabue, A. and Moore, C.M., 2017. Nutrient co-limitation at the boundary of an oceanic gyre. Nature, 551(7679), p.242.

Browning, T.J., Rapp, I., Schlosser, C., Gledhill, M., Achterberg, E.P., Bracher, A. and Le Moigne, F.A., 2018. Influence of iron, cobalt, and vitamin B12 supply on phytoplankton growth in the tropical East Pacific during the 2015 El Niño. Geophysical Research Letters, 45(12), pp.6150-6159.

Line 39: synthesized by prokaryotes and archaea?

Line 42: Have not defined 'cobalamin' (In general I recommend choosing B12 or cobalamin and sticking to it throughout)

Line 79: Perhaps mention here succinctly what Gobler et al. (2007) found?

Line 79: the reference Barber-Lluch et al. (2019) does not appear in the reference list

Lines 114–115: How was this water sampled? From the regular stainless CTD? If so, trace element contamination should be acknowledged. Also see general comment.

Line 125: Was there any treatment of the whirl-pak bags (e.g. acid and deionized water rinses) to remove contamination? Also see general comment.

Line 127 and on: What were the chemical stocks of the nutrients (e.g. brand and purity). Again, if these nutrients were not pre-treated to remove trace element contamination, this should be acknowledged. Also see general comment.

Line 137: Were the tanks screened, or open to the air?

Line 147: Was any time given for the fixative to act on cells before flash freezing in liquid nitrogen?

Section 2.5: If known, what was the recovery percent of the B12 pre-concentration/extraction? (i.e. via use of a standard)

Line 271: How was the upwelling index calculated (cannot see this in methods)

Figure 5: It is not clear that the value being displayed is the RR Chla OR RR BB and not the ratio of these.

As the Figure 5 has signs indicating statistical significance, the error in the spread across the treatment replicates must have been prorogated somehow? Can this error be included as error bars in the figure?

337–338: Specifically which experiments showed serial limitation by B vitamins?

Line 402: 'clarify the paper of vitamins'?

Lines 417–419: Please distinguish between the phytoplankton/bacteria responses in this value of 75%

Line 425: No full stop (perhaps also rephrase to 'community assemblage'?)

Lines 491–495: This doesn't quite make sense – in the first sentence it states that phytoplankton responses to B1 supply were restricted, and in the second the stimulation of phytoplankton is discussed.

I would advise including a table summarizing initial conditions (i.e., nutrient concentrations, temperature, chlorophyll-a, initial bacteria and so on)

---

## Author Comment (AC1) · 24 Oct 2019

We very much appreciate the useful and constructive comments made by the reviewer, which surely contribute to improve the manuscript quality. We have considered all the suggestions and made the requested modifications, as detailed below.

My main initial request is to include figures for the actual bacterial and phytoplankton biomass changes in the experiments, rather than simply ratios, including the values for the initial conditions. I believe this should be in the main manuscript, not just the Supporting Information. These data can be displayed as a mean with error bars representing the spread across the three treatment replicates. I believe this will give a

[Figure]

better indication of how the community responded in the experiments. The ratio figures can be included too for discussion/interpretation purposes. Please also label the treatments below each bar in each case – I found treatment identification a little difficult in the current figures.

We have now included in the main manuscript the requested figure (new figure 5). We present the response of phytoplankton and that of bacteria separately. The former figure 5 with the response ratios is now included as supplementary information.

Secondly I think the manuscript should also note how trace metal contamination could have biased the results. This is currently not discussed at all, but could have had an important influence. For instance, if contaminating iron had been inadvertently included in the treatments. Contamination would likely originate from the metal CTD-rosette, the rosette bottles, during bottle sampling, from the incubation bags, from the nutrient additions etc. Where certain procedures were carried out to reduce this, these should be described. This is significant, as this microbes in this region could be experiencing primary iron limitation – see Blain et al. (2004). Blain, S., Guieu, C., Claustre, H., Leblanc, K., Moutin, T., Quéguiner, B., Ras, J. and Sarthou, G., 2004. Availability of iron and major nutrients for phytoplankton in the northeast Atlantic Ocean. Limnology and Oceanography, 49(6), pp.2095-2104.

We did not use trace metal clean techniques for sampling. We used standard stainless CTD-rosette and Niskin metal-free bottles. We did not have a trace metal clean lab on board, so, even though samples were carefully manipulated contamination by trace metals could have eventually occurred. It is important to note that the water for the experiments was pooled into a 20 l acid-cleaned carboy before filling the bags, thus all the bags would have the same incidental input of trace metals. We are aware that microbes could be limited by other trace elements or nutrients not considered in our treatments, such as iron or other B vitamins. For this reason, we based the discussion on the response ratios at the end of the experiments.

Specific comments

In the abstract I would recommend making reference to the study region (i.e. 'North east Atlantic', or 'off the northwest coast of Spain')

The study area has been specified in the abstract (L15-L16)

Figure 1b and c: please indicate when experiments were sampled for (i.e. which day? day 0?)

Sampling day for each experiment has been indicated in the graphs (Fig. 1b and Fig. 1c) Line 15–16: rephrase 'was not of great concern'

This has been rewritten (L18) I would recommend noting the microbial responses to major nutrient supply, in addition to B12/B1, in the abstract.

The response to inorganic nutrient additions has been considered in the abstract (L16-L17)

I would recommend stating the number of the 36 experiments where bacteria/ phytoplankton responded positively/negatively to vitamin supply in the abstract.

This information has been included in the abstract (L19-L21)

Line 21 'Growth stimulation by B1 addition was more frequent on bacteria' – relative to phytoplankton?

This has been clarified (L25)

Lines 35–36 and elsewhere: I would recommend seeing the more recent studies of Browning et al., 2017 and Browning et al., 2018, which also perform trace-metal-clean B12 addition bioassay experiments in upwelling/coastal/offshore regions.

Browning, T.J., Achterberg, E.P., Rapp, I., Engel, A., Bertrand, E.M., Tagliabue, A. and Moore, C.M., 2017. Nutrient co-limitation at the boundary of an oceanic gyre. Nature, 551(7679), p.242. Browning, T.J., Rapp, I., Schlosser, C., Gledhill, M., Achterberg,

E.P., Bracher, A. and Le Moigne, F.A., 2018. Influence of iron, cobalt, and vitamin B12 supply on phytoplankton growth in the tropical East Pacific during the 2015 El Niño. Geophysical Research Letters, 45(12), pp.6150-6159.

Both studies have been cited in the revised version (L40)

Line 39: synthesized by prokaryotes and archaea?

Archaea is included within the prokaryote organisms. This has been clarified in the manuscript (L43)

Line 42: Have not defined 'cobalamin' (In general I recommend choosing B12 or cobalamin and sticking to it throughout)

This has been corrected (L47)

Line 79: Perhaps mention here succinctly what Gobler et al. (2007) found?

This has been explained in the revised text (L85-L86)

Line 79: the reference Barber-Lluch et al. (2019) does not appear in the reference list âËŸA.

This citation has been included in the reference list (L599-L601)

Lines 114–115: How was this water sampled? From the regular stainless CTD? If so, trace element contamination should be acknowledged. Also see general comment.

This has been acknowledged in the revised manuscript (L122; L127; L137-L138; L147-149)

Line 125: Was there any treatment of the whirl-pak bags (e.g. acid and deionized water rinses) to remove contamination? Also see general comment.

We used these bags mainly because they are sterile, non-toxic and transparent to the whole solar spectrum, thus avoiding UVR absorption of most other materials, and have been frequently used for experimentation with plankton communities (Gonzalez et al

1990, Davidson et al 2000, Pakulski et al 2007, Teixeira et al 2018). The bags were not additionally treated as were used only once (L138-L139)

Line 127 and on: What were the chemical stocks of the nutrients (e.g. brand and purity). Again, if these nutrients were not pre-treated to remove trace element contamination, this should be acknowledged. Also see general comment.

All the reagents were from Sigma of highest purity. Stocks were prepared with autoclaved Milli-Q water. No additional treatment for trace element contamination was applied.

Line 137: Were the tanks screened, or open to the air?

Tanks were screened to attenuate light intensity (L152-L154)

Line 147: Was any time given for the fixative to act on cells before flash freezing in liquid nitrogen?

Samples were incubated 20 min for fixative to act (L164-L165)

Section 2.5: If known, what was the recovery percent of the B12 preconcentration/ extraction? (i.e. via use of a standard)

Average B12 recovery percentage was 93% (L205-L208)

Line 271: How was the upwelling index calculated (cannot see this in methods).

Upwelling index was calculated by calculating the Ekman transport from surface winds at fix-station (st3) located at 42° N and 8.88° W. This information has been included in the revised text (L131-L135) Figure 5: It is not clear that the value being displayed is the RR Chla OR RR BB and not the ratio of these.

This has been clarified (Fig. S2 in the supplement)

As the Figure 5 has signs indicating statistical significance, the error in the spread across the treatment replicates must have been prorogated somehow? Can this error

be included as error bars in the figure?

The error bars representing the standard error of the three replicates have been included in the new figure 5

337–338: Specifically which experiments showed serial limitation by B vitamins?

This has been specified in the revised manuscript (L363-L366)

Line 402: 'clarify the paper of vitamins'?

This fragments has been corrected for clarity (L428)

Lines 417–419: Please distinguish between the phytoplankton/bacteria responses in this value of 75%.

Taking into account the responses of phytoplankton and bacteria separately, the percentages were 75% for phytoplankton and 50% for bacteria (L444)

Line 425: No full stop (perhaps also rephrase to 'community assemblage'?)

This has been rewritten (L451)

Lines 491–495: This doesn't quite make sense – in the first sentence it states that phytoplankton responses to B1 supply were restricted, and in the second the stimulation of phytoplankton is discussed.

Phytoplankton responses to B1 are overall restricted. The second sentence refers to the particular simultaneous stimulation of phytoplankton and bacteria by B1 addition found in subsurface oceanic waters in February (L520-L521)

I would advise including a table summarizing initial conditions (i.e., nutrient concentrations, temperature, chlorophyll-a, initial bacteria and so on).

We have added a supplementary table including detailed information about initial conditions (Table S2 in the supplement)

In addition to the modifications suggested by the reviewer, we have made the following change. The named OTUs (operation taxonomic units) has been replaced by ASV (amplicon sequence variant) due to the sequence analysis method used DADA2.

Please also note the supplement to this comment:
https://www.biogeosciences-discuss.net/bg-2019-306/bg-2019-306-AC1-supplement.zip
* * *
[Figure]

[Figure]

Figure 1

**Fig. 1.** The NW Iberian margin (rectangle) and locations of the stations, distribution of daily coastal upwelling index (Iw) and registered precipitations

[Figure]

**Fig. 2.** Vertical distribution in the coastal station of fluorescence ($\mu$g l-1), temperature ($^\circ$C) and salinity

[Figure]

Figure 3

**Fig. 3.** Initial biological conditions and abiotic factors at the coastal

[Figure]

[Figure]

Figure 4

**Fig. 4.** Averaged relative contribution of reads to the major taxonomic groups of eukaryotes and prokaryotes and B12 concentration

[Figure]

Figure 5

**Fig. 5.** Phytoplankton biomass (estimated as Chl-a concentration) ($\mu$g l-1) in the time-zero of each experiment and in the final-time of each treatment (colored bars)

[Figure]

Figure 6

**Fig. 6.** Bacterial biomass ($\mu$g C l-1) in the time-zero of each experiment (striped bars) and in the final-time of each treatment (colored bars)

[Figure]

Figure 7

**Fig. 7.** Monthly averaged response ratio (RR) of (a) total phytoplankton community and of (b) bacterial community at surface and SCM in the coastal and oceanic station.

[Figure]

Figure 8

**Fig. 8.** Distance based redundancy analysis (dbRDA) of B vitamin responses by microbial plankton based on Bray-Curtis similarity.

---

## Referee Comment (RC2) · Anonymous Referee #2 · 28 Oct 2019

The role that the availability of B-vitamins, specifically vitamin B12 and B1, play in shaping the marine microbial community is very relevant. The authors of this manuscript conducted an extensive experimental campaign with the goal of providing some insight to these processes. Unfortunately, their findings are poorly communicated and overstated in this manuscript. Most of the discussion is highly speculative and is insufficiently referenced. The authors have gone "all-in" on the poorly justified concept of "response ratio". I feel like this calculated metric is overly general and prevents an in-depth analysis of the actual data which likely contains subtle variations that could either support or undermine the authors primary conclusions. I don't understand why the authors chose to use response ratios rather more traditional ecological and physiological metrics. While response ratios could be a useful part of the discussion, they should be just that, a part of the discussion. Additionally, the authors ignore the rates of community growth and dynamics and only assess the response at the end time point relative to the initial point. While it is not possible at this point to change the experimental design, the authors need to change their interpretation of the data to acknowledge the limits of their data.

It is unfortunate that the only measures of biomass performed by these authors during their experiments were bacterial abundance and chlorophyll A. These are very broad, unspecific measures of community structure, that can be impacted by a myriad of environmental factors. The authors make some substantial claims about the roles that B-vitamin additions are playing on the microbial community; however, I wonder if they really have enough resolution in their measurements to make these claims. The author's use of "response rate" to obscures the fact that they are only measuring bacterial abundance and chlorophyll concentration. There are so many variables that impact these measures, it's not clear to me that the authors are actually looking at responses from B-vitamins.

I have some substantial concerns about the conclusions the authors make about community diversity and B-vitamins. Their exact statistical methods need to be better explained. Additionally, the authors need to fully explain the limits of their statistical methods, and not overstate or be overly speculative about the observed correlations between abiotic/biotic factors, B-vitamins, and the amplicon data.

The manuscript needs substantial copy editing/English language editing. All sections need to be streamlined. The interpretation of results tends to be far too speculative. The authors need to only make claims that their data can support.

The B12 analytical method appears to be derived from previously published methods. Specifically, those published by Heal et al. 2014, Sañudo et al. 2012, and Suffridge et al. 2017. It is troubling to me that the authors do not cite any of these papers in

the methods section, despite the fact that the described method is a nearly an exact match of those described in the above papers. Additionally, SPE extraction efficiency and limits of detection need to be included.

How were the whirl-pak bags prepared? Were they prepared to be trace clean? Were they sterile? What sort of plastic are they made out of? Trace metal or trace organic (B-vitamin) contamination is a real concern in experiments like these, especially when the authors want to make conclusions about the impact of a trace-component. Many plastics contain trace contamination from the factory, and if the bags were not properly prepared, this variability could interfere with all results.

---

## Author Comment (AC2) · 13 Nov 2019

**Referee1 comments**

AC/ We very much appreciate the useful and constructive comments made by the reviewer, which surely contribute to improve the manuscript quality. We have considered all the suggestions and made the requested modifications, as detailed below.

RC/ My main initial request is to include figures for the actual bacterial and phytoplankton biomass changes in the experiments, rather than simply ratios, including the values for the initial conditions. I believe this should be in the main manuscript, not just the

Supporting Information. These data can be displayed as a mean with error bars representing the spread across the three treatment replicates. I believe this will give a better indication of how the community responded in the experiments. The ratio figures can be included too for discussion/interpretation purposes. Please also label the treatments below each bar in each case – I found treatment identification a little difficult in the current figures.

AC/ We have now included in the main manuscript the requested figure (new figure 5 and 6). We present the response of phytoplankton and that of bacteria separately. The former figure 5 with the response ratios is now included as supplementary information.

RC/ Secondly I think the manuscript should also note how trace metal contamination could have biased the results. This is currently not discussed at all, but could have had an important influence. For instance, if contaminating iron had been inadvertently included in the treatments. Contamination would likely originate from the metal CTD-rosette, the rosette bottles, during bottle sampling, from the incubation bags, from the nutrient additions etc. Where certain procedures were carried out to reduce this, these should be described. This is significant, as this microbes in this region could be experiencing primary iron limitation – see Blain et al. (2004). Blain, S., Guieu, C., Claustre, H., Leblanc, K., Moutin, T., Quéguiner, B., Ras, J. and Sarthou, G., 2004. Availability of iron and major nutrients for phytoplankton in the northeast Atlantic Ocean. Limnology and Oceanography, 49(6), pp.2095-2104.

AC/ We did not use trace metal clean techniques for sampling. We used standard stainless CTD-rosette and Niskin metal-free bottles. We did not have a trace metal clean lab on board, so, even though samples were carefully manipulated contamination by trace metals could have eventually occurred. It is important to note that the water for the experiments was pooled into a 20 l acid-cleaned carboy before filling the bags, thus all the bags would have the same incidental input of trace metals. We are aware that microbes could be limited by other trace elements or nutrients not considered in our treatments, such as iron or other B vitamins. For this reason, we based the discussion

on the response ratios at the end of the experiments.

Specific comments

RC/ In the abstract I would recommend making reference to the study region (i.e. 'North east Atlantic', or 'off the northwest coast of Spain') AC/ The study area has been specified in the abstract (L15)

RC/ Figure 1b and c: please indicate when experiments were sampled for (i.e. which day? day 0?) AC/ Sampling day for each experiment has been indicated in the graphs (Fig. 1b and Fig. 1c)

RC/ I would recommend noting the microbial responses to major nutrient supply, in addition to B12/B1, in the abstract. AC/ The response to inorganic nutrient additions has been considered in the abstract (L16-L17)

RC/ Line 15–16: rephrase 'was not of great concern' AC/ This has been rewritten (L18)

RC/ I would recommend stating the number of the 36 experiments where bacteria/ phytoplankton responded positively/negatively to vitamin supply in the abstract. AC/ This information has been included in the abstract (L19-L21)

RC/ Line 21 'Growth stimulation by B1 addition was more frequent on bacteria' – relative to phytoplankton? AC/ This has been clarified (L25)

RC/ Lines 35–36 and elsewhere: I would recommend seeing the more recent studies of Browning et al., 2017 and Browning et al., 2018, which also perform trace-metal-clean B12 addition bioassay experiments in upwelling/coastal/offshore regions. Browning, T.J., Achterberg, E.P., Rapp, I., Engel, A., Bertrand, E.M., Tagliabue, A. and Moore, C.M., 2017. Nutrient co-limitation at the boundary of an oceanic gyre. Nature, 551(7679), p.242. Browning, T.J., Rapp, I., Schlosser, C., Gledhill, M., Achterberg, E.P., Bracher, A. and Le Moigne, F.A., 2018. Influence of iron, cobalt, and vitamin B12 supply on phytoplankton growth in the tropical East Pacific during the 2015 El Niño. Geophysical Research Letters, 45(12), pp.6150-6159. AC/ Both studies have been

cited in the revised version (L40)

RC/ Line 39: synthesized by prokaryotes and archaea? AC/ Archaea is included within the prokaryote organisms. This has been clarified in the manuscript (L43)

RC/ Line 42: Have not defined 'cobalamin' (In general I recommend choosing B12 or cobalamin and sticking to it throughout) AC/ This has been corrected (L47)

RC/ Line 79: Perhaps mention here succinctly what Gobler et al. (2007) found? AC/ This has been explained in the revised text (L85-L86)

RC/ Line 79: the reference Barber-Lluch et al. (2019) does not appear in the reference list âËŸA. AC/ This citation has been included in the reference list (L621-L623)

RC/ Lines 114–115: How was this water sampled? From the regular stainless CTD? If so, trace element contamination should be acknowledged. Also see general comment. AC/ This has been acknowledged in the revised manuscript (L122; L136-L138)

RC/ Line 125: Was there any treatment of the whirl-pak bags (e.g. acid and deionized water rinses) to remove contamination? Also see general comment. AC/ We used these bags mainly because they are sterile, non-toxic and transparent to the whole solar spectrum, thus avoiding UVR absorption of most other materials, and have been frequently used for experimentation with plankton communities (Gonzalez et al., 1990; Davidson and van der Heijden, 2000; Pakulski et al., 2007; Teixeira et al., 2018). The bags were not additionally treated as were used only once.

RC/ Line 127 and on: What were the chemical stocks of the nutrients (e.g. brand and purity). Again, if these nutrients were not pre-treated to remove trace element contamination, this should be acknowledged. Also see general comment. AC/ All the reagents were from Sigma of highest purity. Stocks were prepared with autoclaved Milli-Q water. No additional treatment for trace element contamination was applied.

RC/ Line 137: Were the tanks screened, or open to the air? AC/ Tanks were screened to attenuate light intensity (L151-L152)

RC/ Line 147: Was any time given for the fixative to act on cells before flash freezing in liquid nitrogen? AC/ Samples were incubated 20 min for fixative to act (L162-L163)

RC/ Section 2.5: If known, what was the recovery percent of the B12 preconcentration/ extraction? (i.e. via use of a standard) AC/ Average B12 recovery percentage was 93% (L215-L217)

RC/ Line 271: How was the upwelling index calculated (cannot see this in methods). AC/ Upwelling index was calculated by calculating the Ekman transport from surface winds at fix-station (st3) located at 42° N and 8.88° W. This information has been included in the revised text (L131-L134)

RC/ Figure 5: It is not clear that the value being displayed is the RR Chla OR RR BB and not the ratio of these. AC/ This has been clarified (Fig. S2 in the supplement)

RC/ As the Figure 5 has signs indicating statistical significance, the error in the spread across the treatment replicates must have been prorogated somehow? Can this error be included as error bars in the figure? AC/ The error bars representing the standard error of the three replicates have been included in the new figure 5 (now Figure 5 and 6).

RC/ 337–338: Specifically which experiments showed serial limitation by B vitamins? AC/ This has been specified in the revised manuscript (L385-L388)

RC/ Line 402: 'clarify the paper of vitamins'? AC/ This fragments has been corrected for clarity (L444)

RC/ Lines 417–419: Please distinguish between the phytoplankton/bacteria responses in this value of 75%. AC/ Taking into account the responses of phytoplankton and bacteria separately, the percentages were 75% for phytoplankton and 50% for bacteria (L457-L548)

RC/ Line 425: No full stop (perhaps also rephrase to 'community assemblage'?) AC/ This sentence was removed.

RC/ Lines 491–495: This doesn't quite make sense – in the first sentence it states that phytoplankton responses to B1 supply were restricted, and in the second the stimulation of phytoplankton is discussed. AC/ Phytoplankton responses to B1 are overall restricted. The second sentence refers to the particular simultaneous stimulation of phytoplankton and bacteria by B1 addition found in subsurface oceanic waters in February (L533-L534)

RC/ I would advise including a table summarizing initial conditions (i.e., nutrient concentrations, temperature, chlorophyll-a, initial bacteria and so on). AC/ We have added a supplementary table including detailed information about initial conditions (Table S2 in the supplement)

AC/ In addition to the modifications suggested by the reviewer, we have made the following change. The named OTUs (operation taxonomic units) has been replaced by ASV (amplicon sequence variant) due to the sequence analysis method used DADA2. We have eliminated the Pearson correlation between response ratios and the clr (centered-log-ratio) abundance of taxonomic groups (reported in former table 1) as was redundant with the dbRDA. Regarding the RELATE analysis to explore the relationship between the responses to B vitamin treatments (response ratios of phytoplankton and bacteria) and (1) the environmental variables (including nutrients, temperature, salinity, B12, chla and BB), (2) the prokaryotic, or (3) eukaryotic community structure, we believe that nicely shows that the responses are only significantly related to the prokaryotic community structure.

Please also note the supplement to this comment:
https://www.biogeosciences-discuss.net/bg-2019-306/bg-2019-306-AC2-supplement.zip

[Figure]

**Fig. 1.** (a) The NW Iberian margin and locations of the stations that were sampled, (b) distribution of daily coastal upwelling index (Iw) and (c) registered precipitations.

[Figure]

**Fig. 2.** Vertical distribution in the coastal and oceanic station of fluorescence ($\mu$g l-1), temperature ($^\circ$C) and salinity (PSU) over time for February, April and August.

[Figure]

**Fig. 3.** nitial biological conditions and abiotic factors at the coastal and oceanic sampling stations.

[Figure]

**Fig. 4.** (a) Averaged relative contribution of reads to the major taxonomic groups of eukaryotes and prokaryotes. (b) Averaged B12 concentration (pM)

[Figure]

**Fig. 5.** Phytoplankton biomass (estimated as Chl-a concentration) ($\mu$g l-1) in the time-zero of each experiment (striped bars) and in the final-time of each treatment (colored bars)

[Figure]

**Fig. 6.** Bacterial biomass ($\mu$gC l-1) in the time-zero of each experiment (striped bars) and in the final-time of each treatment (colored bars) in the experiments

[Figure]

[Figure]

**Fig. 7.** Monthly averaged response ratio (RR) of (a) total phytoplankton community and of (b) bacterial community at surface and SCM in the coastal and oceanic station.

[Figure]

**Fig. 8.** Distance based redundancy analysis (dbRDA) of B vitamin responses by microbial plankton based on Bray-Curtis similarity.

---

## Author Comment (AC3) · 13 Nov 2019

**Referee 2 comments**

AC/ We are very grateful for the reviewer's comments, which have contributed to improve the manuscript.

RC/ The role that the availability of B-vitamins,specifically vitamin B12 and B1,play in shaping the marine microbial community is very relevant. The authors of this manuscript conducted an extensive experimental campaign with the goal of providing some insight to these processes. Unfortunately, their findings are poorly communi-

cated and overstated in this manuscript. Most of the discussion is highly speculative and is insufficiently referenced.

AC/ We have made our best to refer adequately all the relevant studies and eliminating some speculative statements.

RC/ The authors have gone "all-in" on the poorly justified concept of "response ratio". I feel like this calculated metric is overly general and prevents an in-depth analysis of the actual data which likely contains subtle variations that could either support or undermine the authors primary conclusions. I don't understand why the authors chose to use response ratios rather more traditional ecological and physiological metrics. While response ratios could be a useful part of the discussion, they should be just that, a part of the discussion. Additionally, the authors ignore the rates of community growth and dynamics and only assess the response at the end time point relative to the initial point. While it is not possible at this point to change the experimental design, the authors need to change their interpretation of the data to acknowledge the limits of their data.

AC/ We are aware that sampling only at one endpoint (after 72 h incubation in our case) does not allow to discuss in detail the dynamics during each experiment, however we were particularly interested in extensively exploring the temporal and spatial variability of the response to vitamin enrichment. The experimental design, involved 36 experiments, with 8 triplicate treatments (24 experimental units per experiment). Even sampling only at the beginning and at the end we collected 972 samples for chlorophyll-a, and 972 for bacterial biomass. Initial and endpoint sampling is a common practice in enrichment microcosm experiments (e. g. Mills et al., 2004; Moore et al., 2006; Gobler et al., 2007; Bonnet et al., 2008; Koch et al., 2011), and allows the estimation of net growth rates using the following formula: ln (endpoint biomass/initial biomass)/incubation time. A previous work by Barber-Lluch et al (2019) in the same sampling area allowed us to conclude that sampling at 72 h was adequate to explore the effect of vitamins on both phytoplankton and bacterial biomass. As we agree

with the referee that the dynamics of phytoplankton and bacteria during the experiments are of interest, and following also the advice of referee 1, we now include in the manuscript two new figures where the initial and endpoint value of chlorophyll-a and bacterial biomass is represented. The response ratio figure is now included in the supplementary information. We accordingly now describe the dynamics of both planktonic components in the different experiments. We nevertheless decided to keep the response ratio as a measure of the magnitude of the effect (see below), which is very useful for the sake of comparison. We used here the response ratio as the quotient between the measured quantity of a response variable in experimental and control experimental units. Previous studies dealing with the effects of nutrients additions on microbial communities have noted the importance of expressing the change in the treatment relative to the control (Downing et al., 1999; Hedges et al., 1999; Elser et al., 2007, among others). We find that this variable is particularly adequate as a measure of the experimental effect because it quantifies the proportionate change that results from an experimental manipulation. This metric is widely use in marine ecology, and particularly in nutrient amendment experiments (e.g. Martínez-García et al., 2010; Teira et al., 2013; Barber-Lluch et al., 2019). The use of the response ratio calculated from endpoint biomass data provides the same information as the comparison of growth rates between treatments. Below we plot, as an example, the relationship between the response ratio from biomass (endpoint biomass in treatment divided by endpoint biomass in control) and the difference between growth rates (growth rate in treatment minus growth rate in the control) using data from two of our experiments (represented with different colours). It can be appreciated that the information provided by the response ratios follows exactly the same pattern as that provided by comparing growth rates. Moreover, the range of variation is higher for the response ratio, which allows to statistically detecting more subtle changes.

RC/ It is unfortunate that the only measures of biomass performed by these authors during their experiments were bacterial abundance and chlorophyll A. These are very broad, unspecific measures of community structure, that can be impacted by a myriad of environmental factors. The authors make some substantial claims about the roles that B-vitamin additions are playing on the microbial community; however, I wonder if they really have enough resolution in their measurements to make these claims. The author's use of "response rate" to obscures the fact that they are only measuring bacterial abundance and chlorophyll concentration. There are so many variables that impact these measures, it's not clear to me that the authors are actually looking at responses from B-vitamins.

AC/ B vitamins are essential growth factors for all microorganisms; therefore, the ultimate effect of a vitamin deficiency will be an impairment of growth, which is typically evaluated from changes in biomass. It is true that we only measure the effect on bulk phytoplankton and bacteria, and thus we have toned down all the conclusions about the effect on microbial community structure. We do not think that is unfortunate to have chosen phytoplankton and bacterial biomass as response variable, considering that most previous studies evaluating the role of B vitamins were based on biomass measurements (Sañudo-Wilhelmy et al., 2006; Gobler et al., 2007; Koch et al., 2011, 2012; Browning et al., 2018; Barber-Lluch et al., 2019). We are aware of many variables that could affect bacterial and phytoplankton biomass, for this reason, we compared the response of B vitamin treatments with their corresponding controls. B12, B1 and B12+B1 treatments were compared to the unamended control, while I+B12, I+B1, I+B12+B1 were compared with the I treatment.

RC/ I have some substantial concerns about the conclusions the authors make about community diversity and B-vitamins. Their exact statistical methods need to be better explained. Additionally, the authors need to fully explain the limits of their statistical methods, and not overstate or be overly speculative about the observed correlations between abiotic/biotic factors, B-vitamins, and the amplicon data. The manuscript needs substantial copy editing/English language editing. All sections need to be streamlined. The interpretation of results tends to be far too speculative. The authors need to only make claims that their data can support.

AC/ We agree that some statistical methods needed further clarification and we recognize that some analyses were somehow redundant and have been excluded from this revised version. Specifically, we have eliminated the Pearson correlation between response ratios and the clr (centered-log-ratio) abundance of taxonomic groups (reported in former table 1) as was redundant with the dbRDA. Regarding the RELATE analysis to explore the relationship between the responses to B vitamin treatments (response ratios of phytoplankton and bacteria) and (1) the environmental variables (including nutrients, temperature, salinity, B12, chla and BB), (2) the prokaryotic, or (3) eukaryotic community structure, we believe that nicely shows that the responses are only significantly related to the prokaryotic community structure. We have clarified how we constructed the resemblance matrices (L272-L280). It is important to note, that as we are aware of the statistical limitations when working with relative abundance of sequences, prior to statistical analyses, ASV abundances were transformed using the centered log ratio (Fernandes et al., 2014; Gloor et al., 2017).

RC/ The B12 analytical method appears to be derived from previously published methods. Specifically, those published by Heal et al. 2014, Sañudo et al. 2012, and Suffridge et al. 2017. It is troubling to me that the authors do not cite any of these papers in the methods section, despite the fact that the described method is a nearly an exact match of those described in the above papers. Additionally, SPE extraction efficiency and limits of detection need to be included.

AC/ We now provided all the requested details about the vitamin B12 quantification method and included all the references (L207-L217).

RC/ How were the whirl-pak bags prepared? Were they prepared to be trace clean? Were they sterile? What sort of plastic are they made out of? Trace metal or trace organic (B-vitamin) contamination is a real concern in experiments like these, especially when the authors want to make conclusions about the impact of a trace-component. Many plastics contain trace contamination from the factory, and if the bags were not properly prepared, this variability could interfere with all results.

AC/ We did not use strict trace metal clean techniques for sampling. It is important to note that the water for the experiments was pooled into a 20 l acid-cleaned carboy before filling the bags, thus all the bags would have the same incidental input of trace metals. We are aware that microbes could be limited by other trace elements or nutrients not considered in our treatments, such as iron or other B vitamins. For this reason, we based the discussion on the response ratios at the end of the experiments. The whirl-pak® bags are made of low density polyethylene, are sterile, non-toxic and transparent to the whole solar spectrum, thus avoiding UVR absorption of most other materials, and have been frequently used for experimentation with plankton communities (Gonzalez et al., 1990; Davidson and van der Heijden, 2000; Pakulski et al., 2007; Teixeira et al., 2018). The bags were not additionally treated as were used only once.

References

Barber-Lluch, E., Hernández-Ruiz, M., Prieto, A., Fernández, E. and Teira, E.: Role of vitamin B12 in the microbial plankton response to nutrient enrichment, Mar. Ecol. Prog. Ser., 626, 29–42, doi:10.3354/meps13077, 2019.

Bonnet, S., Guieu, C., Bruyant, F., Prášil, O., Van Wambeke, F., Raimbault, P., Moutin, T., Grob, C., Gorbunov, M. Y., Zehr, J. P., Masquelier, S. M., Garczarek, L. and Claustre, H.: Nutrient limitation of primary productivity in the Southeast Pacific (BIOSOPE cruise), Biogeosciences, 5, 215–225, doi:10.5194/bg-5-215-2008, 2008.

Browning, T. J., Rapp, I., Schlosser, C., Gledhill, M., Achterberg, E. P., Bracher, A. and Le Moigne, F. A. C.: Influence of Iron, Cobalt, and Vitamin B12 Supply on Phytoplankton Growth in the Tropical East Pacific During the 2015 El Niño, Geophys. Res. Lett., 45, 6150–6159, doi:10.1029/2018GL077972, 2018.

Davidson, A. and van der Heijden, A.: Exposure of natural Antarctic marine microbial assemblages to ambient UV radiation: effects on bacterioplankton, Aquat. Microb. Ecol., 21, 257–264, doi:10.3354/ame021257, 2000.

Downing, J. A., Osenberg, C. W. and Sarnelle, O.: Meta-analysis of marine nutrient-enrichment experiments: Variation in the magnitude of nutrient limitation, Ecology, 80, 1157–1167, doi:10.1890/0012-9658(1999)080[1157:MAOMNE]2.0.CO;2, 1999.

Elser, J. J., Bracken, M. E. S., Cleland, E. E., Gruner, D. S., Harpole, W. S., Hillebrand, H., Ngai, J. T., Seabloom, E. W., Shurin, J. B. and Smith, J. E.: Global analysis of nitrogen and phosphorus limitation of primary producers in freshwater, marine and terrestrial ecosystems, Ecol. Lett., 10, 1135–1142, doi:10.1111/j.1461-0248.2007.01113.x, 2007.

Fernandes, A. D., Reid, J. N., Macklaim, J. M., McMurrough, T. A., Edgell, D. R. and Gloor, G. B.: Unifying the analysis of high-throughput sequencing datasets: characterizing RNA-seq, 16S rRNA gene sequencing and selective growth experiments by compositional data analysis, Microbiome, 2, 15, doi:10.1186/2049-2618-2-15, 2014.

Gloor, G. B., Macklaim, J. M., Pawlowsky-Glahn, V. and Egozcue, J. J.: Microbiome datasets are compositional: And this is not optional, Front. Microbiol., 8, 1–6, doi:10.3389/fmicb.2017.02224, 2017.

Gobler, C. J., Norman, C., Panzeca, C., Taylor, G. T. and Sañudo-Wilhelmy, S. A.: Effect of B-vitamins (B1, B12) and inorganic nutrients on algal bloom dynamics in a coastal ecosystem, Aquat. Microb. Ecol., 49, 181–194, doi:10.3354/ame01132, 2007.

Gonzalez, J. M., Sherr, E. B. and Sherr, B. F.: Size-selective grazing on bacteria by natural assemblages of estuarine flagellates and ciliates, Appl. Environ. Microbiol., 56, 583–589, 1990.

Hedges, L. V., Gurevitch, J. and Curtis, P. S.: The Meta-Analysis of Response Ratios in Experimental Ecology, Ecology, 80(4), 1150, doi:10.2307/177062, 1999.

Koch, F., Marcoval, M. A., Panzeca, C., Bruland, K. W., Sañudo-Wilhelmy, S. A. and Gobler, C. J.: The effect of vitamin B12 on phytoplankton growth and community structure in the Gulf of Alaska, Limnol. Oceanogr., 56, 1023–1034,

doi:10.4319/lo.2011.56.3.1023, 2011.

Koch, F., Hattenrath-Lehmann, T. K., Goleski, J. A., Sañudo-Wilhelmy, S., Fisher, N. S. and Gobler, C. J.: Vitamin B1 and B12 uptake and cycling by plankton communities in coastal ecosystems, Front. Microbiol., 3, 1–11, doi:10.3389/fmicb.2012.00363, 2012.

Martínez-García, S., Fernández, E., Calvo-Díaz, A., Marañ̀Ąn, E., Morán, X. A. G. and Teira, E.: Response of heterotrophic and autotrophic microbial plankton to inorganic and organic inputs along a latitudinal transect in the Atlantic Ocean, Biogeosciences, 7, 1701–1713, doi:10.5194/bg-7-1701-2010, 2010.

Mills, M. M., Ridame, C., Davey, M., La Roche, J. and Geider, R. J.: Iron and phosphorus co-limit nitrogen fixation in the eastern tropical North Atlantic, Nature, 429, 292–294, doi:10.1038/nature02550, 2004.

Moore, C. M., Mills, M. M., Milne, A., Langlois, R., Achterberg, E. P., Lochte, K., Geider, R. J. and La Roche, J.: Iron limits primary productivity during spring bloom development in the central North Atlantic, Glob. Chang. Biol., 12, 626–634, doi:10.1111/j.1365-2486.2006.01122.x, 2006.

Pakulski, J., Baldwin, A., Dean, A., Durkin, S., Karentz, D., Kelley, C., Scott, K., Spero, H., Wilhelm, S., Amin, R. and Jeffrey, W.: Responses of heterotrophic bacteria to solar irradiance in the eastern Pacific Ocean, Aquat. Microb. Ecol., 47, 153–162, doi:10.3354/ame047153, 2007.

Sañudo-Wilhelmy, S. A., Gobler, C. J., Okbamichael, M. and Taylor, G. T.: Regulation of phytoplankton dynamics by vitamin B12, Geophys. Res. Lett., 33, 10–13, doi:10.1029/2005GL025046, 2006.

Teira, E., Hernando-Morales, V., Martínez-García, S., Figueiras, F. G., Arbones, B. and álvarez-Salgado, X. A.: Response of bacterial community structure and function to experimental rainwater additions in a coastal eutrophic embayment, Estuar. Coast. Shelf Sci., 119, 44–53, doi:10.1016/j.ecss.2012.12.018, 2013.
Teixeira, I. G., Arbones, B., Froján, M., Nieto-Cid, M., Álvarez-Salgado, X. A., Castro, C. G., Fernández, E., Sobrino, C., Teira, E. and Figueiras, F. G.: Response of phytoplankton to enhanced atmospheric and riverine nutrient inputs in a coastal upwelling embayment, Estuar. Coast. Shelf Sci., 210, 132–141, doi:10.1016/j.ecss.2018.06.005, 2018.

Please also note the supplement to this comment:
https://www.biogeosciences-discuss.net/bg-2019-306/bg-2019-306-AC3-supplement.zip

[Figure]

**Fig. 1.** (a) The NW Iberian margin and locations of the stations that were sampled, (b) distribution of daily coastal upwelling index (Iw)and (c) registered precipitations.

[Figure]

[Figure]

**Fig. 2.** Vertical distribution in the coastal and oceanic station of fluorescence ($\mu$g l-1), temperature (°C) and salinity (PSU) over time for February, April and August

[Figure]

**Fig. 3.** Initial biological conditions and abiotic factors at the coastal (st3) and oceanic (st6) sampling stations.

[Figure]

**Fig. 4.** (a) Averaged relative contribution of reads to the major taxonomic groups of eukaryotes and prokaryotes. (b) Averaged B12 concentration (pM).

[Figure]

**Fig. 5.** Phytoplankton biomass (estimated as Chl-a concentration) ($\mu$g l-1) in the time-zero of each experiment (striped bars) and in the final-time of each treatment (colored bars) in the experiments.

[Figure]

**Fig. 6.** Bacterial biomass ($\mu$gC l-1) in the time-zero of each experiment (striped bars) and in the final-time of each treatment (colored bars) in the experiments.

[Figure]

**Fig. 7.** Monthly averaged response ratio (RR) of (a) total phytoplankton community and of (b) bacterial community at surface and SCM in the coastal and oceanic station.

[Figure]

**Fig. 8.** Distance based redundancy analysis (dbRDA) of B vitamin responses by microbial plankton based on Bray-Curtis similarity.

---

## Author Response (AR1)

Vanessa Joglar

Biological Oceanography Group

Koji Suzuki

Associate Editor

Biogeosciences

Vigo, 29th November 2019

Dear Koji

Please find attached a revised version of manuscript entitled "Spatial and temporal variability in the response of phytoplankton and bacterioplankton to B-vitamin amendments in an upwelling system". The manuscript was co-authored by myself, Antero Prieto, Esther Barber-Lluch, Marta Hernández-Ruíz, Emilio Fernandez and Eva Teira.

We would like to acknowledge the insightful and constructive comments of the reviewers, which clearly helped us to improve the overall merit of the manuscript. We have taken into account all the suggestions raised resulting in a higher quality work. We attach a detailed response to all the comments made by the reviewers. In the individual responses to referee comments (RCs), the suggestions and comments of the individual reviewers are in plain font and our responses are in italics and blue font. The major changes are summarized below:

- Both referees agreed that the calculated response ratios could be useful in the discussion and that the actual chlorophyll-a and bacterial biomass data at the beginning and after the 72 h incubation in all the treatments should be shown. As we agree that this information is of great value, we have prepared two new figures (Fig. 5 and 6) where we plot all the requested information. In the revised version, the former figure 5, showing the response ratios has been included as supplementary information.

- Another suggestion by both referees was to provide details about the methodology related to B12 measurement. This section has been expanded by referring to previous work and indicating the particular conditions and instruments used in our laboratory.

- We have fully reviewed the statistical analyses. We now better explain the RELATE analyses. We have also reconsidered the usefulness of the Pearson's correlation analyses between the individual responses to vitamins and prokaryotic taxa (summarized in former table 1). We believe that this analyses could be redundant and its conclusions somewhat speculative. Based on this, we have removed this analysis, eliminated the former table 1, and fully reviewed the discussion, eliminating speculative statements, and toning down some of our conclusions.

- The genomic data of this study will be publicly available at the European Nucleotide Archive (ENA) at EMBL-EBI (https://www.ebi.ac.uk/ena) as soon as possible.

Looking forward to hearing from you,

Vanessa Joglar

Full address for corresponding is:

    Grupo de Oceanografía Biológica

    Departamento de Ecología y Biología Animal

    Universidad de Vigo

    Campus Universitario Lagoas-Marcosende

    36310-Vigo Spain

E-mail: vjoglar@uvigo.es

**Referee1 comments**

*We very much appreciate the useful and constructive comments made by the reviewer, which surely contribute to improve the manuscript quality. We have considered all the suggestions and made the requested modifications, as detailed below.*

My main initial request is to include figures for the actual bacterial and phytoplankton biomass changes in the experiments, rather than simply ratios, including the values for the initial conditions. I believe this should be in the main manuscript, not just the Supporting Information. These data can be displayed as a mean with error bars representing the spread across the three treatment replicates. I believe this will give a better indication of how the community responded in the experiments. The ratio figures can be included too for discussion/interpretation purposes. Please also label the treatments below each bar in each case – I found treatment identification a little difficult in the current figures.

*We have now included in the main manuscript the requested figure (new figure 5 and 6). We present the response of phytoplankton and that of bacteria separately. The former figure 5 with the response ratios is now included as supplementary information.*

Secondly I think the manuscript should also note how trace metal contamination could have biased the results. This is currently not discussed at all, but could have had an important influence. For instance, if contaminating iron had been inadvertently included in the treatments. Contamination would likely originate from the metal CTD-rosette, the rosette bottles, during bottle sampling, from the incubation bags, from the nutrient additions etc. Where certain procedures were carried out to reduce this, these should be described. This is significant, as this microbes in this region could be experiencing primary iron limitation – see Blain et al. (2004).
Blain, S., Guieu, C., Claustre, H., Leblanc, K., Moutin, T., Quéguiner, B., Ras, J. and Sarthou, G., 2004. Availability of iron and major nutrients for phytoplankton in the northeast Atlantic Ocean. Limnology and Oceanography, 49(6), pp.2095-2104.

*We did not use trace metal clean techniques for sampling. We used standard stainless CTD-rosette and Niskin metal-free bottles. We did not have a trace metal clean lab on board, so, even though samples were carefully manipulated contamination by trace metals could have eventually occurred. It is important to note that the water for the*

*experiments was pooled into a 20 l acid-cleaned carboy before filling the bags (L134-L136), thus all the bags would have the same incidental input of trace metals. We are aware that microbes could be limited by other trace elements or nutrients not considered in our treatments, such as iron or other B vitamins. For this reason, we based the discussion on the response ratios at the end of the experiments.*

**Specific comments**

In the abstract I would recommend making reference to the study region (i.e. 'North east Atlantic', or 'off the northwest coast of Spain')

*The study area has been specified in the abstract (L15)*

Figure 1b and c: please indicate when experiments were sampled for (i.e. which day? day 0?)

*Sampling day for each experiment has been indicated in the graphs (Fig. 1b and Fig. 1c)*

I would recommend noting the microbial responses to major nutrient supply, in addition to B12/B1, in the abstract.

*The response to inorganic nutrient additions has been considered in the abstract (L13-L14)*

Line 15–16: rephrase 'was not of great concern'

*This has been rewritten (L15)*

I would recommend stating the number of the 36 experiments where bacteria/ phytoplankton responded positively/negatively to vitamin supply in the abstract.

*This information has been included in the abstract (L16-L19)*

Line 21 'Growth stimulation by B1 addition was more frequent on bacteria' – relative to phytoplankton?

*This has been clarified (L22-L23)*

Lines 35–36 and elsewhere: I would recommend seeing the more recent studies of Browning et al., 2017 and Browning et al., 2018, which also perform trace-metal-clean B12 addition bioassay experiments in upwelling/coastal/offshore regions.

Browning, T.J., Achterberg, E.P., Rapp, I., Engel, A., Bertrand, E.M., Tagliabue, A. and Moore, C.M., 2017. Nutrient co-limitation at the boundary of an oceanic gyre. Nature, 551(7679), p.242.

Browning, T.J., Rapp, I., Schlosser, C., Gledhill, M., Achterberg, E.P., Bracher, A. and Le Moigne, F.A., 2018. Influence of iron, cobalt, and vitamin B12 supply on phytoplankton growth in the tropical East Pacific during the 2015 El Niño. Geophysical Research Letters, 45(12), pp.6150-6159.

*Both studies have been cited in the revised version (L38)*

Line 39: synthesized by prokaryotes and archaea?

*Archaea is included within the prokaryote organisms. This has been clarified in the manuscript (L41)*

Line 42: Have not defined 'cobalamin' (In general I recommend choosing B12 or cobalamin and sticking to it throughout)

*This has been corrected (L45)*

Line 79: Perhaps mention here succinctly what Gobler et al. (2007) found?

*This has been explained in the revised text (L83-L84)*

Line 79: the reference Barber-Lluch et al. (2019) does not appear in the reference list ă˘A.

*This citation has been included in the reference list (L620-L622)*

Lines 114–115: How was this water sampled? From the regular stainless CTD? If so, trace element contamination should be acknowledged. Also see general comment.

*This has been acknowledged in the revised manuscript (L120; L134-L136)*

Line 125: Was there any treatment of the whirl-pak bags (e.g. acid and deionized water rinses) to remove contamination? Also see general comment.

*We used these bags mainly because they are sterile, non-toxic and transparent to the whole solar spectrum, thus avoiding UVR absorption of most other materials, and have been frequently used for experimentation with plankton communities (Gonzalez et al., 1990; Davidson and van der Heijden, 2000; Pakulski et al., 2007; Teixeira et al., 2018). The bags were not additionally treated as were used only once.*

Line 127 and on: What were the chemical stocks of the nutrients (e.g. brand and purity). Again, if these nutrients were not pre-treated to remove trace element contamination, this should be acknowledged. Also see general comment.

*All the reagents were from Sigma of highest purity. Stocks were prepared with autoclaved Milli-Q water. No additional treatment for trace element contamination was applied.*

Line 137: Were the tanks screened, or open to the air?

*Tanks were screened to attenuate light intensity (L148-L150)*

Line 147: Was any time given for the fixative to act on cells before flash freezing in liquid nitrogen?

*Samples were incubated 20 min for fixative to act (L160-L161)*

Section 2.5: If known, what was the recovery percent of the B12 preconcentration/ extraction? (i.e. via use of a standard)

*Average B12 recovery percentage was 93% (L213-L214)*

Line 271: How was the upwelling index calculated (cannot see this in methods).

*Upwelling index was calculated by calculating the Ekman transport from surface winds at fix-station (st3) located at 42° N and 8.88° W. This information has been included in the revised text (L129-L132)*

Figure 5: It is not clear that the value being displayed is the RR Chla OR RR BB and not the ratio of these.

*This has been clarified (Fig. S3 in the supplement)*

As the Figure 5 has signs indicating statistical significance, the error in the spread across the treatment replicates must have been prorogated somehow? Can this error be included as error bars in the figure?

*The error bars representing the standard error of the three replicates have been included in the new figure 5 (now Figure 5 and 6).*

337–338: Specifically which experiments showed serial limitation by B vitamins?

*This has been specified in the revised manuscript (L387-L390)*

Line 402: 'clarify the paper of vitamins'?

*This fragments has been corrected for clarity (L444)*

Lines 417–419: Please distinguish between the phytoplankton/bacteria responses in this value of 75%.

*Taking into account the responses of phytoplankton and bacteria separately, the percentages were 75% for phytoplankton and 50% for bacteria (L457-L548)*

Line 425: No full stop (perhaps also rephrase to 'community assemblage'?)

*This sentence was removed.*

Lines 491–495: This doesn't quite make sense – in the first sentence it states that phytoplankton responses to B1 supply were restricted, and in the second the stimulation of phytoplankton is discussed.

*Phytoplankton responses to B1 are overall restricted. The second sentence refers to the particular simultaneous stimulation of phytoplankton and bacteria by B1 addition found in subsurface oceanic waters in February (L532-L533)*

I would advise including a table summarizing initial conditions (i.e., nutrient concentrations, temperature, chlorophyll-a, initial bacteria and so on).

*We have added a supplementary table including detailed information about initial conditions (Table S2 in the supplement)*

In addition to the modifications suggested by the reviewer, we have made the following changes. The named OTUs (operation taxonomic units) has been replaced by ASV (amplicon sequence variant) due to the sequence analysis method used DADA2 (section 2.6).

We have eliminated the Pearson correlation between response ratios and the clr (centered-log-ratio) abundance of taxonomic groups (reported in former table 1) as was redundant with the dbRDA. Regarding the RELATE analysis to explore the relationship between the responses to B vitamin treatments (response ratios of phytoplankton and bacteria) and (1) the environmental variables (including nutrients, temperature, salinity, B12, chla and BB), (2) the prokaryotic, or (3) eukaryotic community structure, we believe that nicely shows that the responses are only significantly related to the prokaryotic community structure.

**Referee 2 comments**

*We are very grateful for the reviewer's comments, which have contributed to improve the manuscript.*

The role that the availability of B-vitamins,specifically vitamin B12 and B1,play in shaping the marine microbial community is very relevant. The authors of this manuscript conducted an extensive experimental campaign with the goal of providing some insight to these processes. Unfortunately, their findings are poorly communicated and overstated in this manuscript. Most of the discussion is highly speculative and is insufficiently referenced.

*We have made our best to refer adequately all the relevant studies and eliminating some speculative statements.*

The authors have gone "all-in" on the poorly justified concept of "response ratio". I feel like this calculated metric is overly general and prevents an in-depth analysis of the actual data which likely contains subtle variations that could either support or undermine the authors primary conclusions. I don't understand why the authors chose to use response ratios rather more traditional ecological and physiological metrics. While response ratios could be a useful part of the discussion, they should be just that, a part of the discussion. Additionally, the authors ignore the rates of community growth and dynamics and only assess the response at the end time point relative to the initial point. While it is not possible at this point to change the experimental design, the authors need to change their interpretation of the data to acknowledge the limits of their data.

*We are aware that sampling only at one endpoint (after 72 h incubation in our case) does not allow to discuss in detail the dynamics during each experiment, however we were particularly interested in extensively exploring the temporal and spatial variability of the response to vitamin enrichment. The experimental design, involved 36 experiments, with 8 triplicate treatments (24 experimental units per experiment). Even sampling only at the*

beginning and at the end we collected 972 samples for chlorophyll-a, and 972 for bacterial biomass. Initial and endpoint sampling is a common practice in enrichment microcosm experiments (e. g. Mills et al., 2004; Moore et al., 2006; Gobler et al., 2007; Bonnet et al., 2008; Koch et al., 2011), and allows the estimation of net growth rates using the following formula: ln (endpoint biomass/initial biomass)/incubation time. A previous work by Barber-Lluch et al (2019) in the same sampling area allowed us to conclude that sampling at 72 h was adequate to explore the effect of vitamins on both phytoplankton and bacterial biomass. As we agree with the referee that the dynamics of phytoplankton and bacteria during the experiments are of interest, and following also the advice of referee 1, we now include in the manuscript two new figures where the initial and endpoint value of chlorophyll-a and bacterial biomass is represented. The response ratio figure is now included in the supplementary information. We accordingly now describe the dynamics of both planktonic components in the different experiments. We nevertheless decided to keep the response ratio as a measure of the magnitude of the effect (see below), which is very useful for the sake of comparison.

We used here the response ratio as the quotient between the measured quantity of a response variable in experimental and control experimental units. Previous studies dealing with the effects of nutrients additions on microbial communities have noted the importance of expressing the change in the treatment relative to the control (Downing et al., 1999; Hedges et al., 1999; Elser et al., 2007, among others). We find that this variable is particularly adequate as a measure of the experimental effect because it quantifies the proportionate change that results from an experimental manipulation. This metric is widely use in marine ecology, and particularly in nutrient amendment experiments (e.g. Martínez-García et al., 2010; Teira et al., 2013; Barber-Lluch et al., 2019). The use of the response ratio calculated from endpoint biomass data provides the same information as the comparison of growth rates between treatments. Below we plot, as an example, the relationship between the response ratio from biomass (endpoint biomass in treatment divided by endpoint biomass in control) and the difference between growth rates (growth

*rate in treatment minus growth rate in the control) using data from two of our experiments (represented with different colours). It can be appreciated that the information provided by the response ratios follows exactly the same pattern as that provided by comparing growth rates. Moreover, the range of variation is higher for the response ratio, which allows to statistically detecting more subtle changes.*

[Figure]

It is unfortunate that the only measures of biomass performed by these authors during their experiments were bacterial abundance and chlorophyll A. These are very broad, unspecific measures of community structure, that can be impacted by a myriad of environmental factors. The authors make some substantial claims about the roles that B-vitamin additions are playing on the microbial community; however, I wonder if they really have enough resolution in their measurements to make these claims. The author's use of "response rate" to obscures the fact that they are only measuring bacterial abundance and chlorophyll concentration. There are so many variables that impact these measures, it's not clear to me that the authors are actually looking at responses from B-vitamins.

*B vitamins are essential growth factors for all microorganisms; therefore, the ultimate effect of a vitamin deficiency will be an impairment of growth, which is typically evaluated*

*from changes in biomass. It is true that we only measure the effect on bulk phytoplankton and bacteria, and thus we have toned down all the conclusions about the effect on microbial community structure. We do not think that is unfortunate to have chosen phytoplankton and bacterial biomass as response variable, considering that most previous studies evaluating the role of B vitamins were based on biomass measurements (Sañudo-Wilhelmy et al., 2006; Gobler et al., 2007; Koch et al., 2011, 2012; Browning et al., 2018; Barber-Lluch et al., 2019). We are aware of many variables that could affect bacterial and phytoplankton biomass, for this reason, we compared the response of B vitamin treatments with their corresponding controls. B12, B1 and B12+B1 treatments were compared to the unamended control, while I+B12, I+B1, I+B12+B1 were compared with the I treatment.*

I have some substantial concerns about the conclusions the authors make about community diversity and B-vitamins. Their exact statistical methods need to be better explained. Additionally, the authors need to fully explain the limits of their statistical methods, and not overstate or be overly speculative about the observed correlations between abiotic/biotic factors, B-vitamins, and the amplicon data. The manuscript needs substantial copy editing/English language editing. All sections need to be streamlined. The interpretation of results tends to be far too speculative. The authors need to only make claims that their data can support.

*We agree that some statistical methods needed further clarification and we recognize that some analyses were somehow redundant and have been excluded from this revised version. Specifically, we have eliminated the Pearson correlation between response ratios and the clr (centered-log-ratio) abundance of taxonomic groups (reported in former table 1) as was redundant with the dbRDA. Regarding the RELATE analysis to explore the relationship between the responses to B vitamin treatments (response ratios of phytoplankton and bacteria) and (1) the environmental variables (including nutrients, temperature, salinity, B12, chla and BB), (2) the prokaryotic, or (3) eukaryotic community structure, we believe that nicely shows that the responses are only significantly related to the prokaryotic community structure. We have clarified how we constructed the resemblance matrices (L273-L283).*

*It is important to note, that as we are aware of the statistical limitations when working with relative abundance of sequences, prior to statistical analyses, ASV abundances were transformed using the centered log ratio (Fernandes et al., 2014; Gloor et al., 2017).*

The B12 analytical method appears to be derived from previously published methods. Specifically, those published by Heal et al. 2014, Sañudo et al. 2012, and Suffridge et al. 2017. It is troubling to me that the authors do not cite any of these papers in the methods section, despite the fact that the described method is a nearly an exact match of those described in the above papers. Additionally, SPE extraction efficiency and limits of detection need to be included.

*We now provided all the requested details about the vitamin B12 quantification method and included all the references (Section 2.5, L191-L194, L205-L215).*

How were the whirl-pak bags prepared? Were they prepared to be trace clean? Were they sterile? What sort of plastic are they made out of? Trace metal or trace organic (B-vitamin) contamination is a real concern in experiments like these, especially when the authors want to make conclusions about the impact of a trace-component. Many plastics contain trace contamination from the factory, and if the bags were not properly prepared, this variability could interfere with all results.

*We did not use strict trace metal clean techniques for sampling. It is important to note that the water for the experiments was pooled into a 20 l acid-cleaned carboy before filling the bags, thus all the bags would have the same incidental input of trace metals. We are aware that microbes could be limited by other trace elements or nutrients not considered in our treatments, such as iron or other B vitamins. For this reason, we based the discussion on the response ratios at the end of the experiments.*

*The whirl-pak® bags are made of low density polyethylene, are sterile, non-toxic and transparent to the whole solar spectrum, thus avoiding UVR absorption of most other materials, and have been frequently used for experimentation with plankton communities*

*(Gonzalez et al., 1990; Davidson and van der Heijden, 2000; Pakulski et al., 2007; Teixeira et al., 2018). The bags were not additionally treated as were used only once.*

[revised manuscript text omitted]

fig. 02

[Figure]

fig. 03

Initial Nutrients and Biomasses

[Figure]

[Figure]

fig. 04

fig. 05

[Figure]

fig. 06

[Figure]

fig. 07

[Figure]

fig. 08

[Figure]

**Supplement information**

**Table S1:** concentration of hydroxocobalamin (OHB12) and cyanocobalamin (CNB12)

in seawater samples corresponding to the initial time of the experiments. Abbreviations:

Not detected (nd) and lower concentration of the quantification limit (<LOQ).

| Sample ID | Station | Depth | Month | OHB12 pM | CNB 2 pM |
|---|---|---|---|---|---|
| 1602_st3_d1_p1 | coast | surface | February | 0.21 | nd |
| 1602_st3_d3_p1 | coast | surface | February | 0.20 | nd |
| 1602_st3_d5_p1 | coast | surface | February | 0.26 | nd |
| 1604_st3_d1_p1 | coast | surface | April | 0.47 | nd |
| 1604_st3_d3_p1 | coast | surface | April | 0.66 | nd |
| 1604_st3_d5_p1 | coast | surface | April | 0.23 | nd |
| 1608_st3_d1_p1 | coast | surface | August | 0.30 | nd |
| 1608_st3_d3_p1 | coast | surface | August | 0.38 | nd |
| 1608_st3_d5_p1 | coast | surface | August | 0.19 | nd |
| 1602_st3_d1_p2 | coast | SCM | February | 0.36 | nd |
| 1602_st3_d3_p2 | coast | SCM | February | 0.10 | nd |
| 1602_st3_d5_p2 | coast | SCM | February | 0.41 | nd |
| 1604_st3_d1_p2 | coast | SCM | April | 0.32 | nd |
| 1604_st3_d3_p2 | coast | SCM | April | 0.27 | nd |
| 1604_st3_d5_p3 | coast | SCM | April | 0.15 | nd |
| 1608_st3_d1_p2 | coast | SCM | August | 0.46 | nd |
| 1608_st3_d3_p2 | coast | SCM | August | 0.21 | nd |
| 1608_st3_d5_p2 | coast | SCM | August | 0.39 | nd |
| 1602_st6_d1_p1 | ocean | surface | February | 0.31 | nd |
| 1602_st6_d3_p1 | ocean | surface | February | 0.09 | nd |
| 1602_st6_d5_p1 | ocean | surface | February | 0.06 | nd |
| 1604_st6_d1_p1 | ocean | surface | April | 0.13 | nd |
| 1604_st6_d3_p1 | ocean | surface | April | 0.09 | nd |
| 1604_st6_d6_p1 | ocean | surface | April | 0.04 | nd |
| 1608_st6_d1_p1 | ocean | surface | August | 0.20 | nd |
| 1608_st6_d3_p1 | ocean | surface | August | 0.09 | nd |
| 1608_st6_d6_p1 | ocean | surface | August | 0.14 | nd |
| 1602_st6_d1_p3 | ocean | SCM | February | 0.21 | 0.55 |
| 1602_st6_d3_p2 | ocean | SCM | February | 0.08 | nd |
| 1604_st6_d1_p2 | ocean | SCM | April | nd | nd |
| 1604_st6_d3_p2 | ocean | SCM | April | 0.07 | nd |
| 1604_st6_d6_p2 | ocean | SCM | April | 0.05 | nd |
| 1608_st6_d1_p2 | ocean | SCM | August | 0.19 | nd |
| 1608_st6_d3_p2 | ocean | SCM | August | 0.09 | nd |
| 1608_st6_d6_p2 | ocean | SCM | August | 0.16 | nd |

**Table S2:** Summary of initial conditions for each experiment (expt). Sampling months were February (Feb), April (Apr) and August (Aug).

| Station | Depth | Month | Expt | Temp °C | Sal | NO$_3^-$ µM | NO$_2^-$ µM | NH$_4^+$ µM | HPO$_4^{2-}$ µM | DIN:P µM | SiO$_4^{2-}$ | Chl-a µg l$^{-1}$ | BB µgC l$^{-1}$ |
|---|---|---|---|---|---|---|---|---|---|---|---|---|---|
| Coast | surface | Feb | 3a | 13.75 | 35.02 | 2.86 | 0.19 | 0.35 | 0.17 | 19.65 | 3.62 | 1.39 | 1.84 |
| | | | 3b | 13.22 | 34.27 | 4.89 | 0.36 | 0.51 | 0.33 | 17.25 | 6.77 | 0.73 | 1.91 |
| | | | 3c | 13.43 | 34.21 | 4.63 | 0.19 | 0.09 | 0.18 | 27.68 | 8.57 | 4.86 | 3.45 |
| | | Apr | 3a | 12.96 | 34.58 | 2.21 | 0.24 | 0.32 | 0.19 | 14.55 | 5.24 | 2.73 | 7.88 |
| | | | 3b | 13.31 | 34.25 | 12.46 | 0.36 | 0.54 | 0.41 | 32.73 | 12.57 | 1.40 | 9.17 |
| | | | 3c | 14.04 | 31.83 | 4.18 | 0.16 | 0.55 | 0.19 | 25.90 | 10.52 | 2.18 | 4.30 |
| | | Aug | 3a | 14.14 | 35.60 | 0.50 | 0.10 | 0.84 | 0.12 | 11.77 | 1.11 | 5.73 | 14.64 |
| | | | 3b | 14.36 | 35.61 | 0.81 | 0.08 | 1.08 | 0.20 | 9.95 | 0.28 | 5.52 | 6.39 |
| | | | 3c | 13.66 | 35.16 | 3.93 | 0.17 | 0.12 | 0.33 | 12.78 | 3.86 | 5.64 | 10.61 |
| | SCM | Feb | 3a | 13.73 | 35.71 | 3.58 | 0.14 | 0.04 | 0.31 | 12.13 | 5.25 | 0.21 | 1.30 |
| | | | 3b | 13.91 | 35.27 | 4.16 | 0.15 | 0.07 | 0.37 | 11.91 | 4.63 | 0.99 | 1.83 |
| | | | 3c | 13.45 | 34.66 | 2.94 | 0.09 | 0.10 | 0.17 | 18.37 | 6.13 | 4.98 | 2.36 |
| | | Apr | 3a | 12.80 | 35.34 | 3.22 | 0.34 | 0.46 | 0.28 | 14.34 | 4.39 | 0.99 | 5.90 |
| | | | 3b | 13.22 | 35.28 | 0.24 | 0.07 | 0.12 | 0.04 | 10.19 | 2.83 | 2.15 | 9.47 |
| | | | 3c | 13.92 | 34.95 | 0.21 | 0.07 | 0.10 | 0.06 | 6.52 | 3.41 | 2.18 | 9.51 |
| | | Aug | 3a | 13.58 | 35.62 | 0.91 | 0.13 | 0.23 | 0.15 | 8.32 | 1.68 | 20.75 | 12.71 |
| | | | 3b | 13.82 | 35.61 | 1.40 | 0.16 | 0.14 | 0.23 | 7.49 | 1.40 | 20.07 | 1.73 |
| | | | 3c | 13.38 | 35.63 | 5.29 | 0.13 | 0.14 | 0.41 | 13.47 | 3.93 | 4.63 | 9.21 |
| Ocean | surface | Feb | 6a | 13.98 | 30.20 | 1.32 | 0.18 | 0.11 | 0.16 | 10.07 | 3.23 | 0.82 | 2.38 |
| | | | 6b | 14.16 | 35.86 | 0.90 | 0.11 | 0.04 | 0.12 | 9.15 | 2.29 | 1.20 | 2.98 |
| | | | 6c | 14.10 | 35.40 | 1.03 | 0.15 | 0.13 | 0.16 | 8.43 | 2.97 | 2.08 | 2.92 |
| | | Apr | 6a | 13.44 | 35.68 | 0.95 | 0.11 | 0.06 | 0.12 | 9.63 | 2.31 | 1.51 | 6.58 |
| | | | 6b | 13.59 | 35.66 | 0.47 | 0.11 | 0.06 | 0.08 | 8.33 | 2.71 | 1.29 | 7.37 |
| | | | 6c | 13.93 | 35.57 | 0.12 | 0.03 | 0.06 | 0.04 | 4.90 | 2.08 | 0.75 | 11.76 |
| | | Aug | 6a | 15.97 | 35.61 | 0.05 | 0.01 | 0.06 | 0.02 | 4.88 | 1.46 | 0.65 | 39.38 |
| | | | 6b | 16.04 | 35.59 | 0.26 | 0.01 | 0.09 | 0.05 | 7.46 | 3.21 | 0.99 | 11.46 |
| | | | 6c | 15.34 | 35.53 | 0.45 | 0.04 | 0.05 | 0.07 | 7.38 | 1.37 | 1.30 | 5.63 |
| | SCM | Feb | 6a | 14.08 | 35.75 | 1.73 | 0.20 | 0.04 | 0.18 | 11.18 | 3.47 | 0.88 | 2.28 |
| | | | 6b | 14.10 | 35.76 | 1.60 | 0.19 | 0.02 | 0.15 | 11.75 | 2.86 | 1.22 | 3.18 |
| | | | 6c | 14.13 | 35.82 | 1.13 | 0.18 | 0.12 | 0.16 | 9.17 | 2.92 | 2.39 | 3.49 |
| | | Apr | 6a | 13.28 | 35.69 | 1.63 | 0.31 | 0.10 | 0.18 | 11.51 | 3.16 | 1.61 | 5.38 |
| | | | 6b | 13.28 | 35.68 | 1.45 | 0.33 | 0.12 | 0.16 | 11.88 | 2.42 | 1.50 | 6.96 |
| | | | 6c | 13.72 | 35.60 | 0.03 | 0.06 | 0.07 | 0.05 | 3.01 | 1.89 | 1.45 | 11.74 |
| | | Aug | 6a | 14.90 | 35.60 | 0.00 | 0.04 | 0.10 | 0.03 | 4.20 | 1.44 | 0.84 | 26.55 |
| | | | 6b | 15.95 | 35.60 | 0.27 | 0.00 | 0.07 | 0.05 | 6.45 | 2.79 | 1.11 | 6.04 |
| | | | 6c | 15.41 | 35.62 | 0.35 | 0.06 | 0.06 | 0.07 | 6.51 | 1.66 | 1.41 | 5.45 |

**Figure S1:** A multidimensional scaling (MDS) showing the distance according to similarity in the microbial plankton composition at the beginning of each experiment (each symbol). Filled and open symbols represent samples from coastal and oceanic station, respectively, numbers correspond to the sampling station, triangles and circles represent samples from surface and SCM, respectively, and colours correspond to the months: (green) February, (blue) April and (pink) August.

**Figure S2:** Response ratio (RR) to inorganic nutrient addition (averaged biomass at the end of the experiments by the averaged value in the control) of total phytoplankton community (smooth bars) and of bacterial biomass (striped bars) at (a) coastal and (b) oceanic station. Each bar corresponds to one of the 3 experiments (a, b or c) performed in each depth and station during February, April and August. Colours represent samples from (light grey) surface and (dark grey) SCM. Horizontal line represents a response equal to 1, that means no change relative to control. Asterisks indicate phytoplankton significant response relative to control (t-test; * $p < 0.05$) and circle indicate bacterial significant response relative to the control (t-test; [0] $p < 0.05$). Note that different scales were used.

**Figure S3:** Response ratio (RR) of total phytoplankton community (smooth bars) and of bacterial biomass (striped bars) at (a) surface and (b) SCM in the coastal station and at (c) surface and (d) SCM in the oceanic waters. Treatments represented are: B12; B1; B12+B1 in pink tones and I+B12/I; I+B1/I; I+B12+B1/I in green tones. Pink bars represent primary responses to B vitamins and green bars represent secondary responses to B vitamins. Horizontal line represents a response equal to 1, that means no change relative to control in the primary responses, and no change relative to inorganic treatment in the secondary responses. Asterisks indicate phytoplankton significant response (t-test;

* $p < 0.05$) and circle indicate bacterial significant response (t-test; $^{o}$ $p < 0.05$). Note that different scales were used.

Figure S1

[Figure]

Transform: Square root
Resemblance: Bray Curtis similarity

stationmonthdepth
△ 3Feb0m
● 3FebSCM
▲ 3Apr0m
● 3AprSCM
▲ 3Aug0m
● 3AugSCM
△ 6Feb0m
○ 6FebSCM
△ 6Apr0m
○ 6AprSCM
○ 6Aug0m
△ 6AugSCM

Figure S2

[Figure]

[Figure]

Figure S3

---

## Referee Report (RR1)

Review of "Spatial and temporal variability in the response of phytoplankton and bacterioplankton to B-vitamin amendments in an upwelling system" by Joglar et al.

**General comments**

This manuscript covers a very interesting and highly relevant topic, which the authors focuses on in the manuscript. Dynamics of B-vitamins in the worlds ocean is not often studied and this manuscript attempts to provide important information on this topic. The sampling campaign is definitely impressive, as well as the work that went in to the study. I don't agree with all the comments from the previous reviewers, for instance I find the use of response ratios very informative and a great display of the results. With that said, I have some comments and concerns with the manuscript that need to be addressed.

One main problem with the manuscript is that only one of the vitamins investigated is analyzed. I realize this may be due to problem quantifying B1 in natural sea water, which the authors can state more clearly. It would also have been very interesting to have quantified the cellular content in the two size fractions, but unfortunately this was not done.

Generally, is there any benefit of using st3 and st6 instead of coastal and oceanic station? I feel the readability would increase if you used coastal and oceanic instead.

When referring to figures, state which of the figures, a, b or c. you refer to. Also, look over all figures so that all are labelled a, b, c …. For fig. 2 and 3 I'm having troubles seeing the benefits of having several a's, several b's etc. I would like to see labelling a-r instead for fig. 2, then you can refer to the specific mosaic.

I would like to see a more accurate reporting of statistics. Please provide statistics value (t, F, df) when appropriate.

In my opinion, the results should be presented as averages, per station and cruise and ignore 3a, 3b, 3c etc. I understand that a tremendous amount of work has gone into this experiment, but I believe that the paper would benefit from a more succinct and concise result section.

**Specific comments**

**Abstract**

L15; "… unimportant, …" – I would suggest changing wording, as you cannot know if it is unimportant or not. Maybe "slight" or "limited"?

L15; how can an "unimportant" variability lead to the assumption that there are factors operating at other scales? Requires clarification.

L20; change "alone" to solely?

L22-24; auxotrophy is also high in phytoplankton, causing the argument to halter a bit. I would suggest mentioning this as well and combine it with bacteria dependence.

**Introduction**

L34; state which toxic episodes you refer to.

L60-61; I would suggest reading Cruz-Lopez et al. 2016.

L69-74; I would suggest reviewing if you really need all references to say what you want to say. In a relatively short sentence, you use 13 references.

L100 & L110; decide if you use numbers or text, 36 or thirty-six, and use throughout.

L105; change "synthetize" to synthesize.

**Methods**

L119; What is the timeframe between a, b and c? Looking at fig 1 I realized I can figure it out, but it is a very unclear way to present samplings.

L120-123; To increase clarity, I would recommend to state that surface is 5m deep more clearly.

L123; State which occasion this sampling failed.

L128-129; do you refer to the t0 for each experiment (a, b and c)? Needs clarification.

L129-132; Does the UI provide you with important information?, now sentence feels a bit dropped in the text.

L133-134; What about small zooplankton, copepodites and nauplii? Did you check for this, if so it should be stated. If not, the potential impact of these should be taken into account for.

L138-144; This is a very confusing way to present the treatments. I would suggest providing all of this important information in a table instead. Additionally, the rationale behind the levels of nutrients and vitamins should be given.

L147-150; This is unclear to read. First it is natural conditions, then the conditions were reproduced? How was this done? What screens are you referring to?

L152; change to "t0"?

L160-162; Revise sentence. Suggestion "Samples were incubated 20 min for the fixative to act on cells, immersed in liquid nitrogen for 15 min before being frozen at -80ºC."

L169-170; Could the usage of two different factors cause a problem in the interpretation of the data, when comparing coast and oceanic station?

L173; "… first place…" before all other variables? If so, please clarify.

L174-177; Revise sentence. Suggestion "Polyethylene bottles (50 ml, pre cleaned with 5% HCl were filled with the sample using contamination-free plastic gloves and immediately frozen at –20°C until analysis, using standard colorimetric methods with a Bran-Luebbe segmented flow analyzer (Hansen and Grasshoff 1983)." Or did I misunderstood "free-contamination"?

L182; Unfortunately, you only have samples for dissolved B12. This should be specified.

L183; Specify when the fifth or sixth day was sampled, as it can influence the results.

L188; Do you refer to leftover water? If so, change wording. If not, clarify.

L199-200; State which values apply for length, inner diameter and particle size of column.

L211; You have not used subscript before, change to B12.

L211-212; State which congener had which LOD.

L212; If the case, state that 0.05 is for cyanocobalamin, CNB12. Also, change to hydroxocobalamin.

L214; You have not stated what CNB12 is.

L219; Why was plankton community sampled day 1, 2, 4, 6, while B12 was sampled day 1, 3, 5/6?

L222; Change "litters" to liters.

L237-238; Can you update with the accession numbers?

L245-247; How can this be? Is it fragments of cells going through the 3 μm filter? Would benefit from an explanation for this.

L251-252; Please provide the rationale for this procedure.

L265; "… if necessary to attain normality". Was this not always the case, do you have some samplings where the data was not normalized and some where it is? If so, you should state when this was the case and discuss how this might affect the results and conclusions drawn from them.

L266; When "standardizing", do you refer to using the corrected p value?

L267-273; Why using ANOVA and Z-test? The reasoning behind this choice should be given.

L276-281; how was this data normalized? Change to "chl-a and bacterial biomass".

L283; How many permutations were performed? Should be stated.

L285; I would suggest using bacterioplankton prior to this. Use already in introduction over bacteria.

L287; "… selection criteria)…". Remove ")".

L291; change "responses" to limitations?

**Results**

L294-312; This part is very descriptive, it would benefit from being shortened, to get to the more interesting findings of you paper.

L294; Here and elsewhere, when referring to figures, state which of the figures, a, b or c. you refer to. See general comment.

L296; change "meters" to m?

L310; change "an" to and.

L313-320; why not presenting DIP values by themselves, but only in DIN:DIP ratio?

L319; add 16:1 to Redfield ratio. (…Redfield ratio (16:1))

L321; change "greatly varied" to varied greatly?

L323-324; "cruise" is redundant.

L325; change "bacterial biomass" to BB, as you state this in L323.

L332; Information on MDS analysis is missing from statistics section. Please add information regarding this analysis.

L332-333; Please clarify. Suggestion "… relatively reduced variability within period".

L338; *Mamiellophyceae* is not included in the legend in. As they are the first once you mention, I would suggest including them in the figure 4.

L342; Explain what MALV refers to.

L343; Change to "Flavobacteriales and Rhodobacteriales…"

L343; The reference to fig 4b is incorrect. See general comment regarding labelling of figure and mosaics.

L345; See comment L338. Also, which cyanobacteria are you referring to?

L346; See comment L343.

L347; See comment L338, regarding Archaea.

L349; change "Mean" to Average?

L350; Here and elsewhere, provide t value.

L351; There is no fig 4c. See general comment regarding labelling of figure and mosaics.

L354; change "evolution" to development?

L356; "… in most …" Too general. Please specify the proportion at least.

L361-365; This section does not relate to response ratios (even if stated in L361). Please rephrase.

L362-363 & 367; Here and elsewhere, provide F value and df.

L367-369; Revise English.

L369; Here and elsewhere, provide F value and df.

L369-372; Revise English.

L373-375; Maybe state in which experiments this happens? Similar to L387-390.

L373-383; I would suggest restructuring for clarity. As now it is very difficult to understand when different responses occurred.

L377-378; Maybe state in which experiments this happens? Similar to L387-390.

L391-395; This part appears to belong in Material and Methods section.

L395; 4 sites? 2 stations and 2 depths? Please clarify.

L397-400; To me, these results are the most interesting. I would suggest restructuring the result, putting emphasis on the response ratios.

L405; "Most positive…". State proportion (%).

L418-422; This part appears to belong in Material and Methods section.

L422-423; What was Spearman Rho correlation with eukaryotic community composition.

L426; Where does the 78% originate from? State each dimensions contribution.

L430-431; State each dimensions contribution to the 59.4%.

L431-433; Revise English. Also, I'm struggling to see that the stations are actually separated.

L434; "… highly and positively correlated…". Revise English.

**Discussion**

L443-445; As you don't have measurements on B1, this statement is not fully true. Please tone down this statement.

L446; What expectations are you referring to? These should be stated more clearly before.

L448-452. What about predation pressure? Cellular demand of B vitamins? Actual cellular content of B vitamins? Should be expanded to include more potential explanations.

L452-454; In my opinion, this should have been done for all of the results. I understand that a tremendous amount of work has gone into this experiment, but I believe that the paper would benefit from a more succinct and concise result section.

L456 "… frequent but relatively moderate…". What does this mean, please clarify.

L461-464; What results are this statement based on?

L497-500; Highly speculative. Please rephrase to tone down this statement.

L521; change "potentially" to potential

L522-546; I would suggest reading Fridolfsson et al. 2018 and 2019, as well as Sylvander et al. 2013 to provide additional depth to the discussion on B1 and B12 amendments.

L563-566; Shouldn't dinoflagellates pop out in the analysis then?

L567; Why "strikingly"?

L570; change "revel" to reveal?

L576; Which "predation" are you referring to? Please clarify.

L582-583; What about uptake rates? I would suggest reading Koch et al. 2011, 2012, 2013 and discuss.

L588; "… B12 producers and B1 consumers." This is extremely generalized and implies that you can determine this in your paper. This is not fully true, especially for B1 as you don't have measurement for this B vitamin.

L590; "… cope with B vitamin shortage…". See L588. Once again, it is unfortunate, but you don't have measurements for B1 so your conclusions regarding this B vitamin should be toned down.

**Figure captions**
Please make sure that everything in your graphs can be identified. E.g fig 1, that cruises is illustrated by lines (in 1c legend), dots in fig 2, what 16:1 line refer to in fig 3.

Also, Generally, is there any benefit of using st3 and st6 instead of coastal and oceanic station. I feel the readability would increase if you used coastal and oceanic instead.

L937-941; Change "$\mu mol\ l^{-1}$" to $\mu M$? Pinpoint that axes are broken. Specify what SCM means.

L942-945; If so, state that it refers to t0. Also, what are the error bars showing?

L946-949; Suggestion, …(estimated as Chl-a concentration ($\mu g\ l^{-1}$)). Change "time-zero" to t0. Change "final-time" to endpoint. Pinpoint that axes are broken. Also, what are the error bars showing?

L950-952; Change "time-zero" to t0. Change "final-time" to endpoint. Pinpoint that axes are broken. Also, what are the error bars showing?

L953-960; I would suggest using more mosaics, a-d.

L961-968; change "… microbial plankton…" to microbial bakterioplankton, as it is only prokaryotes? Should be stated in the beginning and not at the end of the figure caption

**Figures**
**Figure 1**; Generally, is there any benefit of using st3 and st6 instead of coastal and oceanic station. I feel the readability would increase if you used coastal and oceanic instead.

**Figure 2**; When referring to figures, state which of the figures, a, b or c. you refer to. Also, look over all figures so that all are labelled a, b, c …. For fig. 2 and 3 I'm having troubles seeing the benefits of having several a, several b etc. I would like to see labelling a-r instead for fig. 2, then you can refer to the specific mosaic.

**Figure 3**; See comment for fig 2. For fig. 2 and 3 I'm having troubles seeing the benefits of having several a, several b etc. For the legend, the depth is stated as 0m and SCM, change to

"surface (5m) and SCM", as you did not sample 0m, correct? Also, state what the 16:1 line refers to. Also, I would suggest providing an average per station and cruise, and not all 3a, 3b and 3c etc, see general comments.

**Figure 4**; change mosaics to cover a-c, as stated in the main text. On the x-axes, the depth is stated as 0m and SCM, change to "surface (5m) and SCM", as you did not sample 0m, correct? You do not use a consistent taxonomy level, some are species whilst other groups are a combination. Could this affect your results? If not, I would still reconsider the different taxonomical levels presented.

**Figure 5 and Figure 6;** The colors are very difficult to distinguish. Also, I would suggest providing an average per station and cruise, and not all 3a, 3b and 3c etc, see general comments.

**Figure 7**; I would suggest changing the layout, to something used frequently when presenting fold change. You don't need to show 0, as every finding is around 1. See oversimplified suggestion below.

[Figure]

**Figure 8**; You do not use a consistent taxonomy level, some are species whilst other groups are a combination. Could this affect your results? If not, I would still reconsider the different taxonomical levels presented. The legend needs formatting prior to publication, much too large as it is now. The depth is stated as 0m and SCM, change to "surface (5m) and SCM", as you did not sample 0m, correct?

**Supplement information**
**Table S2**; This information is the same as in fig 3, correct? To me, this is redundant. If to be included, abbreviations in column names should be explained.

L18-27; "… experiments by the averaged…". Add divided? Change "that means" to which implies. Pinpoint that axes are broken

**Figure S1**; Shouldn't axes present statistics?, Percentages? The legend needs formatting prior to publication, much too large as it is now.

**Figure S2**; I propose including this graph over Fig 5 and 6. If included, it must be formatted to conform to the palette the authors have used, for clarity. How was these stats performed, as RR already considers the control. Clarify.

Figure S3; I would suggest changing the layout, to something used frequently when presenting fold change. You don't need to show 0, as every finding is around 1. See comment for figure 7. As it is now, it is impossible to get any valuable information from the figure.

**References**

Cruz-Lopez R, Maske H. (2016). The Vitamin $B_1$ and $B_{12}$ Required by the Marine Dinoflagellate *Lingulodinium polyedrum* Can be Provided by its Associated Bacterial Community in Culture. Front Microbiol. 7:560. doi: 10.3389/fmicb.2016.00560.

Fridolfsson E, Bunse C, Legrand C, Lindehoff E, Majaneva S, Hylander S. (2019). Seasonal variation and species-specific concentrations of the essential vitamin $B_1$ (thiamin) in zooplankton and seston. Mar Biol. 166(6):70. doi: 10.1007/s00227-019-3520-6.

Fridolfsson E, Lindehoff E, Legrand C, Hylander S. (2018). Thiamin (vitamin $B_1$) content in phytoplankton and zooplankton in the presence of filamentous cyanobacteria. Limnol Oceanogr. 63(6):2423-35. doi: 10.1002/lno.10949.

Koch F, Marcoval MA, Panzeca C, Bruland KW, Sañudo-Wilhelmy SA, Gobler CJ. (2011). The effect of vitamin $B_{12}$ on phytoplankton growth and community structure in the Gulf of Alaska. Limnol Oceanogr. 56(3):1023-34. doi: 10.4319/lo.2011.56.3.1023.

Koch F, Hattenrath-Lehmann TK, Goleski JA, Sañudo-Wilhelmy S, Fisher NS, Gobler CJ. (2012). Vitamin $B_1$ and $B_{12}$ uptake and cycling by plankton communities in coastal ecosystems. Front Microbiol. 3:363. doi: 10.3389/fmicb.2012.00363.

Koch F, Sañudo-Wilhelmy SA, Fisher NS, Gobler CJ. (2013). Effect of vitamins $B_1$ and $B_{12}$ on bloom dynamics of the harmful brown tide alga, *Aureococcus anophagefferens* (Pelagophyceae). Limnol Oceanogr. 58(5):1761-74. doi: 10.4319/lo.2013.58.5.1761.

Sylvander P, Häubner N, Snoeijs P. (2013). The thiamine content of phytoplankton cells is affected by abiotic stress and growth rate. Microb Ecol. 65(3):566-77. doi: 10.1007/s00248-012-0156-1.

---

## Referee Report (RR2)

Review v2 of "Spatial and temporal variability in the response of phytoplankton and bacterioplankton to B-vitamin amendments in an upwelling system" by Joglar et al.

**General comments**

The authors have put considerable effort into responding to all my previous concerns. The sampling campaign is definitely impressive, as well as the work that went in to the study and I think that the results and discussions makes this effort justice now.

I have some minor comments for the author, to further help with the readability and clarity of the manuscript. Some points are purely editorial whilst others needs to be answered and the text changed. I would also like to congratulate the authors on a job well done, both on the cruise, lab and writing a very interesting manuscript.

**Specific comments**

**Introduction**

L36-39; I feel the text would benefit from more precise examples, e.g. cyanobacterial blooms, red tides etc.

L71; change "drive" to thrive?

**Methods**

L213; change "inned" to inner.

L226; For clarity, add pmol l$^{-1}$ after 0.04.

L263; μm is in blue, change to black.

L282; For clarity, I would like that the non-normal variables are stated somewhere, either here or in supplementary material.

L288-289; Did you only compare differences between treatments and the control and not between all treatments? If so, why?

L289-292; I realize this might be due to different traditions, but for me non-metric multidimensional scaling is abbreviated as nMDS. It is no requirement to change, I simply wanted to raise the concern.

**Results**

L339; change "below of" to "below the".

L341-342; Does this statement relate to the average chl a levels, per month? If so it should be stated more clearly. If not, this does not seem to be the case in some days (a, b and c). Please look into this and change statement if needed or clarify.

L343; Add reference to figure 3d-f.

L357; "… sampling dates…", maybe change to cruise if applicable.

L360; "… but their abundance…" add "relative" for clarity.

L372; Add "." before Average…

L373; what does "gl=10" mean? If it is degrees of freedom, use df. In not you should still state df.

L380-381; "However, Chl-a mostly decreased in the coastal experiments conducted in August (Fig. 5a and Fig. 5c)." I do not agree with this statement, as this is not was is shown in the figures. For instance, all bars in a shade of blue is always higher for the than t0 for August samplings.

L446; Even if the eukaryotic community composition did not correlate significantly, you should still present the correlation coefficient and p value for this.

L450; Maybe remove underscore in "SAR11_clade"? If this is common practice, please ignore.

L458; Change to *Planktomarina*.

**Discussion**

L485; Change "bacteria" to prokaryotes.

L486; State which experiment situation you refer to.

L494-499; This sentence is too long (63 words), please restructure to give the reader a chance to follow.

L530-533 and 544; Change "cobalamin" to B12.

L602; "Flavobacteriia", is this correct?

L608; Which predation do you refer to? Zooplankton or mixotrophs? Please clarify.

**Figure captions**

L985-989; Add space between *shelf* and *(Oc)*. You do not have any ns in figure, can be removed?

L991-995. This figure caption is incorrect. Now you have more facets/mosaics, please update the caption accordingly.

L1005-1009. In the figure you have 5m and SCM, but in caption you have surface and SCM. I would suggest changing the figure x axes. Add information about SCM.

L1011-1015; In the figure you have surface and SCM, but in caption you have 5m and SCM. Please be consistent. Change "(c) SCM" to (d) SCM.

L1023-1030; Change "bars" to dots or points. Add information about errorbars.

L1032-1039; You don't have any "numbers" anymore. Can be removed from caption.

**Figures**

**Figure 8;** In the figure you have 5m and SCM, but in the manuscript you have surface and SCM. Please be consistent.

**Supplement information**

**Figure S1**; In the figure you have 5m and SCM, but in the manuscript you have surface and SCM. Please be consistent.

**Figure S3 + caption;** State that y axis is broken for a and b.

---

## Editor Decision (ED1)

Rreferee #1

While the authors have successfully addressed many of my specific comments, they have not addressed the global issue of the manuscript being vague and unspecific to the point where it is not clear that the data matches the conclusions the authors are making. It's very possible they do match, but the way the manuscript is written, it is not currently clear. The manuscript still suffers from major flaws that prevent it to be published in its current form. Most concerning of which is the lack of specificity that the authors use in their language. It is impossible for the reader to know exactly what results the authors are referring to.

For example: throughout the manuscript the authors use phytoplankton biomass and chlorophyll concentration interchangeably (e.g., ln 321, 355). The authors measured chlorophyll concentration, not biomass. It is strongly established that chlorophyll concentration to cellular carbon (biomass) ratios are highly variable in phytoplankton, especially across seasonal changes in light and nutrient concentration (which is the context for the authors experiments). It is not valid to say that phytoplankton biomass is being estimated by chlorophyll concentration. I do think that measuring chlorophyll concentration is a valid method of tracking phytoplankton, but the authors need to be specific throughout the manuscript about what they are actually measuring, and make sure that the conclusions they are making can be supported by their actual data. Reporting changes in chlorophyll concentration have a dramatically different physiological and ecological implications than reporting changes in biomass. I can't be sure what they mean with the manuscript in its current form.

In sum, this manuscript has a lot of potential. I was (and am) excited to see this experiment, and I do think the data should eventually be published. However, as it stands currently, the authors have not done their due diligence making sure that the manuscript is clear, specific, and ready for publication.

Referee #2

Review of "Spatial and temporal variability in the response of phytoplankton and bacterioplankton to B-vitamin amendments in an upwelling system" by Joglar et al.

**General comments**

This manuscript covers a very interesting and highly relevant topic, which the authors focuses on in the manuscript. Dynamics of B-vitamins in the worlds ocean is not often studied and this manuscript attempts to provide important information on this topic. The sampling campaign is definitely impressive, as well as the work that went in to the study. I don't agree with all the comments from the previous reviewers, for instance I find the use of response ratios very informative and a great display of the results. With that said, I have some comments and concerns with the manuscript that need to be addressed.

One main problem with the manuscript is that only one of the vitamins investigated is analyzed. I realize this may be due to problem quantifying B1 in natural sea water, which the authors can state more clearly. It would also have been very interesting to have quantified the cellular content in the two size fractions, but unfortunately this was not done.

Generally, is there any benefit of using st3 and st6 instead of coastal and oceanic station? I feel the readability would increase if you used coastal and oceanic instead.

When referring to figures, state which of the figures, a, b or c. you refer to. Also, look over all figures so that all are labelled a, b, c …. For fig. 2 and 3 I'm having troubles seeing the benefits of having several a's, several b's etc. I would like to see labelling a-r instead for fig. 2, then you can refer to the specific mosaic.

I would like to see a more accurate reporting of statistics. Please provide statistics value (t, F, df) when appropriate.

In my opinion, the results should be presented as averages, per station and cruise and ignore 3a, 3b, 3c etc. I understand that a tremendous amount of work has gone into this experiment, but I believe that the paper would benefit from a more succinct and concise result section.

**Specific comments**

**Abstract**
L15; "… unimportant, …" – I would suggest changing wording, as you cannot know if it is unimportant or not. Maybe "slight" or "limited"?

L15; how can an "unimportant" variability lead to the assumption that there are factors operating at other scales? Requires clarification.

L20; change "alone" to solely?

L22-24; auxotrophy is also high in phytoplankton, causing the argument to halter a bit. I would suggest mentioning this as well and combine it with bacteria dependence.

**Introduction**
L34; state which toxic episodes you refer to.

L60-61; I would suggest reading Cruz-Lopez et al. 2016.

L69-74; I would suggest reviewing if you really need all references to say what you want to say. In a relatively short sentence, you use 13 references.

L100 & L110; decide if you use numbers or text, 36 or thirty-six, and use throughout.

L105; change "synthetize" to synthesize.

**Methods**
L119; What is the timeframe between a, b and c? Looking at fig 1 I realized I can figure it out, but it is a very unclear way to present samplings.

L120-123; To increase clarity, I would recommend to state that surface is 5m deep more clearly.

L123; State which occasion this sampling failed.

L128-129; do you refer to the t0 for each experiment (a, b and c)? Needs clarification.

L129-132; Does the UI provide you with important information?, now sentence feels a bit dropped in the text.

L133-134; What about small zooplankton, copepodites and nauplii? Did you check for this, if so it should be stated. If not, the potential impact of these should be taken into account for.

L138-144; This is a very confusing way to present the treatments. I would suggest providing all of this important information in a table instead. Additionally, the rationale behind the levels of nutrients and vitamins should be given.

L147-150; This is unclear to read. First it is natural conditions, then the conditions were reproduced? How was this done? What screens are you referring to?

L152; change to "t0"?

L160-162; Revise sentence. Suggestion "Samples were incubated 20 min for the fixative to act on cells, immersed in liquid nitrogen for 15 min before being frozen at -80ºC."

L169-170; Could the usage of two different factors cause a problem in the interpretation of the data, when comparing coast and oceanic station?

L173; "… first place…" before all other variables? If so, please clarify.

L174-177; Revise sentence. Suggestion "Polyethylene bottles (50 ml, pre cleaned with 5% HCl were filled with the sample using contamination-free plastic gloves and immediately frozen at –20°C until analysis, using standard colorimetric methods with a Bran-Luebbe segmented flow analyzer (Hansen and Grasshoff 1983)." Or did I misunderstood "free-contamination"?

L182; Unfortunately, you only have samples for dissolved B12. This should be specified.

L183; Specify when the fifth or sixth day was sampled, as it can influence the results.

L188; Do you refer to leftover water? If so, change wording. If not, clarify.

L199-200; State which values apply for length, inner diameter and particle size of column.

L211; You have not used subscript before, change to B12.

L211-212; State which congener had which LOD.

L212; If the case, state that 0.05 is for cyanocobalamin, CNB12. Also, change to hydroxocobalamin.

L214; You have not stated what CNB12 is.

L219; Why was plankton community sampled day 1, 2, 4, 6, while B12 was sampled day 1, 3, 5/6?

L222; Change "litters" to liters.

L237-238; Can you update with the accession numbers?

L245-247; How can this be? Is it fragments of cells going through the 3 µm filter? Would benefit from an explanation for this.

L251-252; Please provide the rationale for this procedure.

L265; "… if necessary to attain normality". Was this not always the case, do you have some samplings where the data was not normalized and some where it is? If so, you should state when this was the case and discuss how this might affect the results and conclusions drawn from them.

L266; When "standardizing", do you refer to using the corrected p value?

L267-273; Why using ANOVA and Z-test? The reasoning behind this choice should be given.

L276-281; how was this data normalized? Change to "chl-a and bacterial biomass".

L283; How many permutations were performed? Should be stated.

L285; I would suggest using bacterioplankton prior to this. Use already in introduction over bacteria.

L287; "… selection criteria)…". Remove ")".

L291; change "responses" to limitations?

**Results**

L294-312; This part is very descriptive, it would benefit from being shortened, to get to the more interesting findings of you paper.

L294; Here and elsewhere, when referring to figures, state which of the figures, a, b or c. you refer to. See general comment.

L296; change "meters" to m?

L310; change "an" to and.

L313-320; why not presenting DIP values by themselves, but only in DIN:DIP ratio?

L319; add 16:1 to Redfield ratio. (…Redfield ratio (16:1))

L321; change "greatly varied" to varied greatly?

L323-324; "cruise" is redundant.

L325; change "bacterial biomass" to BB, as you state this in L323.

L332; Information on MDS analysis is missing from statistics section. Please add information regarding this analysis.

L332-333; Please clarify. Suggestion "… relatively reduced variability within period".

L338; *Mamiellophyceae* is not included in the legend in. As they are the first once you mention, I would suggest including them in the figure 4.

L342; Explain what MALV refers to.

L343; Change to "Flavobacteriales and Rhodobacteriales…"

L343; The reference to fig 4b is incorrect. See general comment regarding labelling of figure and mosaics.

L345; See comment L338. Also, which cyanobacteria are you referring to?

L346; See comment L343.

L347; See comment L338, regarding Archaea.

L349; change "Mean" to Average?

L350; Here and elsewhere, provide t value.

L351; There is no fig 4c. See general comment regarding labelling of figure and mosaics.

L354; change "evolution" to development?

L356; "… in most …" Too general. Please specify the proportion at least.

L361-365; This section does not relate to response ratios (even if stated in L361). Please rephrase.

L362-363 & 367; Here and elsewhere, provide F value and df.

L367-369; Revise English.

L369; Here and elsewhere, provide F value and df.

L369-372; Revise English.

L373-375; Maybe state in which experiments this happens? Similar to L387-390.

L373-383; I would suggest restructuring for clarity. As now it is very difficult to understand when different responses occurred.

L377-378; Maybe state in which experiments this happens? Similar to L387-390.

L391-395; This part appears to belong in Material and Methods section.

L395; 4 sites? 2 stations and 2 depths? Please clarify.

L397-400; To me, these results are the most interesting. I would suggest restructuring the result, putting emphasis on the response ratios.

L405; "Most positive…". State proportion (%).

L418-422; This part appears to belong in Material and Methods section.

L422-423; What was Spearman Rho correlation with eukaryotic community composition.

L426; Where does the 78% originate from? State each dimensions contribution.

L430-431; State each dimensions contribution to the 59.4%.

L431-433; Revise English. Also, I'm struggling to see that the stations are actually separated.

L434; "… highly and positively correlated…". Revise English.

**Discussion**
L443-445; As you don't have measurements on B1, this statement is not fully true. Please tone down this statement.

L446; What expectations are you referring to? These should be stated more clearly before.

L448-452. What about predation pressure? Cellular demand of B vitamins? Actual cellular content of B vitamins? Should be expanded to include more potential explanations.

L452-454; In my opinion, this should have been done for all of the results. I understand that a tremendous amount of work has gone into this experiment, but I believe that the paper would benefit from a more succinct and concise result section.

L456 "… frequent but relatively moderate…". What does this mean, please clarify.

L461-464; What results are this statement based on?

L497-500; Highly speculative. Please rephrase to tone down this statement.

L521; change "potentially" to potential

L522-546; I would suggest reading Fridolfsson et al. 2018 and 2019, as well as Sylvander et al. 2013 to provide additional depth to the discussion on B1 and B12 amendments.

L563-566; Shouldn't dinoflagellates pop out in the analysis then?

L567; Why "strikingly"?

L570; change "revel" to reveal?

L576; Which "predation" are you referring to? Please clarify.

L582-583; What about uptake rates? I would suggest reading Koch et al. 2011, 2012, 2013 and discuss.

L588; "… B12 producers and B1 consumers." This is extremely generalized and implies that you can determine this in your paper. This is not fully true, especially for B1 as you don't have measurement for this B vitamin.

L590; "… cope with B vitamin shortage…". See L588. Once again, it is unfortunate, but you don't have measurements for B1 so your conclusions regarding this B vitamin should be toned down.

**Figure captions**
Please make sure that everything in your graphs can be identified. E.g fig 1, that cruises is illustrated by lines (in 1c legend), dots in fig 2, what 16:1 line refer to in fig 3.

Also, Generally, is there any benefit of using st3 and st6 instead of coastal and oceanic station. I feel the readability would increase if you used coastal and oceanic instead.

L937-941; Change "µmol l$^{-1}$" to µM? Pinpoint that axes are broken. Specify what SCM means.

L942-945; If so, state that it refers to t0. Also, what are the error bars showing?

L946-949; Suggestion, …(estimated as Chl-a concentration (µg l$^{-1}$)). Change "time-zero" to t0. Change "final-time" to endpoint. Pinpoint that axes are broken. Also, what are the error bars showing?

L950-952; Change "time-zero" to t0. Change "final-time" to endpoint. Pinpoint that axes are broken. Also, what are the error bars showing?

L953-960; I would suggest using more mosaics, a-d.

L961-968; change "… microbial plankton…" to microbial bakterioplankton, as it is only prokaryotes? Should be stated in the beginning and not at the end of the figure caption

**Figures**
**Figure 1**; Generally, is there any benefit of using st3 and st6 instead of coastal and oceanic station. I feel the readability would increase if you used coastal and oceanic instead.

**Figure 2**; When referring to figures, state which of the figures, a, b or c. you refer to. Also, look over all figures so that all are labelled a, b, c …. For fig. 2 and 3 I'm having troubles seeing the benefits of having several a, several b etc. I would like to see labelling a-r instead for fig. 2, then you can refer to the specific mosaic.

**Figure 3**; See comment for fig 2. For fig. 2 and 3 I'm having troubles seeing the benefits of having several a, several b etc. For the legend, the depth is stated as 0m and SCM, change to

"surface (5m) and SCM", as you did not sample 0m, correct? Also, state what the 16:1 line refers to. Also, I would suggest providing an average per station and cruise, and not all 3a, 3b and 3c etc, see general comments.

**Figure 4**; change mosaics to cover a-c, as stated in the main text. On the x-axes, the depth is stated as 0m and SCM, change to "surface (5m) and SCM", as you did not sample 0m, correct? You do not use a consistent taxonomy level, some are species whilst other groups are a combination. Could this affect your results? If not, I would still reconsider the different taxonomical levels presented.

**Figure 5 and Figure 6;** The colors are very difficult to distinguish. Also, I would suggest providing an average per station and cruise, and not all 3a, 3b and 3c etc, see general comments.

**Figure 7**; I would suggest changing the layout, to something used frequently when presenting fold change. You don't need to show 0, as every finding is around 1. See oversimplified suggestion below.

[Figure]

**Figure 8**; You do not use a consistent taxonomy level, some are species whilst other groups are a combination. Could this affect your results? If not, I would still reconsider the different taxonomical levels presented. The legend needs formatting prior to publication, much too large as it is now. The depth is stated as 0m and SCM, change to "surface (5m) and SCM", as you did not sample 0m, correct?

**Supplement information**
**Table S2**; This information is the same as in fig 3, correct? To me, this is redundant. If to be included, abbreviations in column names should be explained.

L18-27; "… experiments by the averaged…". Add divided? Change "that means" to which implies. Pinpoint that axes are broken

**Figure S1**; Shouldn't axes present statistics?, Percentages? The legend needs formatting prior to publication, much too large as it is now.

**Figure S2**; I propose including this graph over Fig 5 and 6. If included, it must be formatted to conform to the palette the authors have used, for clarity. How was these stats performed, as RR already considers the control. Clarify.

Figure S3; I would suggest changing the layout, to something used frequently when presenting fold change. You don't need to show 0, as every finding is around 1. See comment for figure 7. As it is now, it is impossible to get any valuable information from the figure.

**References**

Cruz-Lopez R, Maske H. (2016). The Vitamin $B_1$ and $B_{12}$ Required by the Marine Dinoflagellate *Lingulodinium polyedrum* Can be Provided by its Associated Bacterial Community in Culture. Front Microbiol. 7:560. doi: 10.3389/fmicb.2016.00560.

Fridolfsson E, Bunse C, Legrand C, Lindehoff E, Majaneva S, Hylander S. (2019). Seasonal variation and species-specific concentrations of the essential vitamin $B_1$ (thiamin) in zooplankton and seston. Mar Biol. 166(6):70. doi: 10.1007/s00227-019-3520-6.

Fridolfsson E, Lindehoff E, Legrand C, Hylander S. (2018). Thiamin (vitamin $B_1$) content in phytoplankton and zooplankton in the presence of filamentous cyanobacteria. Limnol Oceanogr. 63(6):2423-35. doi: 10.1002/lno.10949.

Koch F, Marcoval MA, Panzeca C, Bruland KW, Sañudo-Wilhelmy SA, Gobler CJ. (2011). The effect of vitamin $B_{12}$ on phytoplankton growth and community structure in the Gulf of Alaska. Limnol Oceanogr. 56(3):1023-34. doi: 10.4319/lo.2011.56.3.1023.

Koch F, Hattenrath-Lehmann TK, Goleski JA, Sañudo-Wilhelmy S, Fisher NS, Gobler CJ. (2012). Vitamin $B_1$ and $B_{12}$ uptake and cycling by plankton communities in coastal ecosystems. Front Microbiol. 3:363. doi: 10.3389/fmicb.2012.00363.

Koch F, Sañudo-Wilhelmy SA, Fisher NS, Gobler CJ. (2013). Effect of vitamins $B_1$ and $B_{12}$ on bloom dynamics of the harmful brown tide alga, *Aureococcus anophagefferens* (Pelagophyceae). Limnol Oceanogr. 58(5):1761-74. doi: 10.4319/lo.2013.58.5.1761.

Sylvander P, Häubner N, Snoeijs P. (2013). The thiamine content of phytoplankton cells is affected by abiotic stress and growth rate. Microb Ecol. 65(3):566-77. doi: 10.1007/s00248-012-0156-1.

---

## Author Response (AR2)

Vanessa Joglar

Biological Oceanography Group

Koji Suzuki

Associate Editor

Biogeosciences

Vigo, 27th February 2020

Dear Koji

Please find attached a new revised version of manuscript entitled "Spatial and temporal variability in the response of phytoplankton and bacterioplankton to B-vitamin amendments in an upwelling system". The manuscript was co-authored by myself, Antero Prieto, Esther Barber-Lluch, Marta Hernández-Ruiz, Emilio Fernández and Eva Teira.

We would like to appreciate the extensive and constructive comments of the anonymous referees, which clearly helped us to improve the overall quality and understanding of the manuscript. We have considered all the issues raised by the reviewers, and a detailed response to all comments is attached. In the individual responses to referee comments, the suggestions and comments of the reviewers are in plain font and our responses are in italic and blue font. The revised version of the manuscript with marked changes is also provided. The major changes are summarized below:

- The language has been revised and improved.
- A table (Table 1) has been included to facilitate the understanding of the experimental treatments.

- The quality of the figures has been improved by changing the layout and/or the colours when appropriate (e.g. figure 5, figure 6, figure 7, figure S3 and figure S4).

- Figure S3 has been modified and replaced by new Fig. S3 and Fig. S4.

- As one of the reviewers required, we have carefully revised the manuscript, including the text and figures, clearly indicating that the response variable is chlorophyll-a, not phytoplankton biomass.

- We have reviewed the manuscript replacing bacterial biomass to prokaryote biomass, as archaea were also included in the cytometer counts.

- The station numbers have been replaced by coastal and oceanic station.

- We reviewed the discussion eliminating speculative statements and toning down some of our conclusions.

Looking forward to hearing from you,

Vanessa Joglar

Full address for corresponding is:

Grupo de Oceanografía Biológica

Departamento de Ecología y Biología Animal

Universidad de Vigo

Campus Universitario Lagoas-Marcosende

36310-Vigo Spain

E-mail: vjoglar@uvigo.es

Referee #1

While the authors have successfully addressed many of my specific comments, they have not addressed the global issue of the manuscript being vague and unspecific to the point where it is not clear that the data matches the conclusions the authors are making. It's very possible they do match, but the way the manuscript is written, it is not currently clear. The manuscript still suffers from major flaws that prevent it to be published in its current form. Most concerning of which is the lack of specificity that the authors use in their language. It is impossible for the reader to know exactly what results the authors are referring to.

For example: throughout the manuscript the authors use phytoplankton biomass and chlorophyll concentration interchangeably (e.g., ln 321, 355). The authors measured chlorophyll concentration, not biomass. It is strongly established that chlorophyll concentration to cellular carbon (biomass) ratios are highly variable in phytoplankton, especially across seasonal changes in light and nutrient concentration (which is the context for the authors experiments). It is not valid to say that phytoplankton biomass is being estimated by chlorophyll concentration. I do think that measuring chlorophyll concentration is a valid method of tracking phytoplankton, but the authors need to be specific throughout the manuscript about what they are actually measuring, and make sure that the conclusions they are making can be supported by their actual data. Reporting changes in chlorophyll concentration have a dramatically different physiological and ecological implications than reporting changes in biomass. I can't be sure what they mean with the manuscript in its current form.

In sum, this manuscript has a lot of potential. I was (and am) excited to see this experiment, and I do think the data should eventually be published. However, as it stands currently, the authors have not done their due diligence making sure that the manuscript is clear, specific, and ready for publication.

*We appreciate the overall positive comments of the reviewer. We believe that this revised version is now more clear and focused. We are aware that chlorophyll-a can be only used as an estimator of phytoplankton biomass. We had clearly indicated in the methodological section that we use chlorophyll-a concentration as a proxy for phytoplankton biomass (line 176 in the*

*former resubmitted version). This pigment is universally used in phytoplankton studies and its use as a phytoplankton biomass estimator is common among the scientific community. Moreover, most experimental studies evaluating responses of phytoplankton to nutrient additions used chlorophyll-a concentration as response variable (e.g., Bertrand et al., 2015; Browning et al., 2017; Caron et al., 2000; Martínez-García et al., 2010, Hernández-Ruiz et al 2020). We are also aware that C:Chla ratio varies with light and nutrients. Regarding light, all the treatments were incubated under the same light conditions (simulating the corresponding in situ irradiance). On the other hand, in previous experiments, using the same inorganic nutrient additions in the same sampling area we found a very good linear relationship with slope ca. 1, between the C:Chla ratio in the control and the corresponding ratio in the inorganic treatment (see plot below built with data from Martínez-García et al 2010b and Teira et al 2011), suggesting that at this short time scale, nutrient levels are not significantly affecting C:Chla ratio. As determining phytoplankton carbon biomass is extremely time-consuming, we decided to use only chlorophyll-a to evaluate the response of phytoplankton in this extensive study. Nevertheless, for clarity we have carefully revised the manuscript, including the text and figures, clearly indicating that the response variable is chlorophyll-a, not phytoplankton biomass.*

[Figure]

References

Bertrand, E. M., McCrow, J. P., Moustafa, A., Zheng, H., McQuaid, J. B., Delmont, T. O.,

Post, A. F., Sipler, R. E., Spackeen, J. L., Xu, K., Bronk, D. A., Hutchins, D. A., Allen, A. E. and Karl, D. M. (2015) Phytoplankton-bacterial interactions mediate micronutrient colimitation at the coastal Antarctic sea ice edge, Proc. Natl. Acad. Sci. U. S. A., 112(32), 9938–9943.

Browning, T. J., Achterberg, E. P., Rapp, I., Engel, A., Bertrand, E. M., Tagliabue, A. and Moore, C. M. (2017) Nutrient co-limitation at the boundary of an oceanic gyre, Nature, 551(7679), 242–246.

Caron, D. A., Lim, E. L., Sanders, R. W., Dennett, M. R. and Berninger, U. G. (2000) Responses of bacterioplankton and phytoplankton to organic carbon and inorganic nutrient additions in contrasting oceanic ecosystems, Aquat. Microb. Ecol., 22(2), 175–184.

Hernández-Ruiz, M., Barber-Lluch, E., Prieto, A., Logares, R., & Teira, E. (2020). Response of pico-nano-eukaryotes to inorganic and organic nutrient additions. Estuarine, Coastal and Shelf Science, 235, 106565.

Martínez-García, S., Fernández, E., Calvo-Díaz, A., Maraňn, E., Morán, X. A. G. and Teira, E.(2010a) Response of heterotrophic and autotrophic microbial plankton to inorganic and organic inputs along a latitudinal transect in the Atlantic Ocean, Biogeosciences, 7(5), 1701–1713.

Martínez-García, S., Fernández, E., Álvarez-Salgado, X. A., González, J., Lønborg, C., Marañón, E., ... & Teira, E. (2010b). Differential responses of phytoplankton and heterotrophic bacteria to organic and inorganic nutrient additions in coastal waters off the NW Iberian Peninsula. Marine Ecology Progress Series, 416, 17-33.

Panzeca, C., Beck, A. J., Tovar-Sanchez, A., Segovia-Zavala, J., Taylor, G. T., Gobler, C. J. and Sañudo-Wilhelmy, S. A. (2009) Distributions of dissolved vitamin B12 and Co in coastal and open-ocean environments, Estuar. Coast. Shelf Sci., 85(2), 223–230.

Sañudo-Wilhelmy, S. A., Gobler, C. J., Okbamichael, M. and Taylor, G. T. (20006) Regulation of phytoplankton dynamics by vitamin B12, Geophys. Res. Lett., 33(4), 10–13.

Teira, E., Martínez-García, S., Carreira, C., & Morán, X. A. G. (2011). Changes in bacterioplankton and phytoplankton community composition in response to nutrient additions in coastal waters off the NW Iberian Peninsula. Marine Ecology Progress Series, 426, 87-104.

Referee #2

Review of "Spatial and temporal variability in the response of phytoplankton and bacterioplankton to B-vitamin amendments in an upwelling system" by Joglar et al.

**General comments**

This manuscript covers a very interesting and highly relevant topic, which the authors focuses on in the manuscript. Dynamics of B-vitamins in the worlds ocean is not often studied and this manuscript attempts to provide important information on this topic. The sampling campaign is definitely impressive, as well as the work that went in to the study. I don't agree with all the comments from the previous reviewers, for instance I find the use of response ratios very informative and a great display of the results. With that said, I have some comments and concerns with the manuscript that need to be addressed.

One main problem with the manuscript is that only one of the vitamins investigated is analyzed. I realize this may be due to problem quantifying B1 in natural sea water, which the authors can state more clearly. It would also have been very interesting to have quantified the cellular content in the two size fractions, but unfortunately this was not done.

Generally, is there any benefit of using st3 and st6 instead of coastal and oceanic station? I feel the readability would increase if you used coastal and oceanic instead.

When referring to figures, state which of the figures, a, b or c. you refer to. Also, look over all figures so that all are labelled a, b, c …. For fig. 2 and 3 I'm having troubles seeing the benefits of having several a's, several b's etc. I would like to see labelling a-r instead for fig. 2, then you can refer to the specific mosaic.

I would like to see a more accurate reporting of statistics. Please provide statistics value (t, F, df) when appropriate.

In my opinion, the results should be presented as averages, per station and cruise and ignore 3a, 3b, 3c etc. I understand that a tremendous amount of work has gone into this experiment, but I believe that the paper would benefit from a more succinct and concise result section.

*We very much appreciate the extraordinarily constructive and extensive review made by the referee, which undoubtedly and substantially improved the manuscript. We now clearly state that*

*we were unable to measure dissolved B1 concentration due to the very low concentration in the water and the reduced pre-concentration volume (ca 1 L) (Lines 245-246). Even though we filtered 2 L of seawater sample for B-vitamin determinations, we could only pre-concentrate 1 L in the C18 columns, as the columns became systematically clogged. We have changed the denomination of the two sampling sites and now we use coast and ocean. We have carefully revised the format of figures including the lettering to refer to the different plots. We also provide now the t or F value and the df when appropriate.*

*We agree that the big picture emerge when averaging the three experiments in each sampling site, and for this reason we only represent the average response ratio in figure 7. We maintain the raw data represented in figure 5 and 6 as other reviewers required to include those figure in the manuscript. Nevertheless, the results description and discussion is mostly based on averaged responses.*

**Specific comments**

**Abstract**

L15; "… unimportant, …" – I would suggest changing wording, as you cannot know if it is unimportant or not. Maybe "slight" or "limited"?

*This has been corrected (L15)*

L15; how can an "unimportant" variability lead to the assumption that there are factors operating at other scales? Requires clarification.

*We have clarified this sentence. We conclude that the availability of B-vitamins might be, in part, controlled by seasonal processes given that inter-season variability was larger than inter-day variability (L15-17)*

L20; change "alone" to solely?

*This has been changed (L21)*

L22-24; auxotrophy is also high in phytoplankton, causing the argument to halter a bit. I would suggest mentioning this as well and combine it with bacteria dependence.

*Phytoplankton B1 auxotrophy has been considered in the abstract (L25-28)*

**Introduction**

L34; state which toxic episodes you refer to.

*This has been clarified (L39-40)*

L60-61; I would suggest reading Cruz-Lopez et al. 2016.

*This reference has been included (L67)*

L69-74; I would suggest reviewing if you really need all references to say what you want to say. In a relatively short sentence, you use 13 references.

*Several references have been removed (L77-80)*

L100 & L110; decide if you use numbers or text, 36 or thirty-six, and use throughout.

*This has been corrected (L106)*

L105; change "synthetize" to synthesize.

*This has been changed (L111)*

**Methods**

L119; What is the timeframe between a, b and c? Looking at fig 1 I realized I can figure it out, but it is a very unclear way to present samplings.

*The timeframe between experiments (a, b and c) was 2 or 3 days. This can be observed in figure 1b, 1c, and figure 2 (black points represent the initial time of each experiment). This information is also now included in the text (L126-129)*

L120-123; To increase clarity, I would recommend to state that surface is 5m deep more clearly.

*This has been clarified (L130-133)*

L123; State which occasion this sampling failed.

*This has been indicated (L134)*

L128-129; do you refer to the t0 for each experiment (a, b and c)? Needs clarification.

*Yes, we refer to t0. This has been clarified (L140).*

L129-132; Does the UI provide you with important information?, now sentence feels a bit dropped in the text.

*UI indicates the upwelling intensity helping to understand the initial hydrographic conditions. We have also included now the source of precipitation data (L144-146). We agree that this information did not fit very well in the experimental design section. We have renamed this section as "Sampling strategy".*

L133-134; What about small zooplankton, copepodites and nauplii? Did you check for this, if so it should be stated. If not, the potential impact of these should be taken into account for.

*Seawater samples were pre-filtered through a 200 μm mesh to exclude large zooplankton. 18S sequence data revealed no presence of these small zooplankton groups.*

L138-144; This is a very confusing way to present the treatments. I would suggest providing all of this important information in a table instead. Additionally, the rationale behind the levels of nutrients and vitamins should be given.

*A table (Table 1) has been included to facilitate the understanding of the treatments administered. So, the description in the text has been summarized (L154-160). The rationale behind the levels of nutrient and vitamins added is also provided in the revised versions (L160-167).*

L147-150; This is unclear to read. First it is natural conditions, then the conditions were reproduced? How was this done? What screens are you referring to?

*Incubation was performed on deck, and therefore under natural solar radiation. The incubation bags were submerged in incubation tanks filled with constantly circulating surface seawater to maintain a temperature similar to that of the surface mixed layer. The tanks where the SCM samples were incubated were covered with several layers of a neutral mesh to attenuate the incident light and simulate the irradiance at the corresponding SCM. We used radiometers to determine the number of layers needed to attenuate incident light. We have revised this paragraphs for clarity (L169-173)*

L152; change to "t0"?

*This has been changed (L175).*

L160-162; Revise sentence. Suggestion "Samples were incubated 20 min for the fixative to act on cells, immersed in liquid nitrogen for 15 min before being frozen at -80ºC."

*This has been corrected (L184-185).*

L169-170; Could the usage of two different factors cause a problem in the interpretation of the data, when comparing coast and oceanic station?

*Given the small biovolume of prokaryotes, the difference between the two conversion factors would be <20%. We decided using two different factors as there were clear differences in the prokaryotic community composition between the coastal and oceanic station.*

L173; "… first place…" before all other variables? If so, please clarify.

*This has been clarified (L198)*

L174-177; Revise sentence. Suggestion "Polyethylene bottles (50 ml, pre cleaned with 5% HCl were filled with the sample using contamination-free plastic gloves and immediately frozen at –20°C until analysis, using standard colorimetric methods with a Bran-Luebbe segmented flow analyzer (Hansen and Grasshoff 1983)." Or did I misunderstood "free-contamination"?

*This has been clarified (L199-201).*

L182; Unfortunately, you only have samples for dissolved B12. This should be specified.

*This has been specified (L245-246).*

L183; Specify when the fifth or sixth day was sampled, as it can influence the results.

*This has been specified (L208-210).*

L188; Do you refer to leftover water? If so, change wording. If not, clarify.
*This has been clarified (L215-217)*

L199-200; State which values apply for length, inner diameter and particle size of column.

*This has been indicated in the text (L227-228).*

L211; You have not used subscript before, change to B12.

*This has been removed (L240).*
L211-212; State which congener had which LOD.
*LODs have been described correctly (L239-241).*

L212; If the case, state that 0.05 is for cyanocobalamin, CNB12. Also, change to hydroxocobalamin.

*This has been corrected (L241-242).*

L214; You have not stated what CNB12 is.

*This has been described (L241)*

L219; Why was plankton community sampled day 1, 2, 4, 6, while B12 was sampled day 1, 3, 5/6?

*We have corrected this as there was a confusion with the denomination of the sampling days. We now use the same nomenclature throughout the manuscript, that is, the first day of the cruise was denominated day 0, and so on. During the ENVISION cruises, due to the large*

*work load, the sampling for plankton community at the coastal and oceanic stations was done only at day 0, 1, 3, and 5 during the three cruises. For the coastal station, samples for B vitamin were also taken at day 1, 3, and 5, while the oceanic station was only sampled for vitamins only at the t0 of the experiments (day 1, 3 and 6). The rationale behind was that we concentrated B vitamin sampling efforts in the coastal station, sampling every second day 5 depths. By contrast the oceanic station was only sampled at those depths and days coinciding with the t0 of an experiment.*

L222; Change "litters" to liters.

*This has been corrected (L253)*

L237-238; Can you update with the accession numbers?

*The accession numbers have been updated (L268-269).*

L245-247; How can this be? Is it fragments of cells going through the 3 μm filter? Would benefit from an explanation for this.

*The filter used to separate size fractions had a diameter of 3 μm. There are numerous organisms which cell size range include the 3 μm (for example a given specie may have cells ranging in size from 2-4 μm), thus during filtration some cells will be retained in the 3 μm filter, and some will pass through. In addition, depending on their morphology, some cells may cross the 3 μm filter and others cannot. For example, cylindrical cells will pass through the filter depending on their position. A short explanation has been included (L277)*

L251-252; Please provide the rationale for this procedure.

*When the centered log ratio (clr) transformation is applied to relative abundance data. Zero values must be replaced (Aitchison, 1982). We replaced zeros by the minimum value that is larger than 0 divided by 2 as it is a common practice to replace zeros with a number less (e.g. 50%) than the detection limit (Martín-Fernández et al., 2003). This has been clarified (L282-284).*

L265; "… if necessary to attain normality". Was this not always the case, do you have some samplings where the data was not normalized and some where it is? If so, you should state when this was the case and discuss how this might affect the results and conclusions drawn from them.

*We transformed all datasets that did not comply with normality. This has been clarified*

*(L297-298).*

L266; When "standardizing", do you refer to using the corrected p value?

*Due to the small number of samples, p value was corrected as recommended by Good, 1982 (L299-300). This is*

$$p\ value \times \sqrt{(\frac{N}{100})}$$

*being*
*N: number of observations (samples)*

L267-273; Why using ANOVA and Z-test? The reasoning behind this choice should be given.

*ANOVA was used to assess differences in the response ratios to nutrient or vitamin additions between stations, seasons, and depths. As we found a significant effect of the three factors, and given that the CV (coefficient of variation) between replicate experiments (i.e. a, b, c) was relatively low, we decided to average the replicate experiments for each station, sampling depth and season. This average RR are represented in figure 7.*

*The significance of these averaged response ratios (represented in Fig. 7) was evaluated by comparing the averaged value with "1" using a Z-test. Thus, the Z-test was only used to test if averaged RRs were significantly different form "1". This has been clarified (L311-313).*

L276-281; how was this data normalized? Change to "chl-a and bacterial biomass".
*To normalise a variable, we subtracted its mean and divided by its standard deviation. A normalised variable has a mean of zero, a standard deviation of 1 and (therefore) a variance of 1.*

*The suggested changes were made (L318). Also bacterial biomass has been changed throughout the manuscript to prokaryote biomass, to account for the presence of archaea, that are also included in the flow cytometer counts.*

L283; How many permutations were performed? Should be stated.
*999 permutations were performed (L324)*

L285; I would suggest using bacterioplankton prior to this. Use already in introduction over bacteria.

*This has been changed were appropriate. For our results we now refer to prokaryotes.*
*(L61, L66).*

L287; "… selection criteria)…". Remove ")".

*This has been corrected (L328)*

L291; change "responses" to limitations?

*This fragment has been removed as it referred to an analysis that we finally did not include (L329-333).*

**Results**

L294-312; This part is very descriptive, it would benefit from being shortened, to get to the more interesting findings of you paper.

*This has been shortened (L351-353)*

L294; Here and elsewhere, when referring to figures, state which of the figures, a, b or c. you refer to. See general comment.

*This has been corrected.*

L296; change meters" to m?

*This has been changed (L339)*

L310; change "an" to and.

*This has been changed (L354)*

L313-320; why not presenting DIP values by themselves, but only in DIN:DIP ratio? L319; add 16:1 to Redfield ratio. (…Redfield ratio (16:1))

*16:1 has been added (L363)*

*DIP values are presented in Table S2.*

L321; change "greatly varied" to varied greatly?

*This has been corrected (L365)*

L323-324; "cruise" is redundant.

*This has been corrected (L368)*

L325; change "bacterial biomass" to BB, as you state this in L323.

*This has been corrected, see also response to comment L285 (L367, 370, 371)*

L332; Information on MDS analysis is missing from statistics section. Please add information regarding this analysis.

*MDS analysis has been included (L305-311)*

L332-333; Please clarify. Suggestion "… relatively reduced variability within period".

*This has been corrected (L378)*

L338; *Mamiellophyceae* is not included in the legend in. As they are the first once you mention, I would suggest including them in the figure 4.

*Mamiellophyceae is a class included in the phylum Chlorophyta, dominated in our samples by two genera: Ostreococcus and Micromonas. We now indicate that we refer to these two genera represented in figure 4 (L384).*

L342; Explain what MALV refers to.

*This has been clarified (L388)*

L343; Change to "Flavobacteriales and Rhodobacteriales…"

*"Flavobacteriales" has been corrected (L389 & 392) however, Rhodobacterales is the correct name.*

L343; The reference to fig 4b is incorrect. See general comment regarding labelling of figure and mosaics.

*The labelling has been corrected throughout the manuscript.*

L345; See comment L338. Also, which cyanobacteria are you referring to?

*This has been clarified (L393)*

L346; See comment L343.

*The labelling has been corrected throughout the manuscript.*

L347; See comment L338, regarding Archaea.

*This has been clarified in the text (L394)*

L349; change "Mean" to Average?

*This has been changed (L397)*

L350; Here and elsewhere, provide t value.

*The t and F values, as well as df have been added (L398-399).*

L351; There is no fig 4c. See general comment regarding labelling of figure and mosaics.

*The labelling has been corrected throughout the manuscript.*

L354; change "evolution" to development?

*This has been changed (L403)*

L356; "… in most …" Too general. Please specify the proportion at least.

*This has been indicated (L406)*

L361-365; This section does not relate to response ratios (even if stated in L361). Please rephrase.

*We believe that there is some confusion here. In this section we deal with responses based on both the raw data figures and the response ratios (Figs. 5, 6, 7 and S2). The ANOVA was done with RR data. This has been clarified (L411-413).*

L362-363 & 367; Here and elsewhere, provide F value and df.

*This has been corrected throughout the manuscript (L415, L420).*

L367-369; Revise English.

*This has been revised (L420-425)*

L369; Here and elsewhere, provide F value and df.

*This has been corrected throughout the manuscript.*

L369-372; Revise English.

*English has been revised (L425-428).*

L373-375; Maybe state in which experiments this happens? Similar to L387-390.

*The significant experiments are indicated by asterisks in figures S2, S3, and S4. We believe that the result section is rather complex as to also include this detailed information for replicate experiments. The reader can go to figures S2, S3 and S4 and check which experiments and treatments were significant.*

L373-383; I would suggest restructuring for clarity. As now it is very difficult to understand when different responses occurred.

*We agree that this paragraph describing the 36 experiments is a bit difficult to follow, and this was the reason of averaging the replicates. We have revised this fragment for clarity (L430-441).*

L377-378; Maybe state in which experiments this happens? Similar to L387-390.

*This information was added as specifically required by a reviewer, but we believe that the*

*reading of this fragment is already complex as to add such details on replicate experiments.*

L391-395; This part appears to belong in Material and Methods section.

*We believe that the calculation of a coefficient of variation is simple, and thus, there should be no need to explain that in the method section.*

L395; 4 sites? 2 stations and 2 depths? Please clarify.

*This has been clarified (L453)*

L397-400; To me, these results are the most interesting. I would suggest restructuring the result, putting emphasis on the response ratios.

*We agree. The description of figure 7 has been moved to section 3.2, which is the one dealing with the responses (L460-474)*

L405; "Most positive…". State proportion (%).

*Proportions haves been indicated (L463).*

L418-422; This part appears to belong in Material and Methods section.

*The description of the RELATE analysis is already in the Methods section, we decided to explain the procedure here as well for clarity.*

L422-423; What was Spearman Rho correlation with eukaryotic community composition.

*Eukaryotic community composition did not correlate with the B-vitamin responses (Spearman Rho = 0.054, p = 0.39).*

L426; Where does the 78% originate from? State each dimensions contribution.

*This value is the % cumulative variation of the DistLM (L501-502).*

L430-431; State each dimensions contribution to the 59.4%.

*This is now indicated (L505)*

L431-433; Revise English. Also, I'm struggling to see that the stations are actually separated.

*This has been revised and toned down (L506-507).*

L434; "… highly and positively correlated…". Revise English.

*English has been revised (L509)*

**Discussion**

L443-445; As you don't have measurements on B1, this statement is not fully true. Please tone down this statement.

*This statement has been toned down (L518-519)*

L446; What expectations are you referring to? These should be stated more clearly before.

*Considering the high short-time variability of the hydrographic conditions in the area (Álvarez-Salgado et al 1996), we expected a large inter-day variation in the responses to B vitamin amendments. This information has been added (L522-524)*

L448-452. What about predation pressure? Cellular demand of B vitamins? Actual cellular content of B vitamins? Should be expanded to include more potential explanations.

*As we have neither measured cellular content of B vitamins nor predation pressure we believe that expanding explanations here would be a largely speculative exercise. In addition, all the suggested explanations (predation pressure B vitamin cellular demand, etc.) are likely related to the seasonal succession of microbial plankton species.*

L452-454; In my opinion, this should have been done for all of the results. I understand that a tremendous amount of work has gone into this experiment, but I believe that the paper would benefit from a more succinct and concise result section.

*We tried to reduce as much as possible the part of the results related to raw data, and centered the discussion mostly in the general patterns that emerge when averaging replicate (a, b, c) experiments. We decided to keep figures 5 and 6 and describe also the raw data as it was specifically suggested by other reviewers. We believe that this revised version represents a good balance among all the received comments and suggestions.*

L456 "… frequent but relatively moderate…". What does this mean, please clarify.

*The significant responses were small, that is, the averaged increase of chl-a or prokaryote biomass after B vitamin additions did not exceed 1.3-fold. We have corrected an error detected here, as the 2.4-fold increase in figure 7 is not significant (L534-536)*

L461-464; What results are this statement based on?

*This is based on data presented in figure 7 and 8. This has been stated in the revised version (L542-543).*

L497-500; Highly speculative. Please rephrase to tone down this statement.

*This has been toned down (L578-580)*

L521; change "potentially" to potential

*This has been corrected (L601)*

L522-546; I would suggest reading Fridolfsson et al. 2018 and 2019, as well as Sylvander et al. 2013 to provide additional depth to the discussion on B1 and B12 amendments.

*We very much appreciate the suggested reading. We have included the study by Fridolfsson et al. 2019 to expand the discussion on B1 (L628-632)*

L563-566; Shouldn't dinoflagellates pop out in the analysis then?

*We also expected dinoflagellates contributing to explain the responses, however, the RELATE analysis found no correlation between eukaryote community and responses to B vitamins (RELATE, Rho = 0.054, p = 0.39). Also, the correlation between the clr abundance of dinoflagellates and the different B vitamin responses were not significant (analyses finally not included in the manuscript).*

L567; Why "strikingly"?

*This has been changed (L653)*

L570; change "revel" to reveal?

*This has been changed (L656)*

L576; Which "predation" are you referring to? Please clarify.

*This has been clarified (L663)*

L582-583; What about uptake rates? I would suggest reading Koch et al. 2011, 2012, 2013 and discuss.

*This has been discussed (L670-676).*

L588; "… B12 producers and B1 consumers." This is extremely generalized and implies that you can determine this in your paper. This is not fully true, especially for B1 as you don't have measurement for this B vitamin.

*This has been rephrased (L681).*

L590; "… cope with B vitamin shortage…". See L588. Once again, it is unfortunate, but you don't have measurements for B1 so your conclusions regarding this B vitamin should be toned down.

*We are aware that we do not have B1 measurements, but it is very likely that the concentration of B1 is also very low in the area, as both B vitamins tend to follow very*

*similar patterns (see e.g. Suffridge et al 2018). Nevertheless, this has been rephrased (L685).*

**Figure captions**

Please make sure that everything in your graphs can be identified. E.g fig 1, that cruises is illustrated by lines (in 1c legend), dots in fig 2, what 16:1 line refer to in fig 3.

Also, Generally, is there any benefit of using st3 and st6 instead of coastal and oceanic station. I feel the readability would increase if you used coastal and oceanic instead.

*The station numbers have been replaced by coastal and oceanic station.*

L937-941; Change "µmol l$^{-1}$" to µM? Pinpoint that axes are broken. Specify what SCM means.

*This has been specified (L1058-1061).*

L942-945; If so, state that it refers to t0. Also, what are the error bars showing?

*No, the community composition showed in this figure is the averaged composition during the cruises. Error bars have been explained (L1066-1067).*

L946-949; Suggestion, …(estimated as Chl-a concentration (µg l$^{-1}$)). Change "time-zero" to t0. Change "final-time" to endpoint. Pinpoint that axes are broken. Also, what are the error bars showing?

*This has been changed (L1070-1074)*

L950-952; Change "time-zero" to t0. Change "final-time" to endpoint. Pinpoint that axes are broken. Also, what are the error bars showing?

*This has been corrected (1077-1080).*

L953-960; I would suggest using more mosaics, a-d.

*The suggestion has been considered. More mosaics have been included.*

L961-968; change "… microbial plankton…" to microbial bakterioplankton, as it is only prokaryotes? Should be stated in the beginning and not at the end of the figure caption

*The Bray- Curtis similarities were built considering the responses of both phytoplankton and prokaryotes responses. This has been clarified (L1094).*

**Figures**

**Figure 1**; Generally, is there any benefit of using st3 and st6 instead of coastal and oceanic station. I feel the readability would increase if you used coastal and oceanic instead.

*Station IDs have been changed*

**Figure 2**; When referring to figures, state which of the figures, a, b or c. you refer to. Also, look over all figures so that all are labelled a, b, c …. For fig. 2 and 3 I'm having troubles seeing the benefits of having several a, several b etc. I would like to see labelling a-r instead for fig. 2, then you can refer to the specific mosaic.

*Labelling has been modified for clarity.*

**Figure 3**; See comment for fig 2. For fig. 2 and 3 I'm having troubles seeing the benefits of having several a, several b etc. For the legend, the depth is stated as 0m and SCM, change to "surface (5m) and SCM", as you did not sample 0m, correct? Also, state what the 16:1 line refers to. Also, I would suggest providing an average per station and cruise, and not all 3a, 3b and 3c etc, see general comments.

*All the suggested changes have been made.*

*We agree that there is much information, but still we strongly believe that most readers will like to see the raw data, to corroborate that the short-term changes were rather limited.*

**Figure 4**; change mosaics to cover a-c, as stated in the main text. On the x-axes, the depth is stated as 0m and SCM, change to "surface (5m) and SCM", as you did not sample 0m, correct? You do not use a consistent taxonomy level, some are species whilst other groups are a combination. Could this affect your results? If not, I would still reconsider the different taxonomical levels presented.

*The suggested changes have been made.*

*The depicted taxonomic groups were carefully defined based on their abundance and relevance. Focusing only at a given level (e.g. phylum, class, order) would omit and put together rather distinct functional groups. For example, within the Alphaproteobacteria class, SAR11 and the order Rhodobacterales occupy different niches, with SAR11 being representative of more oligotrophic conditions. Also within the Rhodobacterales, we found three genera that showed on average high abundance and that appeared to have different dynamics (Amylibacter, Ascidiaceihabitans, Planktomarina). In the case of cyanobacteria, they were dominated by the genus Synechococcus, which we find much more inofrmative. In the case of eukaryotes, the Clorophyta was dominated by ASVs belonging to the order Mamiellales, including two distinct genera (Ostreococcus and Micromonas) that also show*

*different dynamics in this coastal site (see Hernández-Ruiz et al 2018). The criteria to subdivide a taxonomic level was that the relative abundance of the resulting subgroups (lower taxonomic levels) was higher than 5 % in at least one occasion. No subdivisions were made for kingdoms or phyla with low relative abundance (e.g Planctomycetes, Verrucomicrobia, Fungi, Rhizaria).The use of different taxonomic levels is a common practice to depict community composition as functional microbial groups do not always match with taxonomic levels.*

**Figure 5 and Figure 6;** The colors are very difficult to distinguish. Also, I would suggest providing an average per station and cruise, and not all 3a, 3b and 3c etc, see general comments.

*We have corrected the colours for clarity. We decided to keep the figures 5 and 6 showing the raw data (replicate experiments a, b, c). Nevertheless, we focus the description of the results and the all the discussion on figures 7 and 8, based on the averaged data (see also the response to figure 3 comment).*

**Figure 7;** I would suggest changing the layout, to something used frequently when presenting fold change. You don't need to show 0, as every finding is around 1. See oversimplified suggestion below.

[Figure]

*Figure 7 has been modified following the suggestions made by the reviewer.*

**Figure 8;** You do not use a consistent taxonomy level, some are species whilst other groups are a combination. Could this affect your results? If not, I would still reconsider the different taxonomical levels presented. The legend needs formatting prior to publication, much too large as it is now. The depth is stated as 0m and SCM, change to "surface (5m) and SCM", as you did not sample 0m, correct?

*Legend has been modified. For this figure we used the 12 most abundant prokaryote groups as depicted in figure 4. We can only introduce 12 explicative variables as inputs in the DITLM model as we only have 12 data points (2 station x 2 depth x 3 seasons). See also the response to figure 4 comments.*

**Supplement information**

**Table S2**; This information is the same as in fig 3, correct? To me, this is redundant. If to be included, abbreviations in column names should be explained.

*Table S2 shows all information taken from t0 of each experiment while Figure 3 only shows initial biomasses, DIN (which is the sum of nitrate, ammonium and nitrite) and ratio DIN:DIP. Column names have been added.*

L18-27; "… experiments by the averaged…". Add divided? Change "that means" to which implies. Pinpoint that axes are broken.

*This has been modified.*

**Figure S1**; Shouldn't axes present statistics?, Percentages? The legend needs formatting prior to publication, much too large as it is now.

*Axis of this graph do not included values because it is a non-metric multidimensional analysis (MDS). The MDS significance was tested by ANOSIM (analysis of similarity) (L 305--311).*

*Legend has been modified (Fig. S1)*

**Figure S2**; I propose including this graph over Fig 5 and 6. If included, it must be formatted to conform to the palette the authors have used, for clarity. How was these stats performed, as RR already considers the control. Clarify.

*This figure shows the ratio of chlorophyll-a or prokaryote biomass in the inorganic treatment divided by the corresponding variable in the control at the end-point for the 36 experiments (2 months x 2 stations x 2 depths x 3 experiments (a, b, c)).*

*Figure 5 and 6 show the raw chlorophyll-a and prokaryote biomass at t0 and at the end-point for each treatment (control, inorganic nutrients, B12, B1 and all combinations) in the 36 experiments. Therefore, we do not see how to include this graph (RRs) over figures 5 and 6*

*(raw data). The significance of each RR in figures S2, S3 and S4 was tested using a t-test, comparing averaged (from the three replicates) values between two treatments.*

*We represented the response ratios to inorganic nutrient separately as we wanted to focus the attention on the responses ratios to vitamins added either solely (vitamin treatment/control) or in combination with inorganic nutrients (vitamin treatment/inorganic). The response ratios to vitamins are represented in figures S3 and S4. The averaged RRs to vitamins are represented in figure 7 in the manuscript.*

Figure S3; I would suggest changing the layout, to something used frequently when presenting fold change. You don't need to show 0, as every finding is around 1. See comment for figure 7. As it is now, it is impossible to get any valuable information from the figure.

*The layout of Figure S3 has been modified to facilitate its understanding. Now, two figures (Fig. S3 and Fig. S4) have been created for better clarity.*

[revised manuscript text omitted]

**Figure 3:** Initial biological conditions and abiotic factors at the coastal (st3) and oceanic (st6) sampling stations. Each bar corresponds to one of the 3 experiments performed in each depth and station during February, April and August. (a, b, c), Chl-*a*, total Chl-*a* (µg l$^{-1}$). Note that the y-axis is broken; (d, e, f) PB, bacterial prokaryote biomass (µg C l$^{-1}$); (g, h, i) DIN, dissolved inorganic nitrogen ($\mu$mol l$^{-1}$) and (j, k, l) DIN:DIP, ratio inorganic nitrogen:phosphate. The blue line shows the Redfield ratio (16:1) and SCM refers to the sub-surface chlorophyll maximum. Chl-*a*: Chlorophyll-a concentration.

**Figure 4:**  Averaged relative contribution of reads to the major taxonomic groups of (a) eukaryotes and (b) prokaryotes at surface and SCM in the coastal and oceanic station in February, April and August. (c) Averaged B12 concentration (pmol l$^{-1}$) at surface and SCM in the coastal and oceanic station in February, April and August. Error bars represent standard error.

**Figure 5**:

Chlorophyll-a concentration ($\mu$g l$^{-1}$) in the t0 of each experiment (striped bars) and in the endpoint of each treatment (colored bars) in the experiments conducted at (a)

5 m and (b) SCM in the coastal and at (c) surface and (c) SCM in the oceanic station in

February, April and August. Error bars represent standard error. Note that the y-axis is broken. SCM: sub-surface chlorophyll maximum.

**Figure 6**: Prokaryote  biomass ($\mu$g C l$^{-1}$) in the t0 of each experiment (striped bars) and in the endpoint of each treatment (colored bars) in the experiments conducted at (a) surface and (b) SCM in the coastal and at (c) surface and (d) SCM in the oceanic station in February, April and August. Error bars represent standard error. Note that the y-axis is broken. SCM: sub-surface chlorophyll maximum.

**Figure 7:** Monthly averaged response ratio (RR) of (a)  Chl-*a* or (b)

prokaryote biomass at surface and SCM in the coastal and oceanic station

.

Horizontal line represents a response equal to 1, that means no change relative to control in the pink bars (treatments with vitamins alone) and no change relative to inorganic (I)

treatment in the green bars (vitamins combined with I treatments). Asterisks indicate averaged RRs that were significantly different from 1 (Z-test; * $p < 0.05$) and "a" symbols indicate averaged

RRs that were marginally significant (Z-test; [a] $p = 0.05\text{-}0.06$). SCM: sub-surface chlorophyll maximum.

**Figure 8:** Distance based redundancy analysis (dbRDA) of B vitamin responses by phytoplankton and prokaryotes  based on Bray-Curtis similarity. Only prokaryotic taxa that explained variability in the B vitamin responses structure selected in the DistLM model (step-wise procedure with adjusted $R^2$ criterion) were fitted to the ordination. Filled and open symbols represent samples from coastal and oceanic station, respectively, numbers correspond to the sampling station, triangles and circles represent samples from surface and SCM, respectively, and colours correspond to the months:

(green) February, (blue) April and (pink) August.

SCM: sub-surface chlorophyll maximum.

Table 1

| | Treatment | Nutrient included | Concentration |
|---|---|---|---|
| 1. | Control (C) | No nutrient added | |
| 2. | Inoganic nutrients (I) | $NO_3^-$ | 5 μmol l$^{-1}$ |
| | | $NH_4^+$ | 5 μmol l$^{-1}$ |
| | | $HPO_4^{2-}$ | 1 μmol l$^{-1}$ |
| | | $SiO_4^{2-}$ | 5 μmol l$^{-1}$ |
| 3. | Vitamin B12 (B12) | B12 | 100 pmol l$^{-1}$ |
| 4. | Vitamin B1 (B1) | B1 | 600 pmol l$^{-1}$ |
| 5. | B12 + B1 | B12 | 100 pmol l$^{-1}$ |
| | | B1 | 600 pmol l$^{-1}$ |
| 6. | I + B12 | $NO_3^-$ | 5 μmol l$^{-1}$ |
| | | $NH_4^+$ | 5 μmol l$^{-1}$ |
| | | $HPO_4^{2-}$ | 1 μmol l$^{-1}$ |
| | | $SiO_4^{2-}$ | 5 μmol l$^{-1}$ |
| | | B12 | 100 pmol l$^{-1}$ |
| 7. | I + B1 | $NO_3^-$ | 5 μmol l$^{-1}$ |
| | | $NH_4^+$ | 5 μmol l$^{-1}$ |
| | | $HPO_4^{2-}$ | 1 μmol l$^{-1}$ |
| | | $SiO_4^{2-}$ | 5 μmol l$^{-1}$ |
| | | B1 | 600 pmol l$^{-1}$ |
| 8. | I + B12 + B1 | $NO_3^-$ | 5 μmol l$^{-1}$ |
| | | $NH_4^+$ | 5 μmol l$^{-1}$ |
| | | $HPO_4^{2-}$ | 1 μmol l$^{-1}$ |
| | | $SiO_4^{2-}$ | 5 μmol l$^{-1}$ |
| | | B12 | 100 pmol l$^{-1}$ |
| | | B1 | 600 pmol l$^{-1}$ |

Figure 01

[Figure]

**Figure 02**

[Figure]

Figure 03

[Figure]

[Figure]

Figure 04

Figure 05

[Figure]

Figure 06

[Figure]

[Figure]

Figure 08

[Figure]

**Supplement information**

**Table S1:** concentration of hydroxocobalamin (OHB12) and cyanocobalamin (CNB12)

in seawater samples corresponding to the initial time of the experiments. Abbreviations:

Not detected (nd) and lower concentration of the quantification limit (<LOQ).

| Sample ID | Station | Depth | Month | OHB12 pmol l$^{-1}$ | CNB12 pmol l$^{-1}$ |
|---|---|---|---|---|---|
| 1602_st3_d1_p1 | coast | surface | February | 0.21 | nd |
| 1602_st3_d3_p1 | coast | surface | February | 0.20 | nd |
| 1602_st3_d5_p1 | coast | surface | February | 0.26 | nd |
| 1604_st3_d1_p1 | coast | surface | April | 0.47 | nd |
| 1604_st3_d3_p1 | coast | surface | April | 0.66 | nd |
| 1604_st3_d5_p1 | coast | surface | April | 0.23 | nd |
| 1608_st3_d1_p1 | coast | surface | August | 0.30 | nd |
| 1608_st3_d3_p1 | coast | surface | August | 0.38 | nd |
| 1608_st3_d5_p1 | coast | surface | August | 0.19 | nd |
| 1602_st3_d1_p2 | coast | SCM | February | 0.36 | nd |
| 1602_st3_d3_p2 | coast | SCM | February | 0.10 | nd |
| 1602_st3_d5_p2 | coast | SCM | February | 0.41 | nd |
| 1604_st3_d1_p2 | coast | SCM | April | 0.32 | nd |
| 1604_st3_d3_p2 | coast | SCM | April | 0.27 | nd |
| 1604_st3_d5_p3 | coast | SCM | April | 0.15 | nd |
| 1608_st3_d1_p2 | coast | SCM | August | 0.46 | nd |
| 1608_st3_d3_p2 | coast | SCM | August | 0.21 | nd |
| 1608_st3_d5_p2 | coast | SCM | August | 0.39 | nd |
| 1602_st6_d1_p1 | ocean | surface | February | 0.31 | nd |
| 1602_st6_d3_p1 | ocean | surface | February | 0.09 | nd |
| 1602_st6_d5_p1 | ocean | surface | February | 0.06 | nd |
| 1604_st6_d1_p1 | ocean | surface | April | 0.13 | nd |
| 1604_st6_d3_p1 | ocean | surface | April | 0.09 | nd |
| 1604_st6_d6_p1 | ocean | surface | April | 0.04 | nd |
| 1608_st6_d1_p1 | ocean | surface | August | 0.20 | nd |
| 1608_st6_d3_p1 | ocean | surface | August | 0.09 | nd |
| 1608_st6_d6_p1 | ocean | surface | August | 0.14 | nd |
| 1602_st6_d1_p3 | ocean | SCM | February | 0.21 | 0.55 |
| 1602_st6_d3_p2 | ocean | SCM | February | 0.08 | nd |
| 1604_st6_d1_p2 | ocean | SCM | April | nd | nd |
| 1604_st6_d3_p2 | ocean | SCM | April | 0.07 | nd |
| 1604_st6_d6_p2 | ocean | SCM | April | 0.05 | nd |
| 1608_st6_d1_p2 | ocean | SCM | August | 0.19 | nd |
| 1608_st6_d3_p2 | ocean | SCM | August | 0.09 | nd |
| 1608_st6_d6_p2 | ocean | SCM | August | 0.16 | nd |

**Table S2:** Summary of initial conditions for each experiment (expt) at both coastal and oceanic stations (Stn). Sampling months were February (Feb), April (Apr) and August (Aug). The variables measured at t0 were temperature (Temp), salinity (Sal), nitrate ($NO_3^-$

), nitrite ($NO_2^-$), ammonium ($NH_4^+$), phosphate ($HPO_4^{2-}$), ratio inorganic nitrogen:phosphate (DIN:P), silicate ($SiO_4^{2-}$), Chlorophyll-*a* (Chl-*a*) and prokaryote biomass (PB).

| Stn | Depth | Month | Expt | Day | Temp °C | Sal | NO₃⁻ | NO₂⁻ | NH₄⁺ | HPO₄²⁻ | DIN:P | SiO₄²⁻ | Chl-*a* | PB |
|---|---|---|---|---|---|---|---|---|---|---|---|---|---|---|
| | | | | | | | ------------- µmol l⁻¹ -------------- | | | | | µmol l⁻¹ | µg l⁻¹ | µg C l⁻¹ |
| Coast | surface | Feb | 3a | 0 | 13.8 | 35.0 | 2.86 | 0.19 | 0.35 | 0.17 | 19.7 | 3.6 | 1.39 | 1.84 |
| | | | 3b | 2 | 13.2 | 34.3 | 4.89 | 0.36 | 0.51 | 0.33 | 17.3 | 6.8 | 0.73 | 1.91 |
| | | | 3e | 5 | 13.4 | 34.2 | 4.63 | 0.19 | 0.09 | 0.18 | 27.7 | 8.6 | 4.86 | 3.45 |
| | | Apr | 3a | 0 | 13.0 | 34.6 | 2.21 | 0.24 | 0.32 | 0.19 | 14.6 | 5.2 | 2.73 | 7.88 |
| | | | 3b | 2 | 13.3 | 34.3 | 12.46 | 0.36 | 0.54 | 0.41 | 32.7 | 12.6 | 1.40 | 9.17 |
| | | | 3e | 5 | 14.0 | 31.8 | 4.18 | 0.16 | 0.55 | 0.19 | 25.9 | 10.5 | 2.18 | 4.30 |
| | | Aug | 3a | 0 | 14.1 | 35.6 | 0.50 | 0.10 | 0.84 | 0.12 | 11.8 | 1.1 | 5.73 | 14.64 |
| | | | 3b | 2 | 14.4 | 35.6 | 0.81 | 0.08 | 1.08 | 0.20 | 9.9 | 0.3 | 5.52 | 6.39 |
| | | | 3e | 5 | 13.7 | 35.2 | 3.93 | 0.17 | 0.12 | 0.33 | 12.8 | 3.9 | 5.64 | 10.61 |
| | SCM | Feb | 3a | 0 | 13.7 | 35.7 | 3.58 | 0.14 | 0.04 | 0.31 | 12.1 | 5.2 | 0.21 | 1.30 |
| | | | 3b | 2 | 13.9 | 35.3 | 4.16 | 0.15 | 0.07 | 0.37 | 11.9 | 4.6 | 0.99 | 1.83 |
| | | | 3e | 5 | 13.4 | 34.7 | 2.94 | 0.09 | 0.10 | 0.17 | 18.4 | 6.1 | 4.98 | 2.36 |
| | | Apr | 3a | 0 | 12.8 | 35.3 | 3.22 | 0.34 | 0.46 | 0.28 | 14.3 | 4.4 | 0.99 | 5.90 |
| | | | 3b | 2 | 13.2 | 35.3 | 0.24 | 0.07 | 0.12 | 0.04 | 10.2 | 2.8 | 2.15 | 9.47 |
| | | | 3e | 5 | 13.9 | 34.9 | 0.21 | 0.07 | 0.10 | 0.06 | 6.5 | 3.4 | 2.18 | 9.51 |
| | | Aug | 3a | 0 | 13.6 | 35.6 | 0.91 | 0.13 | 0.23 | 0.15 | 8.3 | 1.7 | 20.75 | 12.71 |
| | | | 3b | 2 | 13.8 | 35.6 | 1.40 | 0.16 | 0.14 | 0.23 | 7.5 | 1.4 | 20.07 | 1.73 |
| | | | 3e | 5 | 13.4 | 35.6 | 5.29 | 0.13 | 0.14 | 0.41 | 13.5 | 3.9 | 4.63 | 9.21 |
| Ocean | surface | Feb | 6a | 1 | 14.0 | 30.2 | 1.32 | 0.18 | 0.11 | 0.16 | 10.1 | 3.2 | 0.82 | 2.38 |
| | | | 6b | 3 | 14.2 | 35.9 | 0.90 | 0.11 | 0.04 | 0.12 | 9.2 | 2.3 | 1.20 | 2.98 |
| | | | 6e | 6 | 14.1 | 35.4 | 1.03 | 0.15 | 0.13 | 0.16 | 8.4 | 3.0 | 2.08 | 2.92 |
| | | Apr | 6a | 1 | 13.4 | 35.7 | 0.95 | 0.11 | 0.06 | 0.12 | 9.6 | 2.3 | 1.51 | 6.58 |
| | | | 6b | 3 | 13.6 | 35.7 | 0.47 | 0.11 | 0.06 | 0.08 | 8.3 | 2.7 | 1.29 | 7.37 |
| | | | 6e | 6 | 13.9 | 35.6 | 0.12 | 0.03 | 0.06 | 0.04 | 4.9 | 2.1 | 0.75 | 11.76 |
| | | Aug | 6a | 1 | 16.0 | 35.6 | 0.05 | 0.01 | 0.06 | 0.02 | 4.9 | 1.5 | 0.65 | 39.38 |
| | | | 6b | 3 | 16.0 | 35.6 | 0.26 | 0.01 | 0.09 | 0.05 | 7.5 | 3.2 | 0.99 | 11.46 |
| | | | 6e | 6 | 15.3 | 35.5 | 0.45 | 0.04 | 0.05 | 0.07 | 7.4 | 1.4 | 1.30 | 5.63 |
| | SCM | Feb | 6a | 1 | 14.1 | 35.8 | 1.73 | 0.20 | 0.04 | 0.18 | 11.2 | 3.5 | 0.88 | 2.28 |
| | | | 6b | 3 | 14.1 | 35.8 | 1.60 | 0.19 | 0.02 | 0.15 | 11.7 | 2.9 | 1.22 | 3.18 |
| | | | 6e | 6 | 14.1 | 35.8 | 1.13 | 0.18 | 0.12 | 0.16 | 9.2 | 2.9 | 2.39 | 3.49 |
| | | Apr | 6a | 1 | 13.3 | 35.7 | 1.63 | 0.31 | 0.10 | 0.18 | 11.5 | 3.2 | 1.61 | 5.38 |
| | | | 6b | 3 | 13.3 | 35.7 | 1.45 | 0.33 | 0.12 | 0.16 | 11.9 | 2.4 | 1.50 | 6.96 |
| | | | 6e | 6 | 13.7 | 35.6 | 0.03 | 0.06 | 0.07 | 0.05 | 3.0 | 1.9 | 1.45 | 11.74 |
| | | Aug | 6a | 1 | 14.9 | 35.6 | 0.00 | 0.04 | 0.10 | 0.03 | 4.2 | 1.4 | 0.84 | 26.55 |
| | | | 6b | 3 | 16.0 | 35.6 | 0.27 | 0.00 | 0.07 | 0.05 | 6.5 | 2.8 | 1.11 | 6.04 |
| | | | 6e | 6 | 15.4 | 35.6 | 0.35 | 0.06 | 0.06 | 0.07 | 6.5 | 1.7 | 1.41 | 5.45 |

**Figure S1:** A non-metric multi-dimensional scaling (MDS) showing the distance according to similarity in the microbial plankton composition at the beginning of each experiment (each symbol). Filled and open symbols represent samples from coastal and oceanic station, respectively, numbers correspond to the sampling station, triangles and circles represent samples from surface and SCM, respectively, and colours correspond to the months: (green) February, (blue) April and (pink) August. SCM: sub-surface chlorophyll maximum.

**Figure S2:** Response ratio (RR) to inorganic nutrient addition (averaged biomass at the end of the experiments divided by the averaged value in the control) of total phytoplankton community (smooth bars) and of  prokaryote biomass (PB) (striped bars) at (a) coastal and (b) oceanic station. Each bar corresponds to one of the 3 experiments (a, b or c) performed in each depth and station during February, April and August. Colours represent samples from (light grey) surface and (dark grey) SCM. Horizontal line represents a response equal to 1, which implies no change relative to control. Asterisks indicate phytoplankton significant response  (t-test; * $p < 0.05$) and circle indicate bacterial significant response  (t-test; $^0$ $p < 0.05$). Note that different scales were used. Note that y-axis in Fig. S2 b is broken. SCM: sub-surface chlorophyll maximum.

**Figure S3:** Response ratio (RR) of total phytoplankton  at surface and SCM in the coastal station and at surface and SCM in the oceanic waters in (a-d) February, (e-h) April and (i-l) August. Treatments represented are: B12/C; B1/C; B12+B1/C in pink tones and I+B12/I; I+B1/I; I+B12+B1/I in green tones. Pink symbols represent primary responses to B vitamins and green symbols represent secondary responses to B vitamins. Horizontal dotted-line represents a response equal to 1, that means no change relative to control in the primary responses, and no change relative to inorganic treatment in the secondary responses. Asterisks indicate phytoplankton significant response (t-test; * p < 0.05) and circle indicate bacterial significant response (t-test; ° p < 0.05). Note that different scales were used.

**Figure S4:** Response ratio (RR) of prokaryote biomass at surface and SCM in the coastal station and at surface and SCM in the oceanic waters in (a-d) February, (e-h) April and (i-l) August. Treatments represented are: B12/C; B1/C; B12+B1/C in pink tones and I+B12/I; I+B1/I; I+B12+B1/I in green tones. Pink simbolssymbols represent primary responses to B vitamins and green simbolssymbols represent secondary responses to B vitamins. Horizontal dotted-line represents a response equal to 1, that means no change relative to control in the primary responses, and no change relative to inorganic treatment in the secondary responses. Asterisks indicate prokaryote significant response (t-test; * p < 0.05).

Figure S1

[Figure]

Figure S2

[Figure]

**Chlorophyll-a Responses**

[Figure]

**Prokaryote Biomass Responses**

---

## Author Response (AR3)

Vanessa Joglar

Biological Oceanography Group

Koji Suzuki

Associate Editor

Biogeosciences

Vigo, 03th April 2020

Dear Koji

Please find attached a new revised version of manuscript entitled "Spatial and temporal variability in the response of phytoplankton and bacterioplankton to B-vitamin amendments in an upwelling system". The manuscript was co-authored by myself, Antero Prieto, Esther Barber-Lluch, Marta Hernández-Ruiz, Emilio Fernández and Eva Teira.

We are grateful that you have appreciated the effort to improve this work. All suggested changes have been considered as well as all the issues raised have been answered.

A detailed response to all comments is attached. The suggestions and comments of the reviewer are in plain font and our responses are in italic and blue font. The revised version of the manuscript with marked changes is also provided.

Looking forward to hearing from you,

Vanessa Joglar

Full address for corresponding is:

Grupo de Oceanografía Biológica

Departamento de Ecología y Biología Animal

Universidad de Vigo

Campus Universitario Lagoas-Marcosende

36310-Vigo Spain

E-mail: vjoglar@uvigo.es

Review v2 of "Spatial and temporal variability in the response of phytoplankton and bacterioplankton to B-vitamin amendments in an upwelling system" by Joglar et al.

**General comments**

The authors have put considerable effort into responding to all my previous concerns. The sampling campaign is definitely impressive, as well as the work that went in to the study and I think that the results and discussions makes this effort justice now.

I have some minor comments for the author, to further help with the readability and clarity of the manuscript. Some points are purely editorial whilst others needs to be answered and the text changed. I would also like to congratulate the authors on a job well done, both on the cruise, lab and writing a very interesting manuscript.

*We very much appreciate the very constructive revision made by the reviewer.*

**Specific comments**
**Introduction**

L36-39; I feel the text would benefit from more precise examples, e.g. cyanobacterial blooms, red tides etc.

*Precise examples have been added (L37-39)*

L71; change "drive" to thrive?

*This has been changed (L73)*

**Methods**

L213; change "inned" to inner.

*This has been corrected (L214)*

L226; For clarity, add pmol l$^{-1}$ after 0.04.

*Units have been included (L227)*

L263; μm is in blue, change to black.

*This has been changed (L264)*

L282; For clarity, I would like that the non-normal variables are stated somewhere, either here or in supplementary material.

*Non-normal variables have been included (L284)*

L288-289; Did you only compare differences between treatments and the control and not between all treatments? If so, why?

*We compared all treatments but only reported differences between B vitamins and the control, inorganic nutrients and the control and B vitamins+Inorganic nutrients and the inorganic nutrients, in order to simplify the result section.*

L289-292; I realize this might be due to different traditions, but for me non-metric multidimensional scaling is abbreviated as nMDS. It is no requirement to change, I simply wanted to raise the concern.

*This has been corrected (L292)*

**Results**

L339; change "below of" to "below the".

*This has been changed (L340)*

L341-342; Does this statement relate to the average chl a levels, per month? If so it should be stated more clearly. If not, this does not seem to be the case in some days (a, b and c). Please look into this and change statement if needed or clarify.

*This has been clarified (L342)*

L343; Add reference to figure 3d-f.

*This has been included (L345)*

L357; "… sampling dates…", maybe change to cruise if applicable.

*This has been changed (L358)*

L360; "… but their abundance…" add "relative" for clarity.

*This has been clarified (L361)*

L372; Add "." before Average…

*This has been corrected (L373).*

L373; what does "gl=10" mean? If it is degrees of freedom, use df. In not you should still state df.

*This has been corrected (L374)*

L380-381; "However, Chl-a mostly decreased in the coastal experiments conducted in August (Fig. 5a and Fig. 5c)." I do not agree with this statement, as this is not was is shown in the figures. For instance, all bars in a shade of blue is always higher for the than t0 for August samplings.

*That phrase refers only to changes from t0 to the end-point in the control treatment (grey bars).*

L446; Even if the eukaryotic community composition did not correlate significantly, you should still present the correlation coefficient and p value for this.

*Correlation coefficient and p value for eukaryotes have been included (L446-448).*

L450; Maybe remove underscore in "SAR11_clade"? If this is common practice, please ignore.

*SAR11_clade has been replaced by SAR11 (L453)*

L458; Change to *Planktomarina*.

*This has been corrected (L461)*

**Discussion**

L485; Change "bacteria" to prokaryotes.

*This has been changed (L488)*

L486; State which experiment situation you refer to.

*This has been corrected as we detected an error in this statement (L489-490)*

L494-499; This sentence is too long (63 words), please restructure to give the reader a chance to follow.

*This sentence has been restructured (L499-502)*

L530-533 and 544; Change "cobalamin" to B12.

*This has been changed (L534)*

L602; "Flavobacteriia", is this correct?

*Flavobacteria and Flavobacteriia are both correct, any way, the name has been changed (L606).*

L608; Which predation do you refer to? Zooplankton or mixotrophs? Please clarify.

*This has been clarified (L612)*

**Figure captions**

L985-989; Add space between *shelf* and *(Oc)*. You do not have any ns in figure, can be removed?

*This has been corrected (L990-993)*

L991-995. This figure caption is incorrect. Now you have more facets/mosaics, please update the caption accordingly.

*This has been corrected (L995-1002)*

L1005-1009. In the figure you have 5m and SCM, but in caption you have surface and SCM. I would suggest changing the figure x axes. Add information about SCM.

*This has been corrected (L1013-1016)*

L1011-1015; In the figure you have surface and SCM, but in caption you have 5m and SCM. Please be consistent. Change "(c) SCM" to (d) SCM.

*This has been corrected (L1020)*

L1023-1030; Change "bars" to dots or points. Add information about error bars.

*This has been changed (L1032-1037)*

L1032-1039; You don't have any "numbers" anymore. Can be removed from caption.

*This has been corrected (L1044)*

**Figures**

**Figure 8;** In the figure you have 5m and SCM, but in the manuscript you have surface and SCM. Please be consistent.

*This has been corrected*

**Supplement information**

**Figure S1**; In the figure you have 5m and SCM, but in the manuscript you have surface and SCM. Please be consistent.

*This has been corrected*

**Figure S3 + caption;** State that y axis is broken for a and b.

*This has been added (L40 in the supplement)*

[revised manuscript text omitted]

Fig. 03

[Figure]

[Figure]

Fig. 04

[Figure]

[Figure]

[Figure]

Fig. 07

Fig. 08

[Figure]

    **Supplement information**

    **Table S1:** concentration of hydroxocobalamin (OHB12) and cyanocobalamin (CNB12)

    in seawater samples corresponding to the initial time of the experiments. Abbreviations:

    Not detected (nd) and lower concentration of the quantification limit (<LOQ).

| Sample ID | Station | Depth | Month | OHB12 pmol $l^{-1}$ | CNB12 pmol $l^{-1}$ |
|---|---|---|---|---|---|
| 1602_st3_d1_p1 | coast | surface | February | 0.21 | nd |
| 1602_st3_d3_p1 | coast | surface | February | 0.20 | nd |
| 1602_st3_d5_p1 | coast | surface | February | 0.26 | nd |
| 1604_st3_d1_p1 | coast | surface | April | 0.47 | nd |
| 1604_st3_d3_p1 | coast | surface | April | 0.66 | nd |
| 1604_st3_d5_p1 | coast | surface | April | 0.23 | nd |
| 1608_st3_d1_p1 | coast | surface | August | 0.30 | nd |
| 1608_st3_d3_p1 | coast | surface | August | 0.38 | nd |
| 1608_st3_d5_p1 | coast | surface | August | 0.19 | nd |
| 1602_st3_d1_p2 | coast | SCM | February | 0.36 | nd |
| 1602_st3_d3_p2 | coast | SCM | February | 0.10 | nd |
| 1602_st3_d5_p2 | coast | SCM | February | 0.41 | nd |
| 1604_st3_d1_p2 | coast | SCM | April | 0.32 | nd |
| 1604_st3_d3_p2 | coast | SCM | April | 0.27 | nd |
| 1604_st3_d5_p3 | coast | SCM | April | 0.15 | nd |
| 1608_st3_d1_p2 | coast | SCM | August | 0.46 | nd |
| 1608_st3_d3_p2 | coast | SCM | August | 0.21 | nd |
| 1608_st3_d5_p2 | coast | SCM | August | 0.39 | nd |
| 1602_st6_d1_p1 | ocean | surface | February | 0.31 | nd |
| 1602_st6_d3_p1 | ocean | surface | February | 0.09 | nd |
| 1602_st6_d5_p1 | ocean | surface | February | 0.06 | nd |
| 1604_st6_d1_p1 | ocean | surface | April | 0.13 | nd |
| 1604_st6_d3_p1 | ocean | surface | April | 0.09 | nd |
| 1604_st6_d6_p1 | ocean | surface | April | 0.04 | nd |
| 1608_st6_d1_p1 | ocean | surface | August | 0.20 | nd |
| 1608_st6_d3_p1 | ocean | surface | August | 0.09 | nd |
| 1608_st6_d6_p1 | ocean | surface | August | 0.14 | nd |
| 1602_st6_d1_p3 | ocean | SCM | February | 0.21 | 0.55 |
| 1602_st6_d3_p2 | ocean | SCM | February | 0.08 | nd |
| 1604_st6_d1_p2 | ocean | SCM | April | nd | nd |
| 1604_st6_d3_p2 | ocean | SCM | April | 0.07 | nd |
| 1604_st6_d6_p2 | ocean | SCM | April | 0.05 | nd |
| 1608_st6_d1_p2 | ocean | SCM | August | 0.19 | nd |
| 1608_st6_d3_p2 | ocean | SCM | August | 0.09 | nd |
| 1608_st6_d6_p2 | ocean | SCM | August | 0.16 | nd |

**Table S2:** Summary of initial conditions for each experiment (expt) at both coastal and oceanic stations (Stn). Sampling months were February (Feb), April (Apr) and August (Aug). The variables measured at t0 were temperature (Temp), salinity (Sal), nitrate ($NO_3^-$

), nitrite ($NO_2^-$), ammonium ($NH_4^+$), phosphate ($HPO_4^{2-}$), ratio inorganic nitrogen:phosphate (DIN:P), silicate ($SiO_4^{2-}$), Chlorophyll-*a* (Chl-*a*) and prokaryote biomass (PB).

**Table S2**

| Stn | Depth | Month | Expt | Day | Temp °C | Sal | NO$_3^-$ | NO$_2^-$ | NH$_4^+$ | HPO$_4^{2-}$ | DIN:P | SiO$_4^{2-}$ µmol l$^{-1}$ | Chl-$a$ µg l$^{-1}$ | PB µg C l$^{-1}$ |
|---|---|---|---|---|---|---|---|---|---|---|---|---|---|---|
|  |  |  |  |  |  |  | ------------- µmol l$^{-1}$ -------------- |  |  |  |  |  |  |  |
| Coast | surface | Feb | 3 | 0 | 13.8 | 35.0 | 2.86 | 0.19 | 0.35 | 0.17 | 19.7 | 3.6 | 1.39 | 1.84 |
|  |  |  | 3 | 2 | 13.2 | 34.3 | 4.89 | 0.36 | 0.51 | 0.33 | 17.3 | 6.8 | 0.73 | 1.91 |
|  |  |  | 3 | 5 | 13.4 | 34.2 | 4.63 | 0.19 | 0.09 | 0.18 | 27.7 | 8.6 | 4.86 | 3.45 |
|  |  | Apr | 3 | 0 | 13.0 | 34.6 | 2.21 | 0.24 | 0.32 | 0.19 | 14.6 | 5.2 | 2.73 | 7.88 |
|  |  |  | 3 | 2 | 13.3 | 34.3 | 12.46 | 0.36 | 0.54 | 0.41 | 32.7 | 12.6 | 1.40 | 9.17 |
|  |  |  | 3 | 5 | 14.0 | 31.8 | 4.18 | 0.16 | 0.55 | 0.19 | 25.9 | 10.5 | 2.18 | 4.30 |
|  |  | Aug | 3 | 0 | 14.1 | 35.6 | 0.50 | 0.10 | 0.84 | 0.12 | 11.8 | 1.1 | 5.73 | 14.64 |
|  |  |  | 3 | 2 | 14.4 | 35.6 | 0.81 | 0.08 | 1.08 | 0.20 | 9.9 | 0.3 | 5.52 | 6.39 |
|  |  |  | 3 | 5 | 13.7 | 35.2 | 3.93 | 0.17 | 0.12 | 0.33 | 12.8 | 3.9 | 5.64 | 10.61 |
|  | SCM | Feb | 3 | 0 | 13.7 | 35.7 | 3.58 | 0.14 | 0.04 | 0.31 | 12.1 | 5.2 | 0.21 | 1.30 |
|  |  |  | 3 | 2 | 13.9 | 35.3 | 4.16 | 0.15 | 0.07 | 0.37 | 11.9 | 4.6 | 0.99 | 1.83 |
|  |  |  | 3 | 5 | 13.4 | 34.7 | 2.94 | 0.09 | 0.10 | 0.17 | 18.4 | 6.1 | 4.98 | 2.36 |
|  |  | Apr | 3 | 0 | 12.8 | 35.3 | 3.22 | 0.34 | 0.46 | 0.28 | 14.3 | 4.4 | 0.99 | 5.90 |
|  |  |  | 3 | 2 | 13.2 | 35.3 | 0.24 | 0.07 | 0.12 | 0.04 | 10.2 | 2.8 | 2.15 | 9.47 |
|  |  |  | 3 | 5 | 13.9 | 34.9 | 0.21 | 0.07 | 0.10 | 0.06 | 6.5 | 3.4 | 2.18 | 9.51 |
|  |  | Aug | 3 | 0 | 13.6 | 35.6 | 0.91 | 0.13 | 0.23 | 0.15 | 8.3 | 1.7 | 20.75 | 12.71 |
|  |  |  | 3 | 2 | 13.8 | 35.6 | 1.40 | 0.16 | 0.14 | 0.23 | 7.5 | 1.4 | 20.07 | 1.73 |
|  |  |  | 3 | 5 | 13.4 | 35.6 | 5.29 | 0.13 | 0.14 | 0.41 | 13.5 | 3.9 | 4.63 | 9.21 |
| Ocean | surface | Feb | 6 | 1 | 14.0 | 30.2 | 1.32 | 0.18 | 0.11 | 0.16 | 10.1 | 3.2 | 0.82 | 2.38 |
|  |  |  | 6 | 3 | 14.2 | 35.9 | 0.90 | 0.11 | 0.04 | 0.12 | 9.2 | 2.3 | 1.20 | 2.98 |
|  |  |  | 6 | 6 | 14.1 | 35.4 | 1.03 | 0.15 | 0.13 | 0.16 | 8.4 | 3.0 | 2.08 | 2.92 |
|  |  | Apr | 6 | 1 | 13.4 | 35.7 | 0.95 | 0.11 | 0.06 | 0.12 | 9.6 | 2.3 | 1.51 | 6.58 |
|  |  |  | 6 | 3 | 13.6 | 35.7 | 0.47 | 0.11 | 0.06 | 0.08 | 8.3 | 2.7 | 1.29 | 7.37 |
|  |  |  | 6 | 6 | 13.9 | 35.6 | 0.12 | 0.03 | 0.06 | 0.04 | 4.9 | 2.1 | 0.75 | 11.76 |
|  |  | Aug | 6 | 1 | 16.0 | 35.6 | 0.05 | 0.01 | 0.06 | 0.02 | 4.9 | 1.5 | 0.65 | 39.38 |
|  |  |  | 6 | 3 | 16.0 | 35.6 | 0.26 | 0.01 | 0.09 | 0.05 | 7.5 | 3.2 | 0.99 | 11.46 |
|  |  |  | 6 | 6 | 15.3 | 35.5 | 0.45 | 0.04 | 0.05 | 0.07 | 7.4 | 1.4 | 1.30 | 5.63 |
|  | SCM | Feb | 6 | 1 | 14.1 | 35.8 | 1.73 | 0.20 | 0.04 | 0.18 | 11.2 | 3.5 | 0.88 | 2.28 |
|  |  |  | 6 | 3 | 14.1 | 35.8 | 1.60 | 0.19 | 0.02 | 0.15 | 11.7 | 2.9 | 1.22 | 3.18 |
|  |  |  | 6 | 6 | 14.1 | 35.8 | 1.13 | 0.18 | 0.12 | 0.16 | 9.2 | 2.9 | 2.39 | 3.49 |
|  |  | Apr | 6 | 1 | 13.3 | 35.7 | 1.63 | 0.31 | 0.10 | 0.18 | 11.5 | 3.2 | 1.61 | 5.38 |
|  |  |  | 6 | 3 | 13.3 | 35.7 | 1.45 | 0.33 | 0.12 | 0.16 | 11.9 | 2.4 | 1.50 | 6.96 |
|  |  |  | 6 | 6 | 13.7 | 35.6 | 0.03 | 0.06 | 0.07 | 0.05 | 3.0 | 1.9 | 1.45 | 11.74 |
|  |  | Aug | 6 | 1 | 14.9 | 35.6 | 0.00 | 0.04 | 0.10 | 0.03 | 4.2 | 1.4 | 0.84 | 26.55 |
|  |  |  | 6 | 3 | 16.0 | 35.6 | 0.27 | 0.00 | 0.07 | 0.05 | 6.5 | 2.8 | 1.11 | 6.04 |
|  |  |  | 6 | 6 | 15.4 | 35.6 | 0.35 | 0.06 | 0.06 | 0.07 | 6.5 | 1.7 | 1.41 | 5.45 |

**Figure S1:** A non-metric multi-dimensional scaling (MDS) showing the distance according to similarity in the microbial plankton composition at the beginning of each experiment (each symbol). Filled and open symbols represent samples from coastal and oceanic station, respectively, numbers correspond to the sampling station, triangles and circles represent samples from surface and SCM, respectively, and colours correspond to the months: (green) February, (blue) April and (pink) August. SCM: sub-surface chlorophyll maximum.

**Figure S2:** Response ratio (RR) to inorganic nutrient addition (averaged biomass at the end of the experiments divided by the averaged value in the control) of total phytoplankton community (smooth bars) and of prokaryote biomass (PB) (striped bars) at (a) coastal and (b) oceanic station. Each bar corresponds to one of the 3 experiments (a, b or c) performed in each depth and station during February, April and August. Colours represent samples from (light grey) surface (surf) and (dark grey) SCM. Horizontal line represents a response equal to 1, which implies no change relative to control. Asterisks indicate phytoplankton significant response (t-test; * $p < 0.05$) and circle indicate bacterial significant response (t-test; [0] $p < 0.05$). Note that different scales were used. Note that y-axis in Fig. S2 b is broken. SCM: sub-surface chlorophyll maximum.

**Figure S3:** Response ratio (RR) of total phytoplankton at surface and SCM in the coastal station and at surface and SCM in the oceanic waters in (a-d) February, (e-h) April and (i-l) August. Treatments represented are: B12/C; B1/C; B12+B1/C in pink tones and I+B12/I; I+B1/I; I+B12+B1/I in green tones. Pink symbols represent primary responses to B vitamins and green symbols represent secondary responses to B vitamins. Horizontal dotted-line represents a response equal to 1, that means no change relative to control in the primary responses, and no change relative to inorganic treatment in the secondary responses. Asterisks indicate phytoplankton significant response (t-test; * p < 0.05). Note that the y-axis is broken in *a* and *b.*

**Figure S4:** Response ratio (RR) of prokaryote biomass at surface and SCM in the coastal station and at surface and SCM in the oceanic waters in (a-d) February, (e-h) April and (i-l) August. Treatments represented are: B12/C; B1/C; B12+B1/C in pink tones and

I+B12/I; I+B1/I; I+B12+B1/I in green tones. Pink symbols represent primary responses to B vitamins and green symbols represent secondary responses to B vitamins.

Horizontal dotted-line represents a response equal to 1, that means no change relative to control in the primary responses, and no change relative to inorganic treatment in the secondary responses. Asterisks indicate prokaryote significant response (t-test; * p <

0.05).

Figure S1

[Figure]

Figure S2

[Figure]

**Chlorophyll-a Responses**

[Figure]

**Prokaryote Biomass Responses**

---

## Author Response (AR4)

Dear Editor,

We are grateful that you have appreciated the effort to improve this work.

Your technical corrections and comments are in plain font and our responses are in italic and blue font. The revised version of the manuscript with marked changes is also provided.

L19: Please use the math symbol ×, not the alphabet x.

*This has been corrected (L19)*

L38: Add a period immediately after spp (i.e., spp.).

*This has been corrected (L38)*

L76: Insert a space immediately before Fuhrman.

*This has been corrected (L76)*

L169–170: Please cite a reference for the absorption coefficient of pure Chl-a standard.

*This has been corrected (L170)*

L178–179: Gasol and Del Giorgio (2000)

*This has been corrected (L179)*

L189: The symbol of minus is different from those at L198 and L203. So please amend it.

*This has been corrected (L189)*

L245–L247: Please cite references for the forward and reverse primer pairs for prokaryotes and eukaryotes.

*Both references has been included (L245;L247)*

L302: Not chl-a, but Chl-a.

*This has been corrected (L302)*

L311: Please use $R^2$ (cf. see L1037).

*This has been corrected (L311)*

L468: Use a semi-colon between 2012 and Barber-Lluch.

*This has been corrected (L468)*

L536: Use an en dash for 0.1–10.

*This has been corrected (L536)*

L982: inorganic

*This has been corrected (L996)*

L1005: The "a" in Chlorophyll-a should be italic.

*This has been corrected (L1020)*

Figures 2 and 5: The "a" in Chl-a should be italic following the text.

*This has been changed*

*Additionally, references has been revised and Table 1 has been included in the text (L985).*

[revised manuscript text omitted]

Fig. 03

[Figure]

[Figure]

Fig. 04

[Figure]

[Figure]

**(a)**

Surface Coastal station

PB (μg C l⁻¹)

**(b)**

SCM Coastal station

T0
C
I
B12
B1
B12+B1
I+B12
I+B1
I+B12+B1

PB (μg C l⁻¹)

a --------February-------- b --------c
a -----------April-----------
a ---------August---------

**(c)**

Surface Oceanic station

T0
C
I
B12
B1
B12+B1
I+B12
I+B1
I+B12+B1

PB (μg C l⁻¹)

**(d)**

SCM Oceanic station

PB (μg C l⁻¹)

a ----------February---------- b c
a -----------April------------
a ---------August---------

[Figure]

Fig. 07

Fig. 08

[Figure]